# Distributionally Robust Causal Abstractions

**Yorgos Felekis** [1]   **Theodoros Damoulas** [1 2]   **Paris Giampouras** [1]

## Abstract

Causal Abstraction (CA) theory provides a principled framework for relating causal models that describe the same system at different levels of granularity while ensuring interventional consistency between them. Recent methods for learning CAs, however, assume fixed and well-specified exogenous distributions, leaving them vulnerable to environmental shifts and model misspecification. In this work, we address these limitations by introducing the first class of distributionally robust CAs and their associated learning algorithms. The latter cast robust causal abstraction learning as a constrained min-max optimization problem with Wasserstein ambiguity sets. We provide theoretical guarantees for both empirical and Gaussian environments, enabling principled selection of ambiguity-set radii and establish quantitative guarantees on worst-case abstraction error. Furthermore, we present empirical evidence across different problems and CA learning methods, demonstrating our framework's robustness not only to environmental shifts but also to structural and intervention mapping misspecification.

## 1. Introduction

Causal reasoning provides the foundation for understanding and influencing complex systems: it enables us to move beyond correlation to estimate the effects of interventions, simulate counterfactual scenarios, and uncover the underlying mechanisms that drive a system's behavior. These capabilities are central to out-of-distribution generalization and to building interpretable, fair, and socially aligned machine learning systems. Motivated by this, Causal Representation Learning (Schölkopf et al., 2021) seeks latent representations that both compress data and preserve the causal structure needed for robust reasoning. This challenge is particularly apparent in systems that naturally operate at multiple levels of abstraction, such as cells versus organs, neurons versus brain regions, or individuals versus populations, where causal relationships may differ across scales. While fine-grained models are often more expressive, they tend to be opaque and computationally expensive; more abstract models are interpretable and efficient, but only faithful if they preserve the causal semantics of the systems they simplify. This trade-off motivates the need for principled causal reasoning across abstraction levels, allowing reliable reasoning at both detailed and aggregate views of a system.

The theory of Causal Abstractions (CAs) addresses this challenge by formalizing causally consistent models at different granularities while also characterizing the maps that relates them. Two main frameworks have shaped this space: $\tau$-$\omega$ abstractions (Rubenstein et al., 2017; Beckers & Halpern, 2019), which define a single mapping between model domains, and $(R, a, \alpha)$ abstractions (Rischel, 2020), which separate mappings between variables and their domains. Their relation was recently analyzed in (Schooltink & Zennaro, 2025). Building on this, Causal Abstraction Learning (CAL) (Zennaro et al., 2022) aims to learn abstraction maps directly from data, enabling cross-scale representation learning, evidence integration from heterogeneous sources, and more efficient modeling.

Early CAL methods established distinct computational frameworks: joint differentiable programming for $(R, a, \alpha)$ abstractions (Zennaro et al., 2023) and multi-marginal Optimal Transport for the $\tau$-$\omega$ framework (Felekis et al., 2024), followed by several approaches, see Fig. 1, for learning approximate abstractions (D'Acunto et al., 2025; Kekić et al., 2024; Xia & Bareinboim, 2024). Parallel efforts have applied CAs to diverse domains, ranging from climate modeling (Chalupka et al., 2017) and neural network interpretability (Geiger et al., 2021) to agent-based modeling (Dyer et al., 2024), bandits (Zennaro et al., 2024), and causal discovery (Massidda et al., 2024).

A key challenge in CAL is understanding how the environmental conditions, i.e. the exogenous distributions under which an abstraction is learned, influence its generalization and causal consistency properties. While exact abstractions

[1]Department of Computer Science, University of Warwick, Coventry, UK [2]Department of Statistics, University of Warwick, Coventry, UK. Correspondence to: Yorgos Felekis <yorgos.felekis@warwick.ac.uk>.

*Proceedings of the 43rd International Conference on Machine Learning*, Seoul, South Korea. PMLR 306, 2026. Copyright 2026 by the author(s).

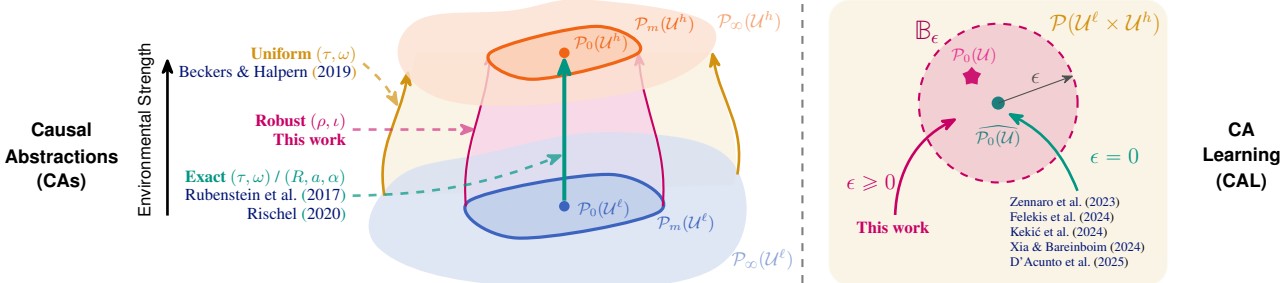

*Figure 1.* **Left:** Hierarchy of CAs based on the *environmental assumptions* required for the abstraction to be valid. Exact CAs assume consistency in a single joint environment, Uniform CAs require consistency across all possible joint environments while our framework of Robust CAs requires the mapping to hold over a constrained subset of relevant joint environments. **Right:** CAL methods positioned relative to this hierarchy on the joint environment $\mathcal{P}(\mathcal{U}^\ell \times \mathcal{U}^h)$. Our framework models environmental uncertainty over a subset of this using a Wasserstein ball $\mathbb{B}_\epsilon$ with $\epsilon \geqslant 0$, around an empirical joint environment $\widehat{\mathcal{P}_0(\mathcal{U})}$. Prior approaches correspond to $\epsilon = 0$, implicitly assuming a fixed environment to learn an approximate CA.

(Rubenstein et al., 2017; Rischel, 2020) assume a fixed environment without explicitly addressing how the abstraction depends on it, Beckers & Halpern (2019) propose a notion of abstraction uniformity that requires causal consistency to hold across *all* infinitely many environments; see Fig. 1. The latter corresponds to a mapping that captures a more fundamental relationship between models: that of their respective deterministic parts; i.e., causal bases (see Definition 2.1). While theoretically stronger, learning such an abstraction is computationally infeasible requiring data from infinite, potentially unattainable environments due to physical or practical constraints.

In this work, we address this challenge by introducing a framework of *robust causal abstractions* that require consistency across a constrained set of relevant environments, providing a notion theoretically stronger than exact abstractions (fixed environment) yet more flexible and tractable than uniform ones (all environments). In the finite sample case, our approach links the level of modeled uncertainty to the strength of the learned abstraction: the broader the ambiguity set, the more powerful the abstraction it approximates.

Overall, we make the following contributions:

- We introduce the first class of *robust causal abstractions*, called $(\rho, \iota)$-abstractions, that requires consistency across a constrained set of environments.

- We propose *Distributionally Robust Causal Abstractions* (DiRoCA), the first framework for learning robust to environmental shifts and misspecifications CAs in Additive Noise SCMs via distributionally robust optimization over Wasserstein ambiguity sets.

- We derive theoretical concentration bounds to guide ambiguity radius selection and provide provable robustness guarantees.

- We demonstrate the effectiveness of DiRoCA across diverse problems and settings, outperforming existing methods and baselines under environmental shifts, and different types of misspecifications.

## 2. Background

We work with Pearl (2009)'s Structural Causal Models framework, where each variable is a function of its direct causes and the exogenous noise.

**Definition 2.1** (Structural Causal Model). A $d$-dimensional *Structural Causal Model (SCM)* is a pair $\mathcal{M}^d \coloneqq (\mathcal{S}^d, \rho^d)$, where $\mathcal{S}^d = \langle \mathcal{X}, \mathcal{U}, \mathcal{F} \rangle$ defines the deterministic *causal basis*, consisting of a set of $d$ endogenous variables $\mathcal{X}$, a set of $d$ exogenous variables $\mathcal{U}$ and a set of $d$ structural functions $\mathcal{F}$, each defining the value of an endogenous variable as $X_i = f_i(\text{PA}(X_i), U_i) \quad \forall \, i \in [d]$, where $\text{PA}(X_i) \subseteq \mathcal{X} \backslash X_i$ denotes the direct causes (parents) of $X_i$. The *environment* $\rho^d$ is a joint probability distribution over $\mathcal{U}$.

**Causal Assumptions.** We focus on *Markovian* SCMs, which entail two key properties: (1) *Acyclicity*, meaning $\mathcal{M}^d$ entails a directed acyclic graph (DAG) $\mathcal{G}_{\mathcal{M}^d}$ whose nodes are the endogenous variables $\mathcal{X}$ and and whose edges follow the signatures of the functions $f_i$ in $\mathcal{F}$; and (2) *Joint Independence* of the exogenous variables, i.e., $\rho^d = \prod_{i=1}^d \mathbb{P}(U_i)$. This independence implies *causal sufficiency*, ensuring that there are no unobserved confounders interacting with the system. Furthermore, we assume *faithfulness*, meaning that independencies in the data are captured in the graphical model. Acyclicity allows us to recursively compose the structural functions into a single deterministic map $\mathbf{g} : \text{dom}[\mathcal{U}] \to \text{dom}[\mathcal{X}]$, referred to as the *mixing function*. This defines the SCM's *reduced form* $\mathcal{X} = \mathbf{g}(\mathcal{U})$, where endogenous variables are expressed purely in terms of exogenous noise. Consequently, the induced distribution is the pushforward $\mathbb{P}_{\mathcal{M}^d}(\mathcal{X}) = \mathbf{g}_\#(\rho^d)$,

making explicit the generative process by which exogenous uncertainty propagates through the model.

We specifically consider *Additive Noise Models (ANMs)*, where structural assignments take the form

$$X_i = f_i(\text{PA}(X_i)) + U_i, \quad \forall i \in [d] \tag{1}$$

This yields the general decomposition $\mathcal{X} = \mathcal{D} + \mathcal{U}$, where $\mathcal{D} = (f_i(\text{PA}(X_i)))_{i=1}^d$ is the deterministic part and $\mathcal{U} = (U_i)_{i=1}^d$ is the stochastic part (see Appendix A.5). A special case is the *Linear ANMs (LANs)*, where $\mathcal{X} = \mathbf{B}^\top \mathcal{X} + \mathcal{U}$. Due to acyclicity, $\mathbf{B}$ is a weighted adjacency matrix (permutable to strictly upper triangular), and the reduced form of the SCM becomes $\mathcal{X} = \mathbf{M}\mathcal{U}$, with mixing matrix $\mathbf{M} := (I - \mathbf{B}^\top)^{-1}$.

**Interventions.** SCMs facilitate reasoning about *interventions*. An exact intervention $\iota = \text{do}(\mathcal{A} = \mathbf{a})$ fixes variables $\mathcal{A} \subseteq \mathcal{X}$ to values $\mathbf{a}$, while allowing the rest of the system to evolve as usual. Graphically, this *mutilates* $\mathcal{G}_{\mathcal{M}^d}$ by removing incoming edges to $\mathcal{A}$. This yields a post-interventional SCM $\mathcal{M}_\iota^d$ with joint distribution $\mathbb{P}_{\mathcal{M}_\iota^d}(\mathcal{X})$.

**Causal Abstractions.** Causal Abstraction (CA) theory formalizes the relation between SCMs defined at different levels of granularity by requiring *interventional consistency* between them. This enables flexible shifts in the level of representation used for causal reasoning, depending on the specific inquiry or the nature of the available data.

**Definition 2.2** ((Rubenstein et al., 2017)). Let SCMs $\mathcal{M}^\ell$ and $\mathcal{M}^h$ with fixed respective environments $\rho^\ell$, $\rho^h$ and intervention sets $\mathcal{I}^\ell$, $\mathcal{I}^h$ and a surjective and order-preserving map $\omega : \mathcal{I}^\ell \to \mathcal{I}^h$. An abstraction $\tau : \text{dom}[\mathcal{X}^\ell] \to \text{dom}[\mathcal{X}^h]$, with $\ell \geqslant h$, is called an *exact transformation* if:

$$\tau_\#(\mathbb{P}_{\mathcal{M}_\iota^\ell}(\mathcal{X}^\ell)) = \mathbb{P}_{\mathcal{M}_{\omega(\iota)}^h}(\mathcal{X}^h), \quad \forall \iota \in \mathcal{I}^\ell \tag{2}$$

This is illustrated in Fig. 2 by comparing the intervene → abstract and abstract → intervene paths; their mismatch defines what is known as the *abstraction error*.

## 3. Distributionally Robust Causal Abstractions

In this section, we introduce a new class of CAs that extends exact transformations by ensuring interventional consistency of the two models across a set of environments. *This framework fills the gap between exact and uniform abstractions by introducing a stronger notion of abstraction that is environment-aware, enabling robustness to environmental misspecification and shifts during learning.* We explicitly model the uncertainty inherent in each SCM through its corresponding environment: $\rho^\ell$ for the low-level model $\mathcal{M}^\ell$ and $\rho^h$ for the high-level model $\mathcal{M}^h$. Assuming independence between them, we define the *joint environment*

$\boldsymbol{\rho} = \rho^\ell \otimes \rho^h$, a product measure that captures the uncertainty across abstraction levels. Formally, $\boldsymbol{\rho} \in \mathcal{P}(\mathcal{U})$, where $\mathcal{U} = \text{dom}[\mathcal{U}^\ell] \times \text{dom}[\mathcal{U}^h]$ denotes the product measure space of the exogenous domains.

In practice, the realization of certain environments is infeasible or unrealistic. Hence, we introduce the notion of *relevant environments*, denoted $\mathcal{A}^\ell \subseteq \mathcal{P}(\mathcal{U}^\ell)$ and $\mathcal{A}^h \subseteq \mathcal{P}(\mathcal{U}^h)$ for the low- and high-level SCMs, respectively. Accordingly, we define the *relevant joint environment space* as:

$$\mathcal{A} = \mathcal{A}^\ell \otimes \mathcal{A}^h, \tag{3}$$

which represents all joint distributions formed by pairing any $\rho^\ell \in \mathcal{A}^\ell$ with any $\rho^h \in \mathcal{A}^h$. This mirrors the concept of *relevant interventions* $\mathcal{I}$ that is a partially ordered set that restricts attention to interventions that are semantically meaningful or practically implementable. Furthermore, to align interventions across abstraction levels, we assume the existence of a surjective and order-preserving map $\omega : \mathcal{I}^\ell \to \mathcal{I}^h$. To ensure consistent alignment of joint interventions across levels, we define $\mathcal{I} = \{(\iota, \omega(\iota)) \mid \iota \in \mathcal{I}^\ell\}$, pairing each low-level intervention with its high-level counterpart. Finally, we define the *abstraction context* as the pair $(\mathcal{A}, \mathcal{I})$, which specifies the set of interventions and environments over which an abstraction is evaluated. Building on this setup, we introduce a refined notion of abstraction that operates on the reduced form of the SCM, making explicit the dependence of the abstraction on both the environment spaces and the intervention sets associated with the low- and high-level models.

**Definition 3.1** (($\rho, \iota$)-Abstraction). Let $(\mathcal{A}, \mathcal{I})$ be an abstraction context, where $\mathcal{M}^\ell = (\mathcal{S}^\ell, \rho^\ell)$, $\mathcal{M}^h = (\mathcal{S}^h, \rho^h)$ the low-level and high-level SCMs with $\ell \geqslant h$, and let $\mathbf{g}^\ell$ and $\mathbf{g}^h$ denote the respective mixing functions of their reduced forms. Then a map $\tau : \text{dom}[\mathcal{X}^\ell] \to \text{dom}[\mathcal{X}^h]$ is called a ($\rho, \iota$)-*abstraction* if, for all $\boldsymbol{\rho} = \rho^\ell \otimes \rho^h \in \mathcal{A}$ and for all $\boldsymbol{\iota} = (\iota, \omega(\iota)) \in \mathcal{I}$:

$$\tau_\# \left( \mathbf{g}_{\iota\#}^\ell(\rho^\ell) \right) = \mathbf{g}_{\omega(\iota)\#}^h(\rho^h) \tag{4}$$

A ($\rho, \iota$)-abstraction ensures commutativity between interventions and abstractions under a given joint environment $\boldsymbol{\rho} = \rho^\ell \otimes \rho^h$: intervening via $\iota$ and then abstracting via $\tau$ yields the same result as abstracting first and then intervening via $\omega(\iota)$ $\forall \iota \in \mathcal{I}^\ell$.

In general, we say that $\mathcal{M}^h$ and $\mathcal{M}^\ell$ are ($\rho, \iota$)-*interventionally consistent* if there exists an abstraction context $(\mathcal{A}, \mathcal{I})$ such that Eq. 4 holds. Furthermore, while the underlying SCMs may be non-linear, we focus on the case of *linear abstractions* (Massidda et al., 2024) between them, an important and well-behaved class of abstractions according to which the abstraction map can be represented through a matrix $\mathbf{T} \in \mathbb{R}^{h \times \ell}$ and thus the map $\tau$ can be represented as

a matrix-vector multiplication $\tau(x) = \mathrm{T}\,x \in X^h$, $x \in X^\ell$. Our $(\rho, \iota)$-framework generalizes and bridges prior notions

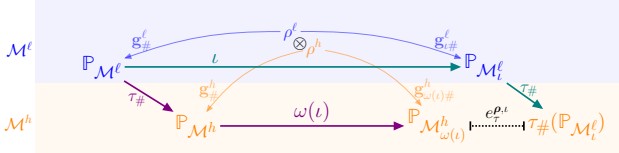

*Figure 2. Computation of the environment–intervention error.* The joint environment $\rho = \rho^\ell \otimes \rho^h$ captures the combined uncertainty from the low- and high-level SCMs. By pushing forward these components through the reduced forms $\mathbf{g}^\ell$ and $\mathbf{g}^h$ of the respective SCMs, we evaluate two interventional pathways: (a) apply an intervention $\iota$ to $\mathcal{M}^\ell$, then map the resulting distribution to the high-level space via $\tau_\#$: $\tau_\#(\mathbb{P}_{\mathcal{M}_\iota^\ell})$; and (b) first map $\mathcal{M}^\ell$ to $\mathcal{M}^h$ via $\tau_\#$, then apply the corresponding intervention $\omega(\iota)$: $\mathbb{P}_{\mathcal{M}_{\omega(\iota)}^h}$. The divergence $\mathcal{D}_{\mathcal{X}^h}$ between the resulting interventional distributions computes $e_\tau^{\rho,\iota}$. Aggregating $e_\tau^{\rho,\iota}$ over an $(\mathcal{A}, \mathcal{I})$ abstraction context, recovers the $(\mathcal{A}, \mathcal{I})$–abstraction error (Eq. 5). If zero, the diagram commutes and $\tau$ defines a $\tau$–0–approximate abstraction.

of causal abstraction. Unlike exact transformations (Rubenstein et al., 2017), which assume consistency under some fixed (but unspecified) environment, and uniform transformations (Beckers & Halpern, 2019), which require consistency across all environments, $(\rho, \iota)$ allows for a range of finite, plausible environments. This enables principled abstraction under realistic data constraints. Further discussion on this is in Appendix A.1.

**The $(\mathcal{A}, \mathcal{I})$–Abstraction Error.** Def. 3.1 suggests a perfect form of abstraction where interventional consistency holds exactly, implying that under a given abstraction context $(\mathcal{A}, \mathcal{I})$ using the high-level model entails no loss in predictive accuracy. However, in reality, such types of abstractions are rare. This stems in part from the very nature of abstraction, which reduces the size of the representation by disregarding minor differences, often leading to some degree of information loss. More importantly, in CAL, where the goal is to learn abstractions from data, all inferences are inherently approximate. Thus we need a notion of *abstraction error* (Beckers et al., 2020) tailored to our framework. The core idea is to measure the discrepancy between the two sides of Eq. 4 while accounting for the given context $(\mathcal{A}, \mathcal{I})$. To this end, we assume access to a metric over high-level interventional distributions, which induces a discrepancy between SCMs via $\tau$ enabling approximate abstractions. Specifically, we introduce a novel notion of abstraction error by aggregating over environments and interventions.

**Definition 3.2.** Let $\tau : \mathrm{dom}[\mathcal{X}^\ell] \to \mathrm{dom}[\mathcal{X}^h]$ be a map between the SCMs $\mathcal{M}^\ell$ and $\mathcal{M}^h$, defined wrt context $(\mathcal{A}, \mathcal{I})$. Let $D_{\mathcal{X}^h}$ also be a discrepancy measure over high-level distributions, and $q$ be a distribution over interventions in

$\mathcal{I}^\ell$. We define the $(\mathcal{A}, \mathcal{I})$-*Abstraction Error* as:

$$e_\tau(\mathcal{M}^\ell, \mathcal{M}^h) = \mathop{\mathfrak{g}}_{\rho \in \mathcal{A}} \mathop{\mathfrak{h}}_{\iota \sim q} \left[ e_\tau^{\rho,\iota}(\mathcal{M}^\ell, \mathcal{M}^h) \right], \quad (5)$$

where we define the *environment–intervention* approximation error induced by $\tau$ for a fixed $\rho \in \mathcal{A}$ and intervention $\iota \in \mathcal{I}^\ell$ as $e_\tau^{\rho,\iota}(\mathcal{M}^\ell, \mathcal{M}^h) = \mathcal{D}_{\mathcal{X}^h}\left( \tau_\#(\mathbf{g}_{\iota\#}^\ell(\rho^\ell)), \mathbf{g}_{\omega(\iota)\#}^h(\rho^h) \right)$.

The operators $\mathfrak{g}$ and $\mathfrak{h}$ aggregate the error over environments and interventions, respectively. Prior work typically fixes the environment, aggregating only over interventions via expectation (Felekis et al., 2024; Dyer et al., 2024; Kekić et al., 2024) or supremum (Zennaro et al., 2023). Beckers et al. (2020) proposed a similar notion of abstraction error, though their framework is defined under uniform abstractions and does not consider restrictions to subsets of the environment space, as we do. We call $\mathcal{M}^h$ an *approximate $(\rho, \iota)$-abstraction* if $e_\tau(\mathcal{M}^\ell, \mathcal{M}^h) > 0$ (see Figure 2). The connection between Def. 3.2 and $(\rho, \iota)$-abstractions now becomes apparent. In particular, when the total abstraction error vanishes for a given $\tau$, the pushforward of the low-level interventional distributions $\mathbf{g}_{\iota\#}^\ell(\rho^\ell)$ through $\tau_\#$ matches the corresponding high-level ones $\mathbf{g}_{\omega(\iota)\#}^h$, $q$-almost surely. The statement and proof of this are presented in Proposition A.1, Appendix A.4.

## 4. Learning Distributionally Robust Causal Abstractions

We learn *robust causal abstractions* when data is available from *a single environment per SCM*. Consider a pair $(\mathcal{M}^\ell, \mathcal{M}^h)$ of SCMs and their joint environment $\rho$ as before, representing the unknown data-generating process. We learn an abstraction map $\tau$ that reliably transforms low-level distributions into high-level ones, even in the presence of distributional shifts or test-time noise. We build on Wasserstein *Distributionally Robust Optimization (DRO)* (Kuhn et al., 2019), which provides robustness against distributional misspecification. The standard objective is:

$$\inf_{x \in \mathcal{X}} \sup_{\mathbb{Q} \in \mathbb{B}_{\varepsilon, p}(\widehat{\mathbb{P}}_N)} \mathbb{E}_{\xi \sim \mathbb{Q}}[f(x, \xi)], \quad (6)$$

The goal is to minimize the worst-case expected loss $f(x, \xi) : \mathbb{R}^n \times \Xi \to \mathbb{R}$, where $x \in \mathbb{R}^n$ denotes a decision variable, $\xi \in \Xi = \mathbb{R}^m$ a random vector representing uncertain data, over a set of plausible distributions, called the *ambiguity set*. In our setting, the loss is the *environment-intervention error* $e_\tau^{\rho,\iota}(\mathcal{M}^\ell, \mathcal{M}^h)$, the decision variable is $\tau$, and we minimize the worst-case abstraction error over a *2-Wasserstein product ambiguity set* centered at the empirical joint environment $\widehat{\rho}$ with radius $\epsilon$. More details on DRO are provided in Appendix A.3.

**Abduction.** Since we observe only endogenous variables but need to model uncertainty over the exogenous environment, we recover the unobserved noise realizations via *abduction*. Given endogenous samples $\mathbf{X}^d$ from $\mathcal{M}^d$ and its structural functions $\{\hat{f}_i\}_{i=1}^d$ (known for synthetic tasks, fit by regression otherwise), we treat the noise as the residual:

$$\hat{U}_i = X_i - \hat{f}_i(\mathrm{PA}(X_i)), \qquad i \in [d]. \tag{7}$$

Under an intervention $\mathrm{do}(X_j = a)$, the column for $X_j$ is fixed to $a$ and its residual to zero, preserving the additive identity. Stacking column-wise yields the $(D, U)$ decomposition $\mathbf{X}^{d,(\iota)} = \mathbf{D}^{d,(\iota)} + \mathbf{U}^d$, where $\mathbf{D}^{d,(\iota)}$ holds the deterministic mechanisms evaluated under intervention $\iota$ and $\mathbf{U}^d$ holds the inferred noise. When structural functions are known, abduction is exact; otherwise its quality depends on the regression accuracy (Appendix A.5.1).

**Environmental Robustness via $(\rho, \iota)$-Abstractions.** With $\hat{U}^\ell, \hat{U}^h$ in hand, we form the empirical marginal environments $\widehat{\rho^d} = \frac{1}{N}\sum_{i=1}^N \delta_{\hat{u}_i^d}$ and the empirical joint environment $\hat{\boldsymbol{\rho}} := \widehat{\rho^\ell} \otimes \widehat{\rho^h}$. To introduce distributional robustness, we instantiate the aggregation function $\mathfrak{g}$ in Eq. 5 over environments as a supremum and, for computational practicality, $\mathfrak{h}$ over interventions as an expectation. Following the DRO paradigm and since only one environment per SCM is available, we replace $\mathcal{P}(\mathcal{A})$ with a *joint ambiguity set*; a 2-Wasserstein product ball centered at $\hat{\boldsymbol{\rho}}$:

$$\mathbb{B}_{\epsilon,2}(\hat{\boldsymbol{\rho}}) := \mathbb{B}_{\varepsilon_\ell,2}(\widehat{\rho^\ell}) \times \mathbb{B}_{\varepsilon_h,2}(\widehat{\rho^h}), \tag{8}$$

where $\mathbb{B}_{\varepsilon_d,2}(\widehat{\rho^d}) = \{\rho \in \mathcal{P}(\mathcal{U}^d) : W_2(\rho, \widehat{\rho^d}) \leqslant \varepsilon_d\}$ is a 2-Wasserstein ball around $\widehat{\rho^d}$ for $d \in \{\ell, h\}$. The radius $\varepsilon_d$ controls the robustness level: larger values provide protection against broader shifts but may yield more conservative solutions. The role of $\mathbb{B}_{\epsilon,2}(\hat{\boldsymbol{\rho}})$ is to identify the worst-case joint environment $\boldsymbol{\rho}^\star = \rho^{\ell\star} \otimes \rho^{h\star}$ that maximizes the abstraction error; i.e. the most adversarial shift consistent with our finite-sample estimate. The full construction process is illustrated in Fig. 5 of Appendix A.6

**The DIROCA Objective.** Our objective is to learn a *linear abstraction map* $\mathrm{T}$, that remains reliable across all shifts inside $\mathbb{B}_{\epsilon,p}(\hat{\boldsymbol{\rho}})$. Assuming access to an abstraction context $(\mathcal{A}, \mathcal{I})$ and a divergence measure $\mathcal{D}_{\mathcal{X}^h}$ over high-level interventional distributions, we define the *distributionally robust causal abstraction learning objective* as follows:

$$\min_{\mathrm{T} \in \mathbb{R}^{h \times \ell}} \sup_{\boldsymbol{\rho} \in \mathbb{B}_{\epsilon,2}(\hat{\boldsymbol{\rho}})} \mathbb{E}_{\iota \sim q} \left[e_\tau^{\boldsymbol{\rho}, \iota}(\mathcal{M}^\ell, \mathcal{M}^h)\right] \tag{9}$$

where $\mathrm{T} \in \mathbb{R}^{h \times \ell}$ denotes the class of linear abstraction maps $\tau$, and $e_\tau^{\boldsymbol{\rho}, \iota}(\cdot, \cdot)$ is the environment–intervention error. This objective seeks the linear abstraction map that minimizes the worst-case expected abstraction error over the joint ambiguity set, ensuring robustness to distributional shifts.

**Remark 1.** *Both exact and uniform abstractions arise as special cases of our framework. Specifically, the robustness radius $\epsilon$ flexibly interpolates between these extremes depending on the desired level of robustness; i.e. $\epsilon \to 0$ approximates a $(\tau, \omega)$-transformation, while $\epsilon \to \infty$ approximates a uniform abstraction.*

**Concentration guarantees for the joint environment.** Our theoretical results below show that the true joint environment $\boldsymbol{\rho}$ lies within a 2-Wasserstein product ball centered at $\hat{\boldsymbol{\rho}}$ with high probability for an appropriate radius $\epsilon$. *This result offers a principled way to set the robustness parameter $\epsilon$ to ensure the ambiguity set reliably covers the true environment*, thus justifying our distributionally robust CA objective for independent environments.

**Assumption 1.** For $d \in \{\ell, h\}$, $\rho^d$ is a light-tailed environment; i.e. there exist constants $\alpha > 0$ and $A > 0$ such that $\mathbb{E}^{\rho^d}\left[\exp\left(\|\xi\|_2^\alpha\right)\right] \leqslant A$, where $\xi \sim \rho^d$.

Assumption 1 is the regularity condition required to apply the Wasserstein concentration inequalities of Kuhn et al. (Kuhn et al., 2019) to empirical measures. In our setting, these inequalities are applied separately to the low- and high-level marginal environments and are then combined through the product structure of the joint environment.

**Theorem 4.1** (Empirical $\boldsymbol{\rho}$-Concentration). *Let $\widehat{\rho^\ell}$ and $\widehat{\rho^h}$ be empirical distributions, under Assumption 1, from $N_\ell$ and $N_h$ i.i.d. samples, with $\boldsymbol{\rho} := \rho^\ell \otimes \rho^h$ and $\hat{\boldsymbol{\rho}} := \widehat{\rho^\ell} \otimes \widehat{\rho^h}$. For $d \in \{\ell, h\}$, there exist constants $c_{d,1}, c_{d,2} > 0$ depending only on the $d$-environment and confidence levels $\eta_d$. For any $\delta \in (0, 1]$ with $\delta = 1 - (1 - \eta_\ell)(1 - \eta_h)$, let $N_d(c, \eta) = \log(c_{d,1}/\eta)/c_{d,2}$. If we define:*

$$\tilde{\varepsilon}_d = \begin{cases} \left(\frac{\log(c_{d,1}/\eta)}{c_{d,2} N_d}\right)^{\min\{1/d, 1/2\}} & \text{if } N_d \geqslant N_d(c, \eta), \\ \left(\frac{\log(c_{d,1}/\eta)}{c_{d,2} N_d}\right)^{1/\alpha_d} & \text{otherwise,} \end{cases}$$

*then $\forall \epsilon \geqslant \sqrt{\tilde{\varepsilon}_\ell^2 + \tilde{\varepsilon}_h^2} \implies \mathbb{P}\left[\mathcal{W}_2(\boldsymbol{\rho}, \hat{\boldsymbol{\rho}}) \leqslant \epsilon\right] \geqslant 1 - \delta.$*

This concentration bound for CA shows how the robustness radius $\epsilon$ contracts with sample size to ensure high probability coverage of the true environment. In particular, $\epsilon = \mathcal{O}\left(N^{-1/d}\right)$ for dimension $d > 2$, where $N = \min(N_\ell, N_h)$ and $d = \max(\ell, h)$. Under finite samples, setting $\epsilon > 0$ yields a stronger abstraction, valid over the entire Wasserstein ball. An equivalent result for the case of Elliptical/Gaussian measures, alongside the proofs, can be found in Appendix A.4.

**Remark 2.** *When coefficients are estimated rather than known, the nominal environment $\hat{\boldsymbol{\rho}}$ may be misspecified; however, as the sample size grows, this error vanishes, and the concentration bounds remain asymptotically valid.*

**Remark 3.** *The constants appearing in Theorem 4.1 are distribution-dependent and generally unavailable in closed*

*form. Consequently, the theorem provides a coverage guarantee and the correct scaling of the ambiguity radius but not a parameter-free numerical optimizer for $\epsilon$. This distinction is standard in Wasserstein DRO: concentration results justify ambiguity sets as finite-sample confidence regions, while the numerical value of the radius is often calibrated empirically, for example, via holdout or cross-validation (Mohajerin Esfahani & Kuhn, 2018; Aolaritei et al., 2025; Dellaporta et al., 2025).*

**Optimization.** We now demonstrate how to solve the robust CA problem for ANMs as a min–max optimization. Given samples $X^{d,\iota} \in \mathbb{R}^{N \times d}$ for $\iota \in \mathcal{I}^d$, we extract environmental observational samples $U^d \in \mathbb{R}^{N \times d}$ via abduction. During training, reusing these observational residuals for interventional generation yields a consistent pairing of batch rows across interventions. This is not an alignment assumption but a computational device enabling Frobenius-based discrepancy evaluation and avoiding repeated optimal transport computations (see Appendix A.4). Under the ANM decomposition (Section 2), the endogenous samples for an intervention $\iota \in \mathcal{I}^\ell$ and level $d \in \{\ell, h\}$ are $X^{d,(\iota)} = D^{d,(\iota)} + U^d$. We define the empirical marginal environments as $\widehat{\rho}^d = 1/N \sum_{i=1}^N \delta_{\widehat{u}_i^d}$ for $d \in \{\ell, h\}$, and the empirical joint environment as $\widehat{\boldsymbol{\rho}} = \widehat{\rho^\ell} \otimes \widehat{\rho^h}$. By the finite-dimensional reduction of Kuhn et al. (2019), any distribution in a 2-Wasserstein ball around an empirical measure can be represented as a perturbed empirical distribution. We thus perturb the noise samples as $U^d \mapsto U^d + \Theta_d$, where $\Theta_d \in \mathbb{R}^{N \times d}$, which induces perturbed marginals $\widehat{\rho}^d(\Theta_d) := 1/N \sum_{i=1}^N \delta_{\widehat{u}_i^d + \theta_i^d}$. The ambiguity radii $\varepsilon_\ell, \varepsilon_h$ correspond to Frobenius budgets $r_d = \varepsilon_d \sqrt{N}$, such that the condition $\|\Theta_d\|_F \leqslant r_d, d \in \{\ell, h\}$ is equivalent to restricting the joint environment to $\mathbb{B}_{\epsilon,2}(\widehat{\boldsymbol{\rho}})$. To formulate the robust objective, we define the *nominal misalignment* for an abstraction $\mathrm{T} \in \mathbb{R}^{h \times \ell}$ as:

$$Z_{\mathrm{T}}^\iota(\mathbf{0}) := \mathrm{T}(D_\ell^{(\iota)} + U^\ell)^\top - (D_h^{(\omega(\iota))} + U^h). \quad (10)$$

Introducing perturbations $\boldsymbol{\Theta} := (\Theta_\ell, \Theta_h)$, the *perturbed misalignment* shifts to $Z_{\mathrm{T}}^\iota(\boldsymbol{\Theta}) := Z_{\mathrm{T}}^\iota(\mathbf{0}) + (\mathrm{T}\Theta_\ell^\top - \Theta_h)$. *The DiRoCA objective minimizes the expected perturbed misalignment under the worst-case observational shift within the ambiguity set*:

$$\min_{\mathrm{T}} \sup_{\|\Theta_\ell\|_F \leqslant r_\ell, \, \|\Theta_h\|_F \leqslant r_h} \mathbb{E}_{\iota \sim q}\big[\|Z_{\mathrm{T}}^\iota(\boldsymbol{\Theta})\|_F^2\big]. \quad (11)$$

This formulation highlights that the adversary attempts to align the perturbation shift $(\mathrm{T}\Theta_\ell^\top - \Theta_h)$ with the nominal direction $Z_{\mathrm{T}}^\iota(\mathbf{0})$ to maximize the error. We solve (11) via alternating projected gradient descent–ascent (Alg. 5).

**Specialized variants.** Equation 11 constitutes our primary method, applicable to *any* ANM. the reduced-form mixing

---

**Algorithm 1** General DiRoCA $\qquad\qquad [d \in \{\ell, h\}]$

**Require:** $\omega, \mathcal{I}^d, \mathcal{G}_{\mathcal{M}^d}, \{\mathbf{X}_\iota^d \in \mathbb{R}^{N \times d}\}_{\iota \in \mathcal{I}^d}, \varepsilon_d$.
**Require:** $\eta_\tau, \eta_\theta$ (learning rates), $k_{\min}, k_{\max}$ (inner steps).
1: *// Infer noise and initialize empirical environments*
2: $\mathbf{U}^d \leftarrow \mathrm{Abduct}(\mathbf{X}^d, \mathcal{M}^d); \quad \widehat{\rho}^d \leftarrow \frac{1}{N} \sum_{i=1}^N \delta_{\widehat{u}_i^d}$
3: **Initialize:** $\mathrm{T}^{(0)}, \boldsymbol{\Theta}^{(0)}, \rho^{d(0)} \leftarrow \widehat{\rho}^d, r_d \leftarrow \varepsilon_d \sqrt{N}$
4: $J(\mathrm{T}, \boldsymbol{\Theta}) := \mathbb{E}_\iota[\|Z_{\mathrm{T}}^\iota(\boldsymbol{\Theta})\|_F^2]$ $\qquad$ *#Objective*
5: **repeat until convergence:**
6: $\quad$ *// Projected Gradient Ascent for $k_{\max}$ steps*
7: $\quad \Theta_d^{(t+1)} \leftarrow \mathrm{Proj}_{\|\cdot\|_F \leqslant r_d}\Big[\Theta_d^{(t)} + \eta_\theta \nabla_{\Theta_d} J(\mathrm{T}^{(t)}, \boldsymbol{\Theta}^{(t)})\Big]$
8: $\quad \rho^{d(t+1)} \leftarrow \frac{1}{N} \sum_{i=1}^N \delta_{\widehat{u}_i^d + \theta_i^{d(t+1)}}$ $\quad$ *#Update worst-case*
9: $\quad$ *// Abstraction Gradient Descent for $k_{\min}$ steps*
10: $\quad \mathrm{T}^{(t+1)} \leftarrow \mathrm{T}^{(t)} - \eta_\tau \nabla_{\mathrm{T}} J(\mathrm{T}^{(t)}, \boldsymbol{\Theta}^{(t+1)})$ $\quad$ *#Update T*
$\quad$ **Return:** $\mathrm{T}^\star \in \mathbb{R}^{h \times \ell}, \boldsymbol{\rho}^\star = \rho^{\ell\star} \otimes \rho^{h\star}$

---

functions are explicit linear operators $\mathbf{L}_\iota \in \mathbb{R}^{\ell \times \ell}, \mathbf{H}_{\omega(\iota)} \in \mathbb{R}^{h \times h}$. We distinguish two variations of the DiRoCA objective based on the available environmental information:

**(a) Empirical:** When only samples are available we utilize the linear mixing matrices explicitly. The misalignment becomes a direct matrix operation: $Z_{\mathrm{T}}^\iota(\mathbf{0}) = \mathrm{T}\mathbf{L}_\iota U^{\ell\top} - \mathbf{H}_{\omega(\iota)} U^h$. The objective remains the Frobenius norm minimization from Eq. (11), but the explicit linearity allows for faster gradient computations as the deterministic part $D$ is folded into the matrix multiplication. The optimization follows Algorithm 5, projecting perturbations $\Theta$ onto Frobenius balls.

**(b) Gaussian:** If the environments are known to be Gaussian distributions, i.e., $\rho^d \sim \mathcal{N}(\mu^d, \Sigma^d)$, we exploit the property that Gaussianity is preserved under the linear transformations $\mathbf{L}_\iota$ and $\mathbf{H}_{\omega(\iota)}$ and T. Here, instead of explicit sample perturbation, we define the ambiguity set directly over the Gaussian parameters $(\mu, \Sigma)$ using the Gelbrich distance (see also Appendix A.2):

$$\sqrt{\|m_1 - m_2\|_2^2 + \mathrm{Tr}\left(\Sigma_1 + \Sigma_2 - 2\left(\Sigma_1^{1/2}\Sigma_2\Sigma_1^{1/2}\right)^{1/2}\right)} \quad (12)$$

for $\mathcal{N}(m_1, \Sigma_1)$ and $\mathcal{N}(m_2, \Sigma_2)$. The robust objective simplifies to minimizing the worst-case Gelbrich distance misalignment $Z_{\mathrm{T}}^\iota(\mathbf{0})$ between $\mathcal{N}(\mathrm{T}\mathbf{L}_\iota \mu^\ell, \mathrm{T}\mathbf{L}_\iota \Sigma^\ell \mathbf{L}_\iota^\top \mathrm{T}^\top)$ and $\mathcal{N}(\mathbf{H}_{\omega(\iota)} \mu^h, \mathbf{H}_{\omega(\iota)} \Sigma^h \mathbf{H}_{\omega(\iota)}^\top)$. We optimize via a proximal gradient ascent–descent scheme on the moments $(\mu, \Sigma)$, relaxing the non-smooth trace term via a variational upper bound. Illustrations of the resulting worst-case Gaussian environments appear in Appendix A.9. All DiRoCA variants and their optimization details appear in Appendix A.9.

Further, we establish worst-case guarantees for the learned T. While certified defenses in robust prediction typically

bound a classification radius (Cohen et al., 2019) or outlier influence (Altamirano et al., 2024), our framework certifies the abstraction error itself. Theorem 4.2 establishes that the solution $T^\star$ of Eq. 9 is provably robust, offering a closed form bound on the abstraction error for all perturbations subject to the Frobenius constraints The proof can be found in Appendix A.4.

**Theorem 4.2** (Provable Robustness). *Let* $T \in \mathbb{R}^{h \times \ell}$ *be an abstraction matrix and define its worst-case expected loss as:* $\zeta(T) \coloneqq \sup_{\|\Theta_\ell\|_F \leqslant r_\ell, \|\Theta_h\|_F \leqslant r_h} \mathbb{E}_{\iota \sim q}\big[\|Z_T^\iota(\Theta)\|_F^2\big].$

$$\implies \zeta(T) \leqslant \mathbb{E}_{\iota \sim q}\Big[\big(r_h + \|Z_T^\iota(\mathbf{0})\|_F + r_\ell \|T\|_2\big)^2\Big]. \tag{13}$$

*Consequently, the worst-case abstraction error of any minimizer* $T^\star$ *of* (11) *is upper bounded by the RHS at* $T^\star$.

## 5. Experimental Results

We empirically validate DiRoCA on a diverse suite of synthetic, semi-synthetic, and real-world benchmarks against CA learning methods. Beyond robustness to distributional shifts of the environment, we also test resilience to structural misspecification, errors in the intervention mapping $\omega$, and high-dimensional semantic perturbations.

**Datasets.** We consider four datasets: **(a)** the SLC dataset, a three-variable LAN SCM whose abstraction removes a mediator node; **(b)** LUCAS [1]: a lung cancer diagnosis simulation ($\ell = 6, h = 3$) in linear (LiLUCAS) and nonlinear (nLUCAS) variants; **(c)** the real-world *Electric Battery Manufacturing* (EBM) dataset (Zennaro et al., 2023), where two labs (WMG and LRCS) observe the same lithium-ion manufacturing process at different spatial granularities: the low-level SCM models a control parameter (comma gap) causing mass loading measurements at two locations, while the high-level SCM aggregates these into a single scalar, and the abstraction must remain consistent under real interventions on the control parameter; and **(d)** the semi-synthetic *Colored MNIST* (cMNIST) dataset (Xia & Bareinboim, 2024), built from MNIST with confounded digit-color pairs, where we abstract from $32 \times 32$ RGB images to a 64-dimensional disentangled latent space defined by the pre-trained encoder of the original work, which serves as the ground truth; interventions break the spurious digit-color correlations, exposing the causal mechanism the abstraction must recover. We assume known DAGs, intervention sets, and map $\omega$; structural functions are known for synthetic tasks but estimated for EBM and cMNIST. Complete results, evaluation details, a roadmap of experiments, and dataset analyses are in Appendices A.8 and A.7; code is available at the DiRoCA codebase.

[1]https://www.causality.inf.ethz.ch/data/LUCAS.html

**Baselines.** We compare DiRoCA against the following methods: **(a)** $\text{BARY}_{(\tau,\omega)}$, a variation of the $\text{BARY}_{\text{OT}}$ of (Felekis et al., 2024), which learns an abstraction between the barycenters of the interventional distributions of both abstraction levels; **(b)** $\text{GRAD}_{(\tau,\omega)}$ that ignores environmental uncertainty and approximates a $(\tau, \omega)$ transformation via gradient descent; corresponding to DiRoCA with $(\varepsilon_\ell, \varepsilon_h) = (0, 0)$ as discussed in Remark 1, and for the empirical settings, we also compare against; **(c)** ABS-LiNGAM (Massidda et al., 2024): an observational least-squares method between low- and high-level samples baseline, evaluating both perfect (AbsLin$_\text{p}$) and noisy (AbsLin$_\text{n}$) CAL variants proposed in the original work.

**Evaluation.** We report two radius configurations for DiRoCA. First, DiRoCA$_{\widehat{\epsilon}}$ uses a default radius from Thm. 4.1 after fixing the unknown distribution-dependent constants to manually chosen default values; this configuration tests the behavior of the radius scale suggested by the finite-sample coverage result. Second, DiRoCA$_{\epsilon^\star}$ denotes the best-performing radius in each setting, selected post hoc from a pre-specified grid. Thus, $\epsilon^\star$ should not be interpreted as a theoretically optimal radius. Results for additional robustness radii are deferred to Appendix A.8. For nonlinear SCMs, we make no parametric assumptions on the environment. For linear SCMs, we evaluate both empirical and Gaussian environments, always using the empirical approximation of the $(\mathcal{A}, \mathcal{I})$-abstraction error in Eq. (5). We instantiate $\mathfrak{g}$ and $\mathfrak{h}$ as $\mathbb{E}$ and report mean $\pm$ std abstraction error, with paired $t$-tests ($p < 0.05$). We use 5-fold cross-validation for all experiments, except for EBM where a $90/10$ train–test split is used due to limited data. Structural functions are known for synthetic datasets and estimated for EBM and cMNIST (details in Appendix A.7). For cMNIST, robustness is enforced only in the low-level/pixel space, reflecting the sensory nature of the shifts; i.e. $\varepsilon_h = 0$. Finally, at test time, abstractions are always evaluated on independently generated samples and not on the shared observational noise across interventions as in training.

We perform a robustness analysis using a unified evaluation framework based on a Huber contamination model. This is applied to the held-out test set for each of the $k$ cross-validation folds, controlled by two key parameters: the contamination fraction $\alpha \in [0, 1]$, which controls the *prevalence* of shifted samples, going from a fully clean test set at $\alpha = 0$ to a fully shifted one at $\alpha = 1$ and the contamination strength $\tilde{\sigma}$, which controls the magnitude of their shift. Each test set consists of a collection of paired, clean endogenous datasets corresponding to each intervention $\{(X_\iota^\ell, X_{\omega(\iota)}^h)\}_{\iota \in \mathcal{I}_\ell} \in \mathbb{R}^{N_\text{test}^\ell \times \ell} \times \mathbb{R}^{N_\text{test}^h \times h}$. For each intervention $\iota$, we first generate a noisy version of the data, $\tilde{X}_\iota^d$, by applying an additive stochastic shift to the clean data. This is achieved by creating a noise matrix $\mathbf{N} \in \mathbb{R}^{N_\text{test}^d \times d}$, where

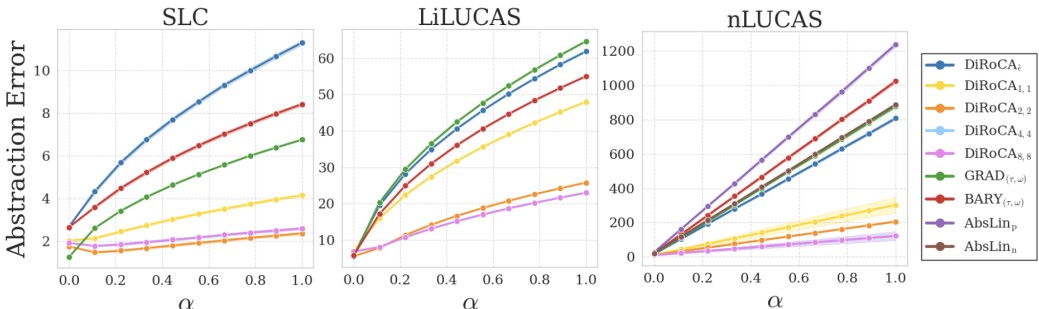

*Figure 3.* Robustness to outlier fraction ($\alpha$) on the SLC (Gaussian) and LiLUCAS (Gaussian) and nLUCAS experiments, evaluated at fixed Gaussian noise intensity ($\tilde{\sigma} = 5.0$ for SLC and $\tilde{\sigma} = 10.0$ for LiLUCAS and nLUCAS). DiRoCA, especially with tuned ambiguity radius, achieves consistently lower error as the proportion of outlier environments increases, while non-robust methods degrade. $(\hat{\varepsilon}_\ell, \hat{\varepsilon}_h) = (0.11, 0.11)$ for SLC and LiLUCAS, and $(0.47, 0.45)$ for nLUCAS. DiRoCA$_{4,4}$ and DiRoCA$_{8,8}$ are numerically identical.

each row $\mathbf{n}_i \sim \mathcal{N}(0, \tilde{\sigma}^2 \cdot \mathbf{I})$ is an independent sample from a zero-mean Gaussian distribution with a scaled covariance and thus, $\tilde{X}_\iota^d = X_\iota^d + \mathbf{N}$. While we focus on Gaussian shifts in the main text for clarity, we present analogous results for Student-t and Exponential noise distributions in Appendix A.8. The final contaminated test sets $\bar{X}_\iota^d$ for every $\iota \in \mathcal{I}^d$ are then formed as:

$$\bar{X}_{\iota,i}^d = (1 - \mathbb{1}_{i \in \mathcal{S}}) X_{\iota,i}^d + \mathbb{1}_{i \in \mathcal{S}} \tilde{X}_{\iota,i}^d, \quad (14)$$

where $\mathcal{S}_\iota^d \subseteq [N_{\text{test}}^d]$ is a set of $|\mathcal{S}_\iota^d| = \lfloor \alpha N_{\text{test}}^d \rfloor$ row indices drawn uniformly without replacement. Although DiRoCA targets environmental shifts, we contaminate the endogenous samples directly, which is a valid proxy as additive exogenous shifts in ANMs propagate additively to endogenous variables. Performance is measured by the squared Frobenius norm for empirical settings and the Wasserstein distance between fitted Gaussians $\mathcal{N}(\hat{\mu}, \hat{\Sigma})$ for linear settings, averaged across all folds, samples, and interventions.

**Analysis.** We analyze robustness by varying the outlier fraction $\alpha$ at fixed noise intensity $\tilde{\sigma}$ (Fig. 3). Consistent trends are observed when varying noise intensity $\tilde{\sigma}$ at fixed $\alpha$. In linear synthetic settings (SLC, LiLUCAS), where the noise is Gaussian (explicitly or implicitly), we observe a clear crossover: at low contamination ($\alpha \approx 0$), the non-robust GRAD$_{(\tau,\omega)}$ marginally outperforms DiRoCA, reflecting the *cost of robustness*: DiRoCA's adversarial training is conservative, prioritizing worst-case safety over best-case precision. However, as $\alpha$ increases, the error of non-robust baselines grows rapidly, whereas DiRoCA maintains a lower error profile under increasing shifts. In nLUCAS setting, this trade-off collapses: DiRoCA outperforms GRAD$_{(\tau,\omega)}$ even at $\alpha = 0$. This occurs because nonlinear mechanisms induce non-Gaussian data geometries, causing GRAD$_{(\tau,\omega)}$ to overfit the nominal environment and fail to generalize to new samples, whereas DiRoCA's environment-level optimization regularizes against this effect (Note in Appendix A.8). For the real-world datasets (EBM, cMNIST), we report results for $\alpha = 1$ in the main text (Table 1), with

*Table 1.* Average abstraction error under distribution shift, full corruption ($\alpha = 1$) for EBM and cMNIST. $(\hat{\varepsilon}_\ell, \hat{\varepsilon}_h) = (0.55, 0.41)$ for EBM and $(61.69, 0)$ for cMNIST, while $\epsilon^\star$ corresponds to $(1, 1)$ for EBM and $(20, 0)$ for cMNIST.

| Method | EBM | cMNIST |
|---|---|---|
| AbsLin$_p$ | $3034.8 \pm 2129.3$ | $2951.3 \pm 51.7$ |
| AbsLin$_n$ | $3330.9 \pm 3238.0$ | $470.7 \pm 8.5$ |
| BARY$_{(\tau,\omega)}$ | $357.4 \pm 144.9$ | $15348.5 \pm 629.4$ |
| GRAD$_{(\tau,\omega)}$ | $320.2 \pm 148.2$ | $22200.7 \pm 1492.7$ |
| DiRoCA$_{\hat{\epsilon}}$ | $\mathbf{297.2 \pm 167.8}$ | $215.0 \pm 8.5$ |
| DiRoCA$_{\epsilon^\star}$ | $\mathbf{268.4 \pm 141.8}$ | $\mathbf{48.2 \pm 0.6}$ |

full robustness curves deferred in Appendix A.8. Here, the structural functions estimation introduces model misspecification in addition to stochastic noise. However, DiRoCA treats these structural residuals as a form of environmental perturbation: the same robustness mechanism that guards against distributional shift also absorbs deviations from the assumed functional form, a perspective aligned with recent approaches that model perturbations as sparse mechanism shifts (Schneider et al., 2025) and explains why it maintains stable performance even under severe shifts ($\alpha = 1$). Also, BARY$_{(\tau,\omega)}$ shows limited robustness via aggregation but lacks the adversarial training needed to withstand worst-case shifts, while ABSLIN fails due to observational reliance, confirming that nominal-environment tailored optimization fails to generalize under contamination.

**Beyond Environmental Robustness.** We also test robustness to violations of modeling assumptions, including: (i) Structural ($\mathcal{F}$), where we evaluate our linearly-trained models on new test data generated from SCMs with non-linear structural equations. These misspecified SCMs share the same causal graph and environments as those used during training, but the structural equation for each variable is non-linear, controlled by a strength parameter $k \in \mathbb{R}$; (ii) Intervention mapping ($\omega$), where we contaminate the

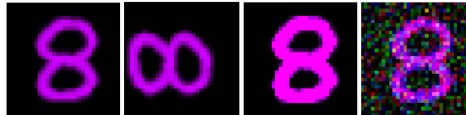

*Figure 4.* cMNIST camera shifts from left to right: Original, rotation, lighting and environmental ($\tilde{\sigma} = 0.5$).

ground-truth $\omega$ map by reassigning a subset of low-level interventions to different high-level ones of the same complexity, thereby preserving intervention dose while introducing realistic mapping errors; and (iii) Semantic ($\mathcal{S}$), where we utilize the cMNIST dataset to simulate unmodeled camera effects such as geometric rotation and photometric lighting shifts (see Figure 4). This setup introduces a severe structural misspecification: we task the models with learning a *linear* abstraction map from high-dimensional pixels, despite the underlying generative process being inherently non-linear. An explanation of these processes can be found in Appendix A.8.

Table 2 shows results for $\omega$- and $\mathcal{F}$-misspecification ($k = 1$ with a sinusoidal (sin) function) for the empirical LiLUCAS and nLUCAS. Full robustness curves illustrating the abstraction error as a function of the non-linearity strength $k$ (both sin and tanh) are provided in Appendix A.8. For the $\mathcal{S}$-misspecification task, Table 3 reports the Relative Squared Error under rotation (lighting shift results in Appendix A.8). We predict latents as $\hat{Z} = X_{\text{pix}} T^{\top} + b$, where $b = \mathbb{E}[Z - X_{\text{pix}} T^{\top}]$ is a test-time per-dimension mean-shift correction. This removes arbitrary intercept mismatch while leaving the learned linear map unchanged. Baselines' performance indicates a catastrophic failure to identify the causal signal whereas DIRoCA maintains a relatively low error. Despite the inherent misspecification of linear abstractions in this high-dimensional non-linear setting, DIRoCA's remarkable stability suggests it recovers a meaningful linear approximation of the underlying geometry, ignoring high-frequency features that are unstable under rotation. Overall, by acting as an implicit regularizer, DIRoCA's min–max objective extends robustness beyond environmental shifts, consistently outperforming baselines across all misspecification settings.

## 6. Conclusion

We introduced $(\rho, \iota)$-abstractions, a framework that bridges the gap between brittle exact abstractions and intractable uniform ones by enforcing consistency across a relevant set of environments. To learn these, we proposed DIRoCA, which casts CAL as a DRO problem. Our theoretical results extend DRO concentration bounds to the CA setting for the true joint environment to guide radius selection and establish provable robustness guarantees, offering a closed-

form analytical bound on the worst-case abstraction error. Experiments across different problems and prior art demonstrated a consistently lower abstraction error under both distributional shifts and structural misspecification. Notably, we observe that robustness to environmental shifts often induces resilience to broader sources of misspecification, such as imperfect structural assumptions or intervention mappings. The current framework is limited to linear abstractions and assumes access to both a known intervention map and the true causal DAGs or an accurate estimate via causal discovery. These present challenges that we aim to address in future work.

*Table 2.* Average abstraction error under misspecification. For LiLUCAS, we report both $\omega$- and $\mathcal{F}$-misspecification under the empirical setting. DIRoCA$_{\hat{\epsilon}}$ uses default radii instantiated from Thm. 4.1: $(\hat{\varepsilon}_\ell, \hat{\varepsilon}_h) = (0.11, 0.11)$ for LiLUCAS and $(0.47, 0.45)$ for nLUCAS. DIRoCA$_{\epsilon^\star}$ denotes the best-performing configuration, corresponding to $(4, 4)$ across all settings.

| Method | LiLUCAS | | nLUCAS |
| --- | --- | --- | --- |
| | $\omega$ | $\mathcal{F}$ | $\omega$ |
| BARY$_{(\tau,\omega)}$ | $548.6 \pm 2.5$ | $578.8 \pm 1.3$ | $21.4 \pm 1.0$ |
| GRAD$_{(\tau,\omega)}$ | $305.5 \pm 1.3$ | $338.7 \pm 1.0$ | $18.7 \pm 0.9$ |
| AbsLin$_{\text{p}}$ | $414.6 \pm 1.1$ | $430.3 \pm 1.2$ | $29.3 \pm 1.2$ |
| AbsLin$_{\text{n}}$ | $359.5 \pm 1.4$ | $381.3 \pm 1.9$ | $22.7 \pm 0.9$ |
| DIRoCA$_{\hat{\epsilon}}$ | $459.1 \pm 2.8$ | $500.1 \pm 1.4$ | $18.2 \pm 0.9$ |
| DIRoCA$_{\epsilon^\star}$ | $\mathbf{304.1 \pm 7.8}$ | $\mathbf{327.4 \pm 9.2}$ | $\mathbf{9.1 \pm 0.3}$ |

*Table 3.* Robustness to geometric rotation on cMNIST (Relative L2 %). DIRoCA$_{\hat{\epsilon}}$ uses default radii from Thm. 4.1. radii $(\hat{\varepsilon}_\ell, \hat{\varepsilon}_h) = (61.69, 0)$, while DIRoCA$_{\epsilon^\star}$ denotes the best-performing configuration, corresponding to $(20, 0)$ for both rotation levels.

| Method | Low ($30°$) | High ($90°$) |
| --- | --- | --- |
| BARY$_{(\tau,\omega)}$ | $2291.0 \pm 196.6$ | $4135.0 \pm 355.8$ |
| GRAD$_{(\tau,\omega)}$ | $1780.9 \pm 237.7$ | $3585.9 \pm 643.3$ |
| AbsLin$_{\text{p}}$ | $541.3 \pm 12.9$ | $604.8 \pm 24.6$ |
| AbsLin$_{\text{n}}$ | $276.7 \pm 6.5$ | $287.9 \pm 9.3$ |
| DIRoCA$_{\hat{\epsilon}}$ | $101.7 \pm 6.2$ | $106.4 \pm 3.7$ |
| DIRoCA$_{\epsilon^\star}$ | $\mathbf{28.0 \pm 0.4}$ | $\mathbf{30.7 \pm 0.7}$ |

## Acknowledgements

**YF** acknowledges support by the Onassis Foundation - Scholarship ID: F ZR 063-1/2021-2022. **TD** acknowledges support from a UKRI Turing AI acceleration Fellowship [EP/V02678X/1].

## Impact Statement

This paper presents work whose goal is to advance the field of Machine Learning. There are many potential societal consequences of our work, none of which we feel must be specifically highlighted here.

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

# A. Appendix

**Overview.** This appendix provides the theoretical foundations, formal proofs, and extended experimental details supporting the main text. In §A.1, we position our framework within the broader causal abstraction literature. We review the necessary mathematical background on the Wasserstein metric and Distributionally Robust Optimization in §A.2 and §A.3, respectively. §A.4 contains the formal proofs, including the concentration results for joint environments and the derivation of the closed-form robustness certificate. We then detail the practical implementation for general ANMs, describing the $(D, U)$ decomposition in §A.5 and the construction of the joint ambiguity set in §A.6. Comprehensive descriptions of the datasets, causal graphs, and intervention mappings are provided in §A.7. In §A.8, we present the full experimental roadmap and additional robustness analyses against model misspecification. §A.9 details the optimization procedures, providing the analytical derivations for both the linear (Gaussian and empirical) and the general DIROCA formulations. Finally, Table 17 summarizes the main mathematical notation used throughout the paper.

## A.1. Relation to Existing Causal Abstraction Frameworks.

Fig. 1 provides an overview of how existing causal abstraction (CA) frameworks and learning methods differ in their treatment of environmental variability. The left panel organizes CA frameworks by the subset of the joint environment space $\mathcal{P}(\boldsymbol{\mathcal{U}})$ over which they require interventional consistency.

The $(\rho, \iota)$ framework differs from the exact transformation formulation of Rubenstein et al. (2017), and the $(R, a, \alpha)$ of Rischel (2020), both of which implicitly assume a fixed environment $(\mathcal{P}_0(\boldsymbol{\mathcal{U}}))$ without addressing how it constrains or influences the abstraction. As a result, these frameworks may yield ill-defined forms of abstractions when applied beyond that fixed setting, as also noted by Beckers & Halpern (2019), limiting their practical utility. Our notion of $(\rho, \iota)$-abstractions also differs from the *uniform transformations* introduced by Beckers & Halpern (2019), which requires interventional consistency to hold across *all* possible pairs of environments $(\mathcal{P}_\infty(\boldsymbol{\mathcal{U}}))$. By design, uniform transformations are environment-independent: they relate the deterministic components of the SCMs; i.e., their deterministic causal bases without regard to the specific distributions that generate observational data. While uniform abstractions offer a theoretically stronger and philosophically valid notion of abstraction, they are practically infeasible to learn, as they assume access to an unbounded number of environments, many of which may be inaccessible due to physical, data, or knowledge constraints. The $(\rho, \iota)$ framework strikes a middle ground: it preserves the formal rigor of exact abstractions while remaining flexible and computationally tractable under real-world distributional limitations by requiring consistency over a constrained relevant subset $\mathcal{P}_m(\boldsymbol{\mathcal{U}})$, with $0 \leq m < \infty$. Just as the notion of relevant interventions restricts attention to those that can be meaningfully abstracted and implemented at the high level, we similarly constrain focus to environments that are plausible or meaningful within a given setting. Together, these relevant environments and interventions define the *abstraction context*, which determines the domain over which causal consistency is required.

As discussed, when exact abstractions are not achievable, consistency between models is relaxed to be approximate. However, in the CAL literature, various formulations of abstraction error (Eq. 5) have been proposed. The divergence term $\mathcal{D}_{\mathcal{X}^h}$ has been instantiated using either the Jensen–Shannon divergence (Zennaro et al., 2023; 2024) or the KL divergence (Dyer et al., 2024; Kekić et al., 2024; D'Acunto et al., 2025). Regarding the aggregation function $\mathfrak{h}$, initial frameworks adopted the maximum over interventions (Beckers et al., 2020; Rischel & Weichwald, 2021), a choice followed by many subsequent CAL works (Zennaro et al., 2023; 2024), while later approaches also considered the expectation over the intervention set (Felekis et al., 2024; Dyer et al., 2024; Kekić et al., 2024). Note that, especially in policy-making scenarios, some interventions matter more than others due to factors like the likelihood of implementation or cost, reflecting an implicit weighting over the intervention set. While prior works such as (Felekis et al., 2024; Dyer et al., 2024) assume this distribution to be uniform, assigning equal weight to all interventions, it can be adapted to reflect practical priorities or domain-specific preferences.

As for the aggregation function $\mathfrak{g}$, no prior CAL framework has explicitly addressed variability across environments of the two SCMs. The closest suggestion came from Beckers et al. (2020), who, within the setting of uniform abstractions, proposed taking an expectation over exogenous samples of the low-level model. Crucially, they showed the existence of a deterministic map $\tau_{\mathcal{U}}$ that maps low-level exogenous variables into their high-level counterparts. This result relies on a strong assumption, induced by the uniformity property: for any low-level exogenous sample, one can define an environment assigning it probability one, and deterministically map it to the high level (see Appendix, Theorem 3.6 in (Beckers et al., 2020)). Their definition further aggregates over these $\tau_{\mathcal{U}}$ maps via a minimum operator. In contrast, our approach does not assume the existence of a deterministic map across environments. Instead, we treat each environment independently, adopting a data-driven formulation avoiding potentially restrictive assumptions or optimality conditions imposed by a global

minimum. That said, since we work with probability measures rather than individual samples, we can always estimate the pushforward of a function, even when its deterministic form is not recoverable. At best, we can approximate a stochastic map derived from the pushforward. In other words, we can always learn a measure-preserving map $\tau_{\mathcal{U}\#} : \rho^\ell \to \rho^h$, though not necessarily the underlying function $\tau_{\mathcal{U}}$. Of course, computing the pushforward measure comes at a cost: it may be complex, computationally expensive, or difficult to work with. This cost could potentially be interpreted as a form of information loss, an idea further explored in (Fullwood & Parzygnat, 2021).

Regarding the task of CAL, the right panel of Fig. 1 aligns learning methods with their underlying environmental assumptions. Prior CAL approaches operate in the $\epsilon = 0$ regime, implicitly assuming a fixed environment, performing learning under varying settings and assumptions. In contrast, our method introduces a robustness parameter $0 \leqslant \epsilon < \infty$ and explicitly models environmental uncertainty using a Wasserstein ball $\mathbb{B}_\epsilon$ centered at the empirical joint environment, enabling generalization to distributional shifts and misspecification within a range controlled by $\epsilon$.

### A.2. The Wasserstein Distance

The Wasserstein distance strongly relates to the theory of optimal transport (Villani et al., 2009), (Peyré & Cuturi, 2019), which seeks to find the most efficient way of transforming one probability distribution to another. In the classical *Monge formulation* (Monge, 1781), the goal is to find a deterministic map $T : \mathbb{R}^n \to \mathbb{R}^n$ that pushes a source measure $\mu$ onto a target measure $\nu$, while minimizing the expected cost of transport, typically taken as the squared Euclidean distance $\|x - T(x)\|_2^2$. However, the Monge problem is highly non-convex and may not admit solutions in general. To address these difficulties, Kantorovich (1942) proposed a relaxation, allowing for *couplings*, these are joint distributions $\pi$ on $\mathbb{R}^n \times \mathbb{R}^n$ with marginals $\mu$ and $\nu$ instead of deterministic maps.

The $W_2$ Wasserstein distance between two probability measures $\mu$ and $\nu$ on $\mathbb{R}^n$ is then defined as the minimal expected squared cost over all such couplings:

$$W_2(\mu, \nu) := \inf_{\pi \in \Pi(\mu, \nu)} \left( \mathbb{E}_{(X,Y) \sim \pi} \|X - Y\|_2^2 \right)^{1/2}, \tag{15}$$

where $\Pi(\mu, \nu)$ denotes the set of all couplings with marginals $\mu$ and $\nu$. Intuitively, $W_2(\mu, \nu)$ measures the least amount of effort required to convert $\mu$ into $\nu$ under a given transport cost.

**Wasserstein Distance between Gaussians.** When $\mu$ and $\nu$ are multivariate Gaussians $\mathcal{N}(m_1, \Sigma_1)$ and $\mathcal{N}(m_2, \Sigma_2)$, the 2-Wasserstein distance admits a closed-form expression (Givens & Shortt, 1984):

$$W_2\left( \mathcal{N}(m_1, \Sigma_1), \mathcal{N}(m_2, \Sigma_2) \right) = \sqrt{\|m_1 - m_2\|_2^2 + \mathrm{Tr}\left( \Sigma_1 + \Sigma_2 - 2\left( \Sigma_1^{1/2} \Sigma_2 \Sigma_1^{1/2} \right)^{1/2} \right)} \tag{16}$$

The squared $W_2$ is also known as the *Gelbrich distance* (Gelbrich, 1990), and it captures both the displacement between the means and the discrepancy between the covariances. In particular, when the covariance matrices commute, the trace term simplifies to the squared Frobenius norm between their matrix square roots.

**Optimal Transport Map between Gaussians.** When $\mu = \mathcal{N}(m_1, \Sigma_1)$ and $\nu = \mathcal{N}(m_2, \Sigma_2)$ are two Gaussian measures on $\mathbb{R}^n$, the optimal transport map $T : \mathbb{R}^n \to \mathbb{R}^n$ pushing $\mu$ to $\nu$ under the $W_2$ distance is affine and given by (Knott & Smith, 1984):

$$T(x) = m_2 + A(x - m_1), \tag{17}$$

where the matrix $A$ is symmetric positive definite and is equal to:

$$A = \Sigma_2^{1/2} \left( \Sigma_2^{1/2} \Sigma_1 \Sigma_2^{1/2} \right)^{-1/2} \Sigma_2^{1/2}. \tag{18}$$

This map is the gradient of a convex function, and is therefore the unique optimal transport map up to sets of $\mu$-measure zero[2], by Brenier's theorem (Brenier, 1991). Moreover, it satisfies the relation

$$T_\sharp \mu = \nu, \tag{19}$$

meaning that the pushforward of $\mu$ under $T$ is exactly $\nu$.

---

[2]Formally, uniqueness holds $\mu$-almost everywhere: if two maps both solve the Monge problem, then they agree except possibly on a set of $\mu$-measure zero.

**Wasserstein Barycenters.** The Wasserstein barycenter (Agueh & Carlier, 2011) provides a principled way to aggregate a collection of probability distributions into a single representative distribution by minimizing the average squared Wasserstein distance to the given distributions. Given a set of probability measures $\{\nu_j\}_{j=1}^n$ and associated weights $\{\lambda_j\}_{j=1}^n$ with $\lambda_j > 0$ and $\sum_{j=1}^n \lambda_j = 1$, the Wasserstein barycenter $\nu^\star$ is defined as the solution of the following problem:

$$\nu^\star = \arg\min_\nu \sum_{j=1}^n \lambda_j W_2^2(\nu, \nu_j), \tag{20}$$

where $W_2(\cdot, \cdot)$ denotes the 2-Wasserstein distance between probability measures. In the special case where all $\nu_j$ are multivariate Gaussians $\mathcal{N}(\mu_j, \Sigma_j)$, it can be shown that the barycenter $\nu^\star$ is also a Gaussian $\mathcal{N}(\mu^\star, \Sigma^\star)$, where the mean $\mu^\star$ is the weighted average:

$$\mu^\star = \sum_{j=1}^n \lambda_j \mu_j \tag{21}$$

and the covariance $\Sigma^\star$ solves the fixed-point equation (Álvarez Esteban et al., 2016):

$$\Sigma^\star = \sum_{j=1}^n \lambda_j \left( \Sigma^{\star 1/2} \Sigma_j \Sigma^{\star 1/2} \right)^{1/2}. \tag{22}$$

Here, the square roots are matrix square roots, and the equation is understood in the positive semidefinite sense.

### A.3. Distributionally Robust Optimization (DRO)

*Distributionally Robust Optimization (DRO)* is a mathematical framework for decision-making under uncertainty, designed to hedge against model misspecification and distributional shifts (Kuhn et al., 2019). Instead of optimizing performance with respect to a single estimated distribution, DRO considers a set of plausible distributions, called the *ambiguity set*, and seeks solutions that minimize the worst-case expected loss across this set.

**Formulation:** Let $x \in \mathbb{R}^n$ denote a decision variable, $\xi \in \Xi = \mathbb{R}^m$ a random vector representing uncertain data, and $f(x, \xi) : \mathbb{R}^n \times \Xi \to \mathbb{R}$ a loss function. The standard *Wasserstein DRO* objective is to solve:

$$\inf_{x \in \mathcal{X}} \sup_{\mathbb{Q} \in \mathbb{B}_{\varepsilon,p}(\widehat{\mathbb{P}}_N)} \mathbb{E}_{\xi \sim \mathbb{Q}}[f(x, \xi)], \tag{23}$$

where $\widehat{\mathbb{P}}_N$ is a nominal distribution estimated from data, and $\mathbb{B}_{\varepsilon,p}(\widehat{\mathbb{P}}_N)$ denotes a Wasserstein ball of radius $\varepsilon$ centered at $\widehat{\mathbb{P}}_N$:

$$\mathbb{B}_{\varepsilon,p}(\widehat{\mathbb{P}}_N) = \left\{ \mathbb{Q} \in \mathcal{P}(\Xi) : W_p(\mathbb{Q}, \widehat{\mathbb{P}}_N) \leq \varepsilon \right\}. \tag{24}$$

The radius $\varepsilon$ controls the size of the ambiguity set and reflects the desired level of robustness: a larger $\varepsilon$ allows for greater deviations from the nominal distribution (increasing robustness), while a smaller $\varepsilon$ yields tighter but potentially less robust decisions. Below, we present two formulations of DRO that are relevant to our work: **(a)** *Elliptical DRO*, which leverages moment-based structural assumptions; and **(b)** *Empirical DRO*, which builds ambiguity sets directly from the empirical data distribution.

**Elliptical DRO.** Elliptical DRO leverages structural assumptions by modeling the data distribution as belonging to the elliptical family (e.g., Gaussian, Student-t), which is characterized by its mean and covariance. Elliptical distributions are fully characterized by their first two moments, which makes them particularly well-suited for moment-based uncertainty modeling. Specifically, a distribution $\mathbb{Q} = \mathcal{E}_g(\mu, \Sigma)$ is called *elliptical* if it admits a density of the form $f(\xi) = C \det(\Sigma)^{-1} g\left( (\xi - \mu)^\top \Sigma^{-1} (\xi - \mu) \right)$, where $g$ is a nonnegative generator function and $C$ is a normalizing constant.

As we saw in the previous section, when $p = 2$, the squared Wasserstein distance between two elliptical distributions admits a closed-form lower bound known as the *Gelbrich distance*:

$$d_G^2((\mu, \Sigma), (\mu', \Sigma')) = \|\mu - \mu'\|^2 + \mathrm{Tr}(\Sigma) + \mathrm{Tr}(\Sigma') - 2\,\mathrm{Tr}\left( (\Sigma^{1/2} \Sigma' \Sigma^{1/2})^{1/2} \right), \tag{25}$$

where $(\mu, \Sigma)$ and $(\mu', \Sigma')$ are the mean-covariance pairs of two distributions.

Assuming elliptical structure, the Wasserstein ambiguity set can be projected onto the space of first and second moments, yielding an *elliptical uncertainty set* (Nguyen et al., 2021):

$$\mathcal{U}_\varepsilon(\widehat{\mu}, \widehat{\Sigma}) = \left\{ (\mu, \Sigma) \in \mathbb{R}^m \times \mathbb{S}_+^m : d_G^2\left( (\widehat{\mu}, \widehat{\Sigma}), (\mu, \Sigma) \right) \leqslant \varepsilon^2 \right\}. \tag{26}$$

This construction offers two key advantages: **(i)** interpretability, as uncertainty is captured through shifts in mean and covariance; and **(ii)** computational tractability, since $\mathcal{U}_\varepsilon$ is a convex and compact set. Furthermore, when the nominal distribution $\widehat{\mathbb{P}}_N$ itself is elliptical, the Wasserstein ambiguity set and the elliptical uncertainty set coincide exactly, i.e., $\mathbb{B}_{\varepsilon,p}(\widehat{\mathbb{P}}_N) = \mathcal{U}_\varepsilon(\widehat{\mu}, \widehat{\Sigma})$.

**Empirical DRO.** In contrast, empirical DRO makes no structural assumptions about the underlying data distribution. Instead, the nominal distribution is the empirical measure:

$$\widehat{\mathbb{P}}_N = \frac{1}{N} \sum_{i=1}^N \delta_{\widehat{\xi}_i}, \tag{27}$$

where $\delta_{\widehat{\xi}_i}$ denotes the Dirac measure at the $i$-th training sample $\widehat{\xi}_i$. A convenient reparameterization restricts attention to ambiguity sets of perturbed empirical distributions of the form:

$$\mathbb{Q}(\Theta) = \frac{1}{N} \sum_{i=1}^N \delta_{\widehat{\xi}_i + \theta_i}, \tag{28}$$

where $\theta_i \in \mathbb{R}^m$ is a displacement vector applied to the $i$-th sample, and $\Theta = (\theta_1, \ldots, \theta_N) \in \mathbb{R}^{m \times N}$ is the perturbation matrix. The Wasserstein constraint $W_p(\mathbb{Q}(\Theta), \widehat{\mathbb{P}}_N) \leqslant \varepsilon$ is then equivalent to:

$$\frac{1}{N} \sum_{i=1}^N \|\theta_i\|^p \leqslant \varepsilon^p. \tag{29}$$

For $p = 2$, this simplifies to $\|\Theta\|_F \leqslant \varepsilon\sqrt{N}$, where $\|\cdot\|_F$ denotes the Frobenius norm. This formulation offers a clear geometric interpretation: empirical DRO controls the total amount of perturbation allowed on the data samples, measured in aggregate via the Frobenius norm. As a result, optimization under empirical DRO can be interpreted as finding decisions that are robust against small, localized deviations from the observed data points.

## A.4. Theorems and Proofs

**Proposition A.1** (Consistency of Metric-based Abstraction Errors). *Let an abstraction context $(\mathcal{A}, \mathcal{I})$ and a distribution $q$ over $\mathcal{I}^\ell$ be given. Suppose $D_{\mathcal{X}^h}$ is a statistical divergence or distance between probability measures satisfying $D_{\mathcal{X}^h}(p, q) = 0 \iff p = q$. If $\mathcal{M}^h$ is a $(\rho, \iota)$-0-approximate abstraction of $\mathcal{M}^\ell$, then for all $\rho \in \mathcal{A}$ and $q$-almost surely, $\mathcal{M}^h$ and $\mathcal{M}^\ell$ are $(\rho, \iota)$-interventionally consistent; i.e. Eq. (4) holds.*

*Proof.* We first consider the case where both aggregation functions $\mathfrak{g}$ and $\mathfrak{h}$ are realized as expectations. Specifically, this transforms the environment-intervention error as follows:

$$e_\tau^{\rho,\iota}(\mathcal{M}^\ell, \mathcal{M}^h) = \mathbb{E}_{\rho \in \mathcal{A}} \mathbb{E}_{\iota \sim q} \left[ D_{\mathcal{X}^h} \left( \tau_\#(\mathbf{g}_{\iota\#}^\ell(\rho^\ell)), \mathbf{g}_{\omega(\iota)\#}^h(\rho^h) \right) \right]. \tag{30}$$

By non-negativity of the divergence $D_{\mathcal{X}^h}$, we have that for every fixed $\rho \in \mathcal{A}$, the inner expectation is non-negative. Since the total abstraction error is assumed to be zero, it follows that for each $\rho \in \mathcal{A}$,

$$\mathbb{E}_{\iota \sim q} \left[ D_{\mathcal{X}^h} \left( \tau_\#(\mathbf{g}_{\iota\#}^\ell(\rho^\ell)), \mathbf{g}_{\omega(\iota)\#}^h(\rho^h) \right) \right] = 0. \tag{31}$$

Taking the expectation of a non-negative random variable implies that

$$D_{\mathcal{X}^h} \left( \tau_\#(\mathbf{g}_{\iota\#}^\ell(\rho^\ell)), \mathbf{g}_{\omega(\iota)\#}^h(\rho^h) \right) = 0, \quad q\text{-almost surely} \tag{32}$$

Since $D_{\mathcal{X}^h}(p, q) = 0 \iff p = q$, we conclude that:

$$\tau_\#(\mathbf{g}^\ell_{\iota\#}(\rho^\ell)) = \mathbf{g}^h_{\omega(\iota)\#}(\rho^h) \quad q\text{-almost surely for each fixed } \boldsymbol{\rho} \in \mathcal{A}. \tag{33}$$

If either one or both of the aggregation functions $\mathfrak{g}$ and $\mathfrak{h}$ are realized as an essential supremum, the argument becomes even simpler: since the supremum of a non-negative function is zero if and only if the function itself is zero everywhere, we again conclude that

$$\tau_\# \left( \mathbf{g}^\ell_{\iota\#}(\rho^\ell) \right) = \mathbf{g}^h_{\omega(\iota)\#}(\rho^h) \tag{34}$$

for all $\boldsymbol{\rho} \in \mathcal{A}$ $q$-almost surely. $\qquad\square$

**Remark 4.** *The proof relies solely on the non-negativity of $D_{\mathcal{X}^h}$ and the fact that $D_{\mathcal{X}^h}(P, Q) = 0$ if and only if $P = Q$. Consequently, the proposition applies to a broad class of divergences and distances, including (but not limited to) the Kullback–Leibler (KL) divergence, the Wasserstein distance $W_p$ for any $p \geqslant 1$, the Total Variation distance, the Hellinger distance, and others.*

**Concentration Results for Joint Environments in Causal Abstraction**    In our causal abstraction setting, each SCM is associated with its own environment: $\rho^\ell$ for the low-level model and $\rho^h$ for the high-level model. We assume these environments are independent and define the joint environment as the product measure $\boldsymbol{\rho} = \rho^\ell \otimes \rho^h$. In practice, however, only empirical samples from each environment are available, leading to an empirical joint distribution $\widehat{\boldsymbol{\rho}} = \widehat{\rho^\ell} \otimes \widehat{\rho^h}$. It is therefore essential to quantify how well the empirical product distribution approximates the true joint environment. To this end, we establish a concentration result in Wasserstein-2 distance, showing that, with high probability, the true joint environment $\boldsymbol{\rho}$ lies within a 2-Wasserstein product ball centered at $\widehat{\boldsymbol{\rho}}$ for an appropriately chosen radius. This result provides the theoretical foundation for defining a distributionally robust causal abstraction objective over joint uncertainty sets. In particular, the following theorems provide principled guidelines for selecting the radius of the joint ambiguity set, ensuring that it contains the true data-generating process. The results build upon: **(a)** the tensorization property of the Wasserstein distance (Villani et al., 2009); **(b)** concentration inequalities for elliptical and empirical distributions (Kuhn et al., 2019, Theorems 18 and 21) and the independence assumption between $\rho^\ell$ and $\rho^h$. We first introduce the notion of a light-tailed distribution, which characterizes the tail behavior necessary for applying the concentration results:

**Definition A.2** (Light-tailed distribution). A probability distribution $\mathbb{P}$ over $\mathbb{R}^d$ is said to be $\alpha$-*light-tailed* if there exist constants $\alpha > 0$ and $A > 0$ such that $\mathbb{E}^{\mathbb{P}}\left[\exp\left(\|\xi\|_2^\alpha\right)\right] \leqslant A$, where $\xi \sim \mathbb{P}$.

Unlike in (Kuhn et al., 2019), where concentration is studied for a single distribution, our setting involves a product of two independent distributions corresponding to the low- and high-level SCMs. Consequently, our concentration theorems specifically account for this product structure.

**Assumption 1.** Both $\rho^\ell$ and $\rho^h$ are light-tailed distributions.

**Theorem 1** (Gaussian $\boldsymbol{\rho}$-Concentration) *Let $\rho^\ell \sim \mathcal{N}(\mu_\ell, \Sigma_\ell)$ and $\rho^h \sim \mathcal{N}(\mu_h, \Sigma_h)$ under Ass.1, let $\widehat{\rho^\ell}$ and $\widehat{\rho^h}$ be the empirical distributions from $N_\ell$ and $N_h$ i.i.d. samples. Also, let $\boldsymbol{\rho} := \rho^\ell \otimes \rho^h$ and $\widehat{\boldsymbol{\rho}} := \widehat{\rho^\ell} \otimes \widehat{\rho^h}$. Then, $\forall\, d \in \{\ell, h\}$, there exist constants $c_d > 0$, depending only on the individual $d$-environment and confidence levels $\eta_d$, such that $\forall \delta \in (0, 1]$, with $\delta = 1 - (1 - \eta_\ell)(1 - \eta_h)$:*

$$\mathbb{P}\left[\mathcal{W}_2(\boldsymbol{\rho}, \widehat{\boldsymbol{\rho}}) \leqslant \epsilon\right] \geqslant 1 - \delta, \ \ \forall\, \epsilon \geqslant \sqrt{\left(\frac{\log(c_\ell/\eta_\ell)}{\sqrt{N_\ell}}\right)^2 + \left(\frac{\log(c_h/\eta_h)}{\sqrt{N_h}}\right)^2}$$

*Proof.* Our goal is to identify a radius $\epsilon > 0$ such that:

$$\mathbb{P}\left[\mathcal{W}_2(\boldsymbol{\rho}, \widehat{\boldsymbol{\rho}}) \leqslant \epsilon\right] \geqslant 1 - \delta. \tag{35}$$

Let the global event $E_p := \{\mathcal{W}_2(\boldsymbol{\rho}, \widehat{\boldsymbol{\rho}}) \leqslant \epsilon\}$. From (Villani et al., 2009), we know that the squared 2-Wasserstein distance tensorizes for product measures. That is, for $\boldsymbol{\rho} = \rho^\ell \otimes \rho^h$ and $\widehat{\boldsymbol{\rho}} = \widehat{\rho^\ell} \otimes \widehat{\rho^h}$, we have:

$$\mathcal{W}_2^2(\boldsymbol{\rho}, \widehat{\boldsymbol{\rho}}) = \mathcal{W}_2^2(\rho^\ell, \widehat{\rho^\ell}) + \mathcal{W}_2^2(\rho^h, \widehat{\rho^h}). \tag{36}$$

Therefore, the event $E_p$ is equivalent to the event:

$$\left\{ \mathcal{W}_2^2(\rho^\ell, \widehat{\rho^\ell}) + \mathcal{W}_2^2(\rho^h, \widehat{\rho^h}) \leqslant \epsilon^2 \right\} \tag{37}$$

Let us now define the per-component local events:

$$E_\ell := \{\mathcal{W}_2(\rho^\ell, \widehat{\rho^\ell}) \leqslant \varepsilon_\ell\} \quad \text{and} \quad E_h := \{\mathcal{W}_2(\rho^h, \widehat{\rho^h}) \leqslant \varepsilon_h\}. \tag{38}$$

If the intersection event $E_\ell \cap E_h$ occurs, then both $\mathcal{W}_2^2(\rho^\ell, \widehat{\rho^\ell}) \leqslant \varepsilon_\ell^2$ and $\mathcal{W}_2^2(\rho^h, \widehat{\rho^h}) \leqslant \varepsilon_h^2$ hold.

Summing these inequalities yields $\mathcal{W}_2^2(\rho^\ell, \widehat{\rho^\ell}) + \mathcal{W}_2^2(\rho^h, \widehat{\rho^h}) \leqslant \varepsilon_\ell^2 + \varepsilon_h^2$. By the tensorization property, this is exactly the condition defining the event $E_p$ and it suffices to select: $\epsilon^2 = \varepsilon_\ell^2 + \varepsilon_h^2$. Thus, the intersection event $E_\ell \cap E_h$ is a subset of the target event $E_p$:

$$E_\ell \cap E_h \subseteq E_p. \tag{39}$$

Consequently, to establish a lower bound on $\mathbb{P}(E_p)$, it suffices to lower bound the probability of the intersection:

$$\mathbb{P}(E_p) \geqslant \mathbb{P}(E_\ell \cap E_h) \tag{40}$$

However, by construction, the empirical measures $\widehat{\rho^\ell}$ and $\widehat{\rho^h}$ are generated independently from $\rho^\ell$ and $\rho^h$. Therefore, the events $E_\ell$ and $E_h$ are statistically independent. This allows us to factor the probability of the intersection according to the product rule for independent events:

$$\mathbb{P}(E_\ell \cap E_h) = \mathbb{P}(E_\ell)\mathbb{P}(E_h). \tag{41}$$

Next, from Theorem 21 of Kuhn et al. (2019), we have that for any confidence level $\eta_i \in (0, 1]$:

$$\mathbb{P}(E_i) = \mathbb{P}(\mathcal{W}_2(\rho_i, \widehat{\rho}_i) \leqslant \varepsilon_i) \geqslant 1 - \eta_i, \tag{42}$$

whenever $\varepsilon_i \geqslant \tilde{\varepsilon}_i = \frac{\log(c_i/\eta_i)}{\sqrt{N_i}}$, for suitable constants $c_i > 0$ depending on the dimension and tail properties of each distribution $\rho_i$.

Consequently, by substituting the Eq.s 41 and 42 into the inequality of Eq. 40, we obtain:

$$\mathbb{P}(E_p) \geqslant \mathbb{P}(E_\ell \cap E_h) = \mathbb{P}(E_\ell)\mathbb{P}(E_h) \geqslant (1 - \eta_\ell)(1 - \eta_h). \tag{43}$$

Recalling that $E_p = \{\mathcal{W}_2(\boldsymbol{\rho}, \widehat{\boldsymbol{\rho}}) \leqslant \sqrt{\varepsilon_\ell^2 + \varepsilon_h^2}\}$ and by expressing the desired global confidence $1 - \delta = (1 - \eta_\ell)(1 - \eta_h)$[3], we have shown:

$$\mathbb{P}\left[\mathcal{W}_2(\boldsymbol{\rho}, \widehat{\boldsymbol{\rho}}) \leqslant \epsilon\right] \geqslant (1 - \eta_\ell)(1 - \eta_h) = 1 - \delta. \tag{44}$$

whenever $\epsilon \geqslant \sqrt{\tilde{\varepsilon}_\ell^2 + \tilde{\varepsilon}_h^2}$, completing the proof. $\square$

**Theorem 2** (Empirical $\boldsymbol{\rho}$-Concentration) *Let $\widehat{\rho^\ell}$ and $\widehat{\rho^h}$ empirical distributions, under Ass.1, from $N_\ell$ and $N_h$ i.i.d. samples, with $\boldsymbol{\rho} := \rho^\ell \otimes \rho^h$ and $\widehat{\boldsymbol{\rho}} := \widehat{\rho^\ell} \otimes \widehat{\rho^h}$. Then, for $\forall d \in \{\ell, h\}$, there exist constants $c_{d,1}, c_{d,2} > 0$, depending only on the individual $d$-environment and confidence levels $\eta_d$, such that $\forall \delta \in (0, 1]$, with $\delta = 1 - (1 - \eta_\ell)(1 - \eta_h)$, for $N_d(c, \eta) = \log(c_{d,1}/\eta)/c_{d,2}$, if we set:*

$$\tilde{\varepsilon}_d = \left(\frac{\log(c_{d,1}/\eta)}{c_{d,2}N_d}\right)^{\min\{1/d, 1/2\}} \quad \text{if } N_d \geqslant N_d(c, \eta), \quad \text{or} \quad \left(\frac{\log(c_{d,1}/\eta)}{c_{d,2}N_d}\right)^{1/\alpha_d} \quad \text{otherwise,}$$

$$\implies \quad \mathbb{P}\left[\mathcal{W}_2(\boldsymbol{\rho}, \widehat{\boldsymbol{\rho}}) \leqslant \epsilon\right] \geqslant 1 - \delta, \quad \forall \epsilon \geqslant \sqrt{\tilde{\varepsilon}_\ell^2 + \tilde{\varepsilon}_h^2}$$

---

[3]A simple choice could be the uniform allocation, given by $\eta_\ell = \eta_d = 1 - (1 - \delta)^{1/2}$.

*Proof.* The core structure of the proof follows exactly that of Theorem 1. Specifically, we define the global event $E_p := \{\mathcal{W}_2(\boldsymbol{\rho}, \widehat{\boldsymbol{\rho}}) \leqslant \epsilon\}$ and local events $E_d := \{\mathcal{W}_2(\rho^d, \widehat{\rho^d}) \leqslant \varepsilon_d\}$, for $d \in \{\ell, h\}$. Using the tensorization property of $\mathcal{W}_2^2$ and the independence of the samples generating $\widehat{\rho^\ell}$ and $\widehat{\rho^h}$, we arrive at the same intermediate conclusion as in the proof of Theorem 1:

$$\mathbb{P}(E_p) \geqslant \mathbb{P}(E_\ell \cap E_h) = \mathbb{P}(E_\ell)\mathbb{P}(E_h), \quad \text{provided that } \epsilon^2 = \varepsilon_\ell^2 + \varepsilon_h^2.$$

The key difference from Theorem 1 lies in the specific concentration inequality used to bound the probabilities of the local events $\mathbb{P}(E_d)$. Here, since $\rho_d^\ell$ are assumed to be general light-tailed distributions, we apply Theorem 18 of Kuhn et al. (2019). This theorem guarantees that for any $\eta_d \in (0,1]$, $\mathbb{P}(E_d) = \mathbb{P}(\mathcal{W}_2(\rho^d, \widehat{\rho^d}) \leqslant \varepsilon_d) \geqslant 1 - \eta_d$, provided that $\varepsilon_d$ is chosen to be at least the threshold value $\tilde{\varepsilon}_d$ where:

$$\tilde{\varepsilon}_d := \begin{cases} \left( \dfrac{\log(c_{d,1}/\eta_d)}{c_{d,2} N_d} \right)^{\min\{1/d, 1/2\}} & \text{if } N_d \geqslant N_d(c, \eta), \\[4mm] \left( \dfrac{\log(c_{d,1}/\eta_d)}{c_{d,2} N_d} \right)^{1/\alpha_d} & \text{otherwise.} \end{cases}$$

To achieve the overall confidence $1 - \delta$, we select local confidence levels $\eta_\ell, \eta_h \in (0,1]$ such that $(1 - \eta_\ell)(1 - \eta_h) = 1 - \delta$, and then by substituting the bounds $\mathbb{P}(E_d) \geqslant 1 - \eta_d$ into the combined probability inequality, we yield:

$$\mathbb{P}(E_p) \geqslant (1 - \eta_\ell)(1 - \eta_h) = 1 - \delta. \tag{45}$$

Thus, the result $\mathbb{P}\left[\mathcal{W}_2(\boldsymbol{\rho}, \widehat{\boldsymbol{\rho}}) \leqslant \epsilon\right] \geqslant 1 - \delta$ is established once again, whenever $\epsilon \geqslant \sqrt{\tilde{\varepsilon}_\ell^2 + \tilde{\varepsilon}_h^2}$. $\qquad\square$

**Remark 5.** *These results justify the use of joint Wasserstein ambiguity sets centered at the empirical product environment. They ensure that, with high probability, the true environment lies within a ball of radius $\epsilon$, allowing us to formulate a robust abstraction objective that is consistent with finite-sample uncertainty. The construction is especially well-suited for our setting, where only a single environment is available from each SCM.*

### A.4.1. PROVABLE ROBUSTNESS

In this section, we prove the core theoretical guarantee of our method: the Generalized Empirical Objective is a tractable, convex min-max problem whose solution $\mathrm{T}^\star$ comes with a provable robustness certificate in terms of the abstraction error. Below, we present the main Theorem A.5, which establishes this result.

For each intervention $\iota$ and abstraction matrix $\mathrm{T} \in \mathbb{R}^{d_h \times d_\ell}$, recall the **nominal misalignment**:

$$Z_{\mathrm{T}}^\iota(\mathbf{0}) := \mathrm{T}\left(D_\ell^{(\iota)} + U^\ell\right)^\top - \left(D_h^{(\omega(\iota))} + U^h\right) \in \mathbb{R}^{d_h \times n_\iota}. \tag{46}$$

Let $\boldsymbol{\Theta} = (\Theta_\ell, \Theta_h)$ denote the adversarial perturbations constrained by Frobenius budgets $r, s \geqslant 0$. The **perturbed misalignment** is defined as $Z_{\mathrm{T}}^\iota(\boldsymbol{\Theta}) := Z_{\mathrm{T}}^\iota(\mathbf{0}) + (\mathrm{T}\Theta_\ell^\top - \Theta_h)$.

We define the worst-case expected loss as:

$$\zeta(\mathrm{T}) := \sup_{\substack{\|\Theta_\ell\|_F \leqslant r \\ \|\Theta_h\|_F \leqslant s}} \mathbb{E}_{\iota \sim q}\left[\left\|Z_{\mathrm{T}}^\iota(\boldsymbol{\Theta})\right\|_F^2\right] = \sup_{\substack{\|\Theta_\ell\|_F \leqslant r \\ \|\Theta_h\|_F \leqslant s}} \mathbb{E}_{\iota \sim q}\left[\left\|Z_{\mathrm{T}}^\iota(\mathbf{0}) + \mathrm{T}\Theta_\ell^\top - \Theta_h\right\|_F^2\right]. \tag{47}$$

Our DiRoCA estimator solves the min-max problem:

$$\min_{\mathrm{T} \in \mathbb{R}^{h \times \ell}} \zeta(\mathrm{T}). \tag{48}$$

We now show that (48) is a tractable convex program and that its optimum provides a formal certificate of robustness. We will need the following two lemmas.

**Lemma A.3** (Inner maximization bound). *Let $\mathrm{T} \in \mathbb{R}^{h \times \ell}$ and $Z \in \mathbb{R}^{h \times N}$ be fixed. Consider the maximization problem over perturbations $\Theta_\ell, \Theta_h$ within budgets $r, s$:*

$$\max_{\|\Theta_\ell\|_F \leqslant r, \, \|\Theta_h\|_F \leqslant s} \left\|Z + \mathrm{T}\Theta_\ell^\top - \Theta_h\right\|_F^2. \tag{49}$$

*The value of this maximization is exactly:*

$$\left( s \ + \ \|Z\|_F \ + \ r\|\mathrm{T}\|_2 \right)^2.\tag{50}$$

*Proof.* The proof proceeds by decomposing the joint maximization into two sequential stages: first analytically solving for the optimal $\Theta_h$ given a fixed $\Theta_\ell$, and subsequently maximizing the resulting expression over $\Theta_\ell$.

**(i) Maximization of $\Theta_h$.** Fix $\mathrm{T}$ and $\Theta_\ell$, and let $A \coloneqq Z + \mathrm{T}\,\Theta_\ell^\top$. The adversary's problem is to choose a $\Theta_h$ that solves:

$$\max_{\|\Theta_h\|_F \leqslant s} \|A - \Theta_h\|_F^2.$$

We expand the squared Frobenius norm:

$$\|A - \Theta_h\|_F^2 = \langle A - \Theta_h, A - \Theta_h \rangle = \|A\|_F^2 - 2\langle A, \Theta_h \rangle + \|\Theta_h\|_F^2.$$

To maximize this expression, the adversary controls both the magnitude and direction of $\Theta_h$, subject to the budget $\|\Theta_h\|_F \leqslant s$.

- **Magnitude:** The term $\|\Theta_h\|_F^2$ is non-negative. It is maximized when the adversary uses their entire budget, setting $\|\Theta_h\|_F = s$.

- **Direction:** The term $-2\langle A, \Theta_h \rangle$ is maximized when the inner product $\langle A, \Theta_h \rangle$ is minimized. By the Cauchy–Schwarz inequality, the minimum value is $-\|A\|_F\|\Theta_h\|_F$, achieved when $\Theta_h$ points in the opposite direction of $A$ ($\Theta_h = -cA$ for $c > 0$).

Both conditions are satisfied simultaneously by choosing:

$$\Theta_h^\star = -s\frac{A}{\|A\|_F}.$$

Substituting this back into the objective yields:

$$\|A - \Theta_h^\star\|_F^2 = \|A\|_F^2 - 2\langle A, -s\frac{A}{\|A\|_F}\rangle + s^2 = \|A\|_F^2 + 2s\|A\|_F + s^2 = \left(\|A\|_F + s\right)^2.$$

Replacing $A$ with $Z + \mathrm{T}\,\Theta_\ell$, we have:

$$\max_{\|\Theta_h\|_F \leqslant s} \|Z + \mathrm{T}\,\Theta_\ell^\top - \Theta_h\|_F^2 = (\|Z + \mathrm{T}\,\Theta_\ell^\top\|_F + s)^2.$$

**(ii) Maximization of $\Theta_\ell$.** The full inner problem reduces to maximizing $\left(\|Z + \mathrm{T}\,\Theta_\ell^\top\|_F + s\right)^2$ over $\|\Theta_\ell\|_F \leqslant r$. Since the function $h(x) = (x + s)^2$ is monotonically increasing for $x \geqslant 0$, this is equivalent to maximizing the term $\|Z + \mathrm{T}\,\Theta_\ell^\top\|_F$.

Using the triangle inequality and the definition of the spectral norm, we obtain the upper bound directly:

$$\max_{\|\Theta_\ell\|_F \leqslant r} \|Z + \mathrm{T}\,\Theta_\ell^\top\|_F \leqslant \|Z\|_F + \max_{\|\Theta_\ell\|_F \leqslant r} \|\mathrm{T}\,\Theta_\ell^\top\|_F$$

$$\leqslant \|Z\|_F + \max_{\|\Theta_\ell\|_F \leqslant r} \|\mathrm{T}\|_2\|\Theta_\ell\|_F$$

$$= \|Z\|_F + r\|\mathrm{T}\|_2.$$

Adding back $s$ and squaring gives the final result: $\left( s + \|Z\|_F + r\|\mathrm{T}\|_2 \right)^2$. $\qquad\square$

**Lemma A.4** (Convexity of the Outer Minimization). *The objective function $\zeta(\mathrm{T})$ defined in (47) is convex in $\mathrm{T}$.*

*Proof.* For any fixed perturbation tuple $c = (\Theta_\ell, \Theta_h)$, define the loss function $g_c(\mathrm{T}) \coloneqq \mathbb{E}_{\iota \sim q}[\| Z_\mathrm{T}^\iota(\mathbf{0}) + \mathrm{T}\,\Theta_\ell^\top - \Theta_h \|_F^2]$. For a fixed $\iota$, the term inside the expectation is the squared Frobenius norm of an affine function of $\mathrm{T}$, which is convex. Since the expectation is a linear operator preserving convexity, $g_c(\mathrm{T})$ is convex in $\mathrm{T}$. The worst-case loss $\zeta(\mathrm{T})$ is the pointwise supremum of the family $\{g_c(\mathrm{T})\}_c$. Since the supremum of convex functions is convex, $\zeta(\mathrm{T})$ is convex. $\qquad\square$

**Theorem A.5** (Provable Robustness)**.** *Let $\zeta(\mathrm{T})$ be the worst-case expected loss defined in* (47)*. Then, for every abstraction matrix $\mathrm{T} \in \mathbb{R}^{d_h \times d_\ell}$:*

$$\zeta(\mathrm{T}) \;\leqslant\; \mathbb{E}_{\iota \sim q}\Big[\big(s + \|Z_\mathrm{T}^\iota(\mathbf{0})\|_F + r\|\mathrm{T}\|_2\big)^2\Big]. \tag{51}$$

*Consequently, minimizing this upper bound (via our proposed algorithm) minimizes a valid upper bound on the worst-case abstraction error.*

*Proof.* The proof relies on the exchange inequality $\sup \mathbb{E}[\cdot] \leqslant \mathbb{E} \sup[\cdot]$. Let $\mathcal{C}$ denote the feasible set of perturbations $\{(\Theta_\ell, \Theta_h) : \|\Theta_\ell\|_F \leqslant r, \|\Theta_h\|_F \leqslant s\}$. We start with the definition of $\zeta(\mathrm{T})$:

$$\zeta(\mathrm{T}) = \sup_{(\Theta_\ell, \Theta_h) \in \mathcal{C}} \mathbb{E}_{\iota \sim q}\left[\big\|Z_\mathrm{T}^\iota(\mathbf{0}) + \mathrm{T}\,\Theta_\ell^\top - \Theta_h\big\|_F^2\right] \tag{52}$$

$$\leqslant \mathbb{E}_{\iota \sim q}\left[\sup_{(\Theta_\ell, \Theta_h) \in \mathcal{C}} \big\|Z_\mathrm{T}^\iota(\mathbf{0}) + \mathrm{T}\,\Theta_\ell^\top - \Theta_h\big\|_F^2\right] \tag{53}$$

The inequality in (53) relaxes the problem by allowing the adversary to choose optimal perturbations specific to each intervention $\iota$ independently, rather than a single fixed perturbation for the entire distribution. We can now apply Lemma A.3 pointwise to the inner maximization term inside the expectation. For a fixed $\iota$, Lemma A.3 guarantees:

$$\sup_{(\Theta_\ell, \Theta_h) \in \mathcal{C}} \big\|Z_\mathrm{T}^\iota(\mathbf{0}) + \mathrm{T}\,\Theta_\ell^\top - \Theta_h\big\|_F^2 \;=\; \big(s + \|Z_\mathrm{T}^\iota(\mathbf{0})\|_F + r\|\mathrm{T}\|_2\big)^2.$$

Substituting this back into (53) yields:

$$\zeta(\mathrm{T}) \leqslant \mathbb{E}_{\iota \sim q}\left[\big(s + \|Z_\mathrm{T}^\iota(\mathbf{0})\|_F + r\|\mathrm{T}\|_2\big)^2\right],$$

which concludes the proof. $\qquad\square$

**Justification of the Generalized Objective.** We adopt the squared Frobenius norm to quantify the discrepancy between abstracted low-level samples and their high-level counterparts. This choice is theoretically grounded in optimal transport: it corresponds to the transport cost under the *empirical identity coupling*. Unlike generic optimal transport solvers, which optimize over *all* admissible couplings (potentially allowing arbitrary re-pairings of units), the identity coupling enforces a strict, sample-wise correspondence. Crucially, in our framework this correspondence is not arbitrary. Through **abduction**, the $i$-th observational datum induces specific exogenous noise values $(u_i^\ell, u_i^h)$ for the low- and high-level SCMs. Consequently, the $i$-th interventional samples at both levels represent counterfactual outcomes of the *same underlying unit*. The identity coupling thus reflects the causally mandated alignment between SCMs.

**Proposition A.6** (Frobenius Norm as the Transport Cost of the Identity Coupling)**.** *Let $\mathbf{A}, \mathbf{B} \in \mathbb{R}^{n \times d_h}$ be matrices of samples where the $i$-th rows correspond to the same observational unit (via abduction). Let $\hat{\mu}_A$ and $\hat{\mu}_B$ be the associated empirical measures. Then the identity coupling $\pi^* = \frac{1}{n}\sum_{i=1}^n \delta_{(a_i, b_i)}$ is a valid element of $\Pi(\hat{\mu}_A, \hat{\mu}_B)$ and its transport cost satisfies:*

$$\int \|x - y\|_2^2 \, d\pi^*(x, y) \;=\; \frac{1}{n}\|\mathbf{A} - \mathbf{B}\|_F^2. \tag{54}$$

*Moreover, since $\mathcal{W}_2^2$ is the infimum over all couplings:*

$$\mathcal{W}_2^2(\hat{\mu}_A, \hat{\mu}_B) \;\leqslant\; \frac{1}{n}\|\mathbf{A} - \mathbf{B}\|_F^2. \tag{55}$$

*Proof.* The equality follows directly from the construction of the identity coupling: the transport cost is the average squared Euclidean distance between paired rows, which is exactly the squared Frobenius norm scaled by $1/n$. For the inequality, recall that the Wasserstein distance is defined as the infimum of the transport cost over the set of all admissible couplings $\Pi(\hat{\mu}_A, \hat{\mu}_B)$. Since $\pi^*$ matches the marginals $\hat{\mu}_A$ and $\hat{\mu}_B$, it is a feasible coupling ($\pi^* \in \Pi$). Therefore, the cost associated with this specific coupling serves as a valid upper bound on the infimum (the Wasserstein distance). $\qquad\square$

**Remark 6** (Frobenius as a Tractable and Causally Correct Surrogate). *Proposition A.6 establishes that the Frobenius objective is not a heuristic; it is the quadratic transport cost under the* causally correct *coupling induced by abduction. In linear Gaussian DiRoCA, this logic leads to the Gelbrich formula (as the optimal coupling is linear). In the general ANM case, solving for the optimal coupling at each step is computationally prohibitive and may ignore unit identity. Minimizing the cost of the identity coupling provides a tractable surrogate that respects causal alignment and upper bounds the Wasserstein distance. Driving the Frobenius norm to zero, therefore guarantees convergence of the interventional distributions in the Wasserstein sense, under the correct unit-level correspondence.*

### A.5. Additive Noise Models and the $(D, U)$ Decomposition

We work with *Markovian* Structural Causal Models (SCMs) $\mathcal{M} = (S, \rho)$, where $S = (\mathcal{X}, \mathcal{U}, \mathcal{F})$ comprises endogenous variables $\mathcal{X} = \{X_i\}_{i=1}^d$, exogenous noises $\mathcal{U} = \{U_i\}_{i=1}^d$, and structural functions $\mathcal{F} = \{f_i\}_{i=1}^d$. Markovianity, defined as the mutual independence of the exogenous variables $U_i$ under the environment $\rho$ implies acyclicity, entailing a DAG $\mathcal{G}_{\mathcal{M}}$. We assume faithfulness and causal sufficiency throughout. The mixing function $\mathbf{g} : \mathrm{dom}[\mathcal{U}] \to \mathrm{dom}[\mathcal{X}]$ is obtained by recursively composing the structural functions along the topological order of $\mathcal{G}_{\mathcal{M}}$ and yields the reduced form of the SCM:

$$\mathcal{X} = \mathbf{g}(\mathcal{U}). \tag{56}$$

Every intervention mutilates the SCM and induces a new reduced form $\mathbf{g}_\iota$. For the CA setting, we denote the corresponding reduced forms by $\mathbf{g}_\iota^\ell$ (low-level) and $\mathbf{g}_{\omega(\iota)}^h$ (high-level). The associated interventional observable distributions are the pushforward measures:

$$\mathbb{P}_{\mathcal{M}_\iota^\ell}(\mathcal{X}^\ell) = (\mathbf{g}_\iota^\ell)_\#(\rho^\ell) \quad \text{and} \quad \mathbb{P}_{\mathcal{M}_{\omega(\iota)}^h}(\mathcal{X}^h) = (\mathbf{g}_{\omega(\iota)}^h)_\#(\rho^h). \tag{57}$$

An *Additive Noise Model (ANM)* specifies that the structural assignment for each node is additive in the noise:

$$X_i = f_i\big(\mathrm{PA}(X_i)\big) + U_i, \quad i \in [d], \tag{58}$$

A *Linear Additive Noise (LAN)* model is a special case of an ANM where the structural functions are linear.

$$\mathcal{X} = \mathbf{B}^\top \mathcal{X} + \mathcal{U}, \tag{59}$$

where $\mathbf{B}$ is (permutable to) a strictly upper triangular matrix. The reduced form is linear:

$$\mathcal{X} = \mathbf{M}\mathcal{U}, \qquad \text{with } \mathbf{M} := (\mathbf{I} - \mathbf{B}^\top)^{-1}. \tag{60}$$

**Note:** If the environments $\rho^\ell, \rho^h$ are Gaussian, the resulting pushforwards $(\mathbf{g}_\iota^\ell)_\#(\rho^\ell)$ and $(\mathbf{g}_{\omega(\iota)}^h)_\#(\rho^h)$ remain Gaussian.

In general ANMs, the functions $f_i$ may be arbitrary (e.g., polynomials, kernels, neural networks, etc). Consequently, the reduced form $\mathbf{g}$ is nonlinear in $\mathcal{U}$, and thus we rely on the $(D, U)$ decomposition described below, which remains valid regardless of structural assumptions.

#### A.5.1. THE $(D, U)$ DECOMPOSITION

We leverage the property that in an SCM with additive noise, the value of any endogenous variable $X_i$ is the sum of a deterministic mechanism $f_i$ and a stochastic residual $U_i$. Given a known causal graph, identifying the SCM reduces to a set of supervised regression problems. This allows us to decompose any observed data, observational or interventional, into a deterministic component matrix $\mathbf{D}$ (representing the causal mechanisms) and a stochastic component matrix $\mathbf{U}$ (representing the specific noise realizations).

**Procedure: Estimation and Abduction.** Assume the DAG is given. The decomposition proceeds in two phases: *estimation* (learning the mechanisms once) and *abduction* (applying them to data).

1. **Fit Mechanisms (If not known):** For each node $X_i$, we estimate the structural function $\hat{f}_i : \mathrm{dom}[\mathrm{PA}(X_i)] \to \mathbb{R}$ by regressing $X_i$ on its parents $\mathrm{PA}(X_i)$ using the observational data.

   - For Linear ANMs, this utilizes standard OLS or Ridge regression.
   - For Nonlinear ANMs, we employ nonparametric regressors (e.g., Random Forests, Neural Networks).

2. **Compute Deterministic Components (D):** For any dataset (observational or interventional), we calculate the deterministic matrix $\mathbf{D}$ by evaluating the *fixed* learned functions on the current parent values: $D_i := \hat{f}_i(\mathrm{PA}(X_i))$.

   - *Intervention Handling:* If variable $X_j$ is under intervention $\mathrm{do}(X_j = a)$, the learned structural mechanism is overridden. The column for $X_j$ in $\mathbf{D}$ is set deterministically to the constant $a$.

3. **Abduce Residuals (U):** We recover the specific exogenous noise realizations for the current samples via subtraction: $U_i := X_i - D_i$.

   - Under a hard intervention on $X_j$, the variable is fully determined by the intervention; we set the corresponding residual to zero to preserve the additive identity.

Stacking these column-wise yields the matrices $\mathbf{D}^{(\iota)}$ and $\mathbf{U}^{(\iota)}$ for any specific interventional environment $\iota$. Note that while the *distribution* of non-intervened noise remains invariant, the specific *samples* $\mathbf{U}^{(\iota)}$ differ across datasets.

**Illustrative Example**    We illustrate this decomposition using a simple nonlinear chain $X \to Y \to Z$ with $N = 5$ samples. Assume we have already performed the estimation step and obtained:

$$\hat{f}_X(\varnothing) = 0, \qquad\qquad \hat{f}_Y(X) = \sin(X), \qquad\qquad \hat{f}_Z(Y) = Y^2.$$

**Case 1: Observational Data ($\iota_0 = \varnothing$).** We observe data matrix $\mathbf{X}^{(0)} \in \mathbb{R}^{5 \times 3}$. We apply the fitted functions to these samples to separate signal from noise:

$$\mathbf{X}^{(0)} = \begin{pmatrix} x_1 & y_1 & z_1 \\ \vdots & \vdots & \vdots \\ x_5 & y_5 & z_5 \end{pmatrix} \implies \mathbf{D}^{(0)} = \begin{pmatrix} 0 & \sin(x_1) & y_1^2 \\ \vdots & \vdots & \vdots \\ 0 & \sin(x_5) & y_5^2 \end{pmatrix}$$

The specific noise realizations for this batch are $\mathbf{U}^{(0)} = \mathbf{X}^{(0)} - \mathbf{D}^{(0)}$.

**Case 2: Interventional Data ($\iota_1 = \mathrm{do}(X = 1.0)$).** We collect a *new* batch of 5 samples $\mathbf{X}^{(1)}$. Even though the underlying noise distribution is the same, these are new realizations. We use the **same** functions $\hat{f}$ to decompose this new data, but override the intervened column. Note that since $X$ is fixed to 1.0, the parent input for $Y$ is now 1.0:

$$\mathbf{X}^{(1)} = \begin{pmatrix} 1.0 & y_1' & z_1' \\ \vdots & \vdots & \vdots \\ 1.0 & y_5' & z_5' \end{pmatrix} \implies \mathbf{D}^{(1)} = \begin{pmatrix} \mathbf{1.0} & \sin(1.0) & (y_1')^2 \\ \vdots & \vdots & \vdots \\ \mathbf{1.0} & \sin(1.0) & (y_5')^2 \end{pmatrix}$$

The residual matrix $\mathbf{U}^{(1)} = \mathbf{X}^{(1)} - \mathbf{D}^{(1)}$ captures the new noise samples for $Y$ and $Z$, while the column for $X$ is strictly zero.

**The DiRoCA Protocol.**    While the standard sampling above involves varying noise across datasets, the DiRoCA optimization (Appendix A.9) utilizes a shared Residuals protocol, as we saw in the main text. During training, we explicitly reuse the observational residuals $\mathbf{U}^{(0)}$ from Case 1 to generate interventional batches as: $\mathbf{X}^{(\iota)} = \mathbf{D}^{(\iota)} + \mathbf{U}^{(0)}$. This protocol is mandated by the empirical DRO formulation defined in Section 4 since we solve for a single worst-case perturbation matrix. As discussed in the main text, this reuse yields a consistent pairing of batch rows across interventions; it is not an alignment assumption but a computational device enabling efficient Frobenius-based discrepancy evaluation and avoiding repeated optimal transport computations necessary for the robust objective since we solve for a single worst-case perturbation of the underlying noise distribution, we must use a single base set of residuals to define the ambiguity set consistently across all environments. Although this technically fixes the noise profile rather than sampling new realizations for every intervention, it effectively forces the model to learn the stable deterministic mechanism $\mathbf{D}^{(\iota)}$ invariant to specific noise samples; empirically, this regularization provides the extra robustness observed during test time, where DiRoCA successfully generalizes to unseen noise realizations. When tested on actual samples where noise naturally varies, the robust DiRoCA method is able to successfully capture the underlying mechanisms, whereas other baselines fail, as demonstrated in the results below.

**Adversarial Perturbation Strategy.**   While linear models admit a simplified surrogate through the reduced form of the SCM, this $(D, U)$ decomposition suggests a universal strategy for nonlinear ANMs. By separating the deterministic mechanisms from the stochastic noise, we can inject adversarial perturbations $\Theta$ directly into the abduced residuals $U$. This allows us to evaluate robustness of the abstraction map T by propagating perturbed noise $U + \Theta$ through the fixed (estimated) mechanisms $\hat{f}$, without requiring re-estimation of structural functions during the optimization loop.

**Structural Misspecification as Residual Perturbation.**   The same decomposition also clarifies why DiRoCA can remain stable when structural assumptions are violated (e.g., imperfect functional estimation, mild mechanism deviations, or errors in the intervention mapping). If the true mechanism $f_i$ differs from the estimated mechanism $\hat{f}_i$, then for non-intervened nodes we can write

$$X_i = f_i(\mathrm{PA}(X_i)) + U_i = \hat{f}_i(\mathrm{PA}(X_i)) + \underbrace{(U_i + \delta_i(\mathrm{PA}(X_i)))}_{\tilde{U}_i}, \qquad \delta_i(\cdot) := f_i(\cdot) - \hat{f}_i(\cdot).$$

Thus, from the perspective of the learned model, structural error $\delta_i(\mathrm{PA}(X_i))$ is observationally absorbed into an *effective residual* $\tilde{U}_i$. Consequently, adversarial perturbations $\Theta$ applied to abduced residuals $U$ can be interpreted as covering both (i) distributional shifts in exogenous noise and (ii) deviations induced by mechanism misspecification or estimation error. This offers a simple explanation for why optimizing against worst-case perturbations in a Wasserstein ambiguity set can improve robustness beyond environmental shift, and it aligns with recent perspectives that model mechanism deviations as structured perturbations of causal systems (Schneider et al., 2025).

### A.6. The Joint Ambiguity Set

Here, we provide a visual and formal illustration of the construction of the *joint ambiguity set* used in our framework, as described in the main text. Our goal is to introduce distributional robustness at the level of environments by following the Distributionally Robust Optimization (DRO) paradigm. Since we assume access to a single observed environment from each SCM (low-level and high-level), we model distributional uncertainty using a 2-Wasserstein product ball centered at the empirical joint environment $\hat{\boldsymbol{\rho}}$ that is formally defined as the joint ambiguity set:

$$\mathbb{B}_{\epsilon,2}(\hat{\boldsymbol{\rho}}) := \mathbb{B}_{\varepsilon_\ell,2}(\widehat{\rho^\ell}) \times \mathbb{B}_{\varepsilon_h,2}(\widehat{\rho^h}), \tag{61}$$

where for each domain $d \in \{|\mathcal{M}^\ell|, |\mathcal{M}^h|\}$, the marginal ambiguity set $\mathbb{B}_{\varepsilon_d,2}(\widehat{\rho^d})$ is a 2-Wasserstein ball defined as

$$\mathbb{B}_{\varepsilon_d,2}(\widehat{\rho^d}) = \left\{ \rho \in \mathcal{P}(\mathcal{U}^d) : W_2(\rho, \widehat{\rho^d}) \leqslant \varepsilon_d \right\}. \tag{62}$$

Each radius $\varepsilon_d$ controls the robustness level for the corresponding SCM, with larger values providing robustness to broader shifts but may lead to more conservative solutions.

**Construction Steps.**   Since we do not observe exogenous environments directly, we reconstruct them via abduction. Specifically, given samples from the endogenous variables, we invert the reduced form of the SCMs to recover approximate exogenous samples:

$$(\mathbf{g}_\#^d)^{-1}(\widehat{\mathcal{X}^d}) \approx \widehat{\mathcal{U}^d}, \quad \text{for } d \in \{\ell, h\}. \tag{63}$$

With known structural functions, this inversion is exact. Otherwise, if the structural functions must be estimated, the quality of the abduction depends on the estimation accuracy of the selected method (e.g. Linear Regression for LAN models, Random Forest/ Neural Networks for general ANMs). After performing abduction, we form the empirical exogenous distributions $\widehat{\mathcal{U}^\ell}$ and $\widehat{\mathcal{U}^h}$ via empirical measures:

$$\widehat{\mathcal{U}^\ell} = \frac{1}{N} \sum_{i=1}^{N} \delta_{\widehat{\mathbf{u}_i^L}}, \quad \widehat{\mathcal{U}^h} = \frac{1}{M} \sum_{j=1}^{M} \delta_{\widehat{\mathbf{u}_j^H}}, \tag{64}$$

where $\widehat{\mathbf{u}_i^L}$ and $\widehat{\mathbf{u}_j^H}$ denote the reconstructed exogenous samples at the low and high levels, respectively. Next, we form the nominal joint distribution $\hat{\boldsymbol{\rho}} := \widehat{\rho^\ell} \otimes \widehat{\rho^h}$ by combining the empirical exogenous distributions. Finally, we build the joint

ambiguity set $\mathbb{B}_{\epsilon,2}(\widehat{\boldsymbol{\rho}})$ by taking the product of two marginal Wasserstein balls. The ambiguity set $\mathbb{B}_{\epsilon,2}(\widehat{\boldsymbol{\rho}})$ allows us to find the worst-case joint environment $\boldsymbol{\rho}^\star = \rho^{\ell\star} \otimes \rho^{h\star}$ that maximizes the abstraction error within the specified robustness radii. This captures the most adversarial distributional shift consistent with the observed data and estimated environments. The radii $\varepsilon_\ell$ and $\varepsilon_h$ quantify the maximum deviation we aim to protect against at each abstraction level. The full construction process, including the abduction, nominal estimation, and radius selection steps, is illustrated in Fig. 5 for the case where low- and high-level environments are sampled independently.

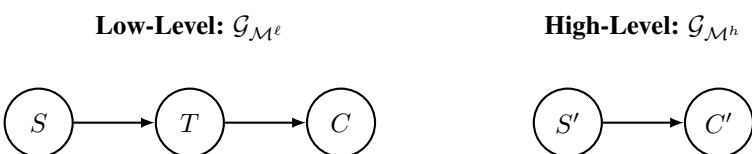

**Abduction**  **Nominal Estimation**  **Radius Selection**

*Figure 5.* Construction of the joint ambiguity set $\mathbb{B}_{\epsilon,2}(\widehat{\boldsymbol{\rho}})$ under the assumption of independently sampled environments. The process involves abduction of exogenous variables, nominal empirical distribution estimation, and selection of robustness radii.

## A.7. Datasets

### A.7.1. SIMPLE LUNG CANCER (SLC) DATASET

The Simple Lung Cancer (SLC) dataset models the causal relationships between three continuous variables: $S$ denotes *Smoking*, $T$ denotes *Tar deposition* in the lungs, and $C$ denotes *Cancer* development. In the low-level representation, smoking causes tar accumulation, which, in turn, causes cancer. The high-level abstraction removes the intermediate variable $T$, resulting in a direct causal link between smoking and cancer. The experiment includes a set of 6 distinct low-level and 3 high-level binary relevant interventions: $\mathcal{I}^\ell = \{\iota_0, \ldots, \iota_5\}$ and $\mathcal{I}^h = \{\eta_0, \eta_1, \eta_2\}$. Table 4 demonstrates the map $\omega : \mathcal{I}^\ell \rightarrow \mathcal{I}^h$. The corresponding low-level and high-level causal graphs are shown in Fig. 6.

**Low-Level:** $\mathcal{G}_{\mathcal{M}^\ell}$  **High-Level:** $\mathcal{G}_{\mathcal{M}^h}$

$$S \rightarrow T \rightarrow C \qquad S' \rightarrow C'$$

*Figure 6.* Low-level ($\mathcal{G}_{\mathcal{M}^\ell}$) and high-level ($\mathcal{G}_{\mathcal{M}^h}$) causal graphs for the Simple Lung Cancer (SLC) dataset. In the high-level abstraction, the intermediate variable $T$ (tar deposition) is omitted, resulting in a direct causal link between smoking and cancer.

*Table 4.* Intervention definitions and the $\omega$ map for the SLC dataset. The map $\omega$ aggregates low-level interventions $\iota$ into high-level interventions $\eta$.

| High-Level ($\eta$) | | Low-Level ($\iota$) mapped to $\eta$ | |
|---|---|---|---|
| **Label** | **Definition** | **Label** | **Definition** |
| $\eta_0$ | $\varnothing$ (Observational) | $\iota_0$ | $\varnothing$ (Observational) |
| $\eta_1$ | $do(S' = 0)$ | $\iota_1$ | $do(S = 0)$ |
| | | $\iota_2$ | $do(S = 0, T = 1)$ |
| $\eta_2$ | $do(S' = 1)$ | $\iota_3$ | $do(S = 1)$ |
| | | $\iota_4$ | $do(S = 1, T = 0)$ |
| | | $\iota_5$ | $do(S = 1, T = 1)$ |

A.7.2. LUCAS DATASETS (LiLUCAS & nLUCAS)

The LUng CAncer dataset (LUCAS)[4], originally designed to simulate realistic causal relationships in lung cancer diagnosis. We consider two versions sharing the same causal structure and abstraction logic but differing in mechanism complexity: the *Linearized LUCAS* (LiLUCAS) dataset, which uses linear mechanisms, and the nLUCAS dataset, which employs non-linear relationships on the low-level model. The low-level model involves several continuous variables: SM denotes *Smoking*, GE denotes *Genetics*, LC denotes *Lung Cancer*, AL denotes *Allergy*, CO denotes *Coughing*, and FA denotes *Fatigue*. In the high-level abstraction, groups of variables are clustered into broader concepts: EN' (Environment), GE' (Genetics), and LC' (Lung Cancer). While sharing the same abstraction logic, the datasets differ in their interventional configurations. LiLUCAS utilizes a comprehensive set of 21 relevant binary low-level interventions (including the observational state) mapped to 11 high-level interventions: $\mathcal{I}^\ell = \{\iota_0, \ldots, \iota_{20}\}$ and $\mathcal{I}^h = \{\eta_0, \ldots, \eta_{10}\}$. In contrast, nLUCAS employs a subset of 11 low-level interventions while keeping the same high-level ones. Table 5 demonstrates the full map $\omega : \mathcal{I}^\ell \to \mathcal{I}^h$ used in the linear setting. The corresponding low-level and high-level causal graphs are shown in Fig. 7.

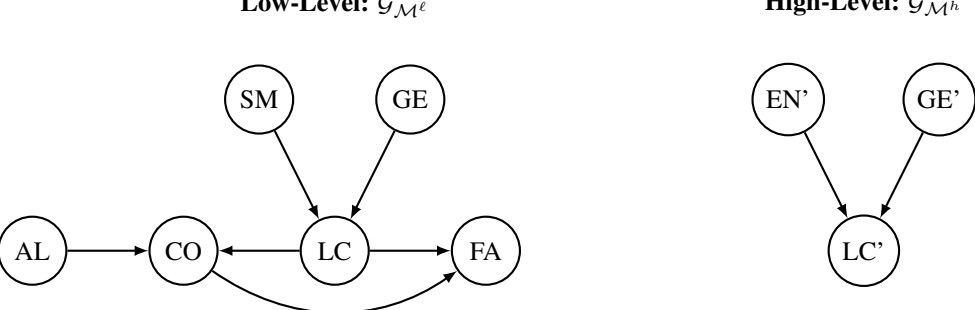

**Low-Level:** $\mathcal{G}_{\mathcal{M}^\ell}$        **High-Level:** $\mathcal{G}_{\mathcal{M}^h}$

*Figure 7.* Low-level ($\mathcal{G}_{\mathcal{M}^\ell}$) and high-level ($\mathcal{G}_{\mathcal{M}^h}$) causal graphs for the LUCAS datasets (LiLUCAS and nLUCAS). The abstraction groups variables related to environment, genetics, and disease outcomes, resulting in a simplified causal structure.

A.7.3. ELECTRIC BATTERY MANUFACTURING (EBM) (ZENNARO ET AL., 2023)

The Electric Battery Manufacturing (EBM) dataset applies our framework to a real-world scenario involving lithium-ion battery production. This dataset contains data collected from two distinct experimental settings representing different levels of granularity. The first setting (WMG) has been modeled through a low-level SCM that captures the effect of a control variable (comma gap) on an output (mass loading) at multiple spatial locations. The second setting (LRCS) is modeled through a high-level SCM that relates the same control variable to a single aggregated output. The low-level variables are $\mathbf{X}_L = [\text{CG}, \text{ML}_1, \text{ML}_2]^\top$, where CG denotes the *Comma Gap* (in $\mu m$) and $\text{ML}_{1,2}$ denote *Mass Loading* measurements (in $mg/cm^2$) at two distinct spatial locations. The high-level variables are $\mathbf{X}_H = [\text{CG}', \text{ML}]^\top$, where mass loading is treated as a single scalar.

**SCM Construction and Abduction**    Unlike the synthetic datasets where structural functions are known, here we must infer them from data. A key methodological divergence from the original work by Zennaro et al. (2023), which used a non-parametric empirical SCM, is our use of a parametric ANM This choice allows us to fully utilize our generalized optimization objective. We fit linear mechanisms with intercepts to both datasets, respecting the causal graph structures (CG → ML):

$$X_i = \sum_{j \in \text{Pa}(i)} \beta_{ji} X_j + c_i + U_i, \tag{65}$$

where $\beta_{ji}$ are the learned coefficients and $c_i$ are the intercepts. We perform exact abduction to recover the noise terms by computing the residuals: $\hat{U}_i = X_i - (c_i + \sum \beta_{ji} X_j)$. To ensure robustness, the noise is centered to be mean-zero per intervention bucket. The deterministic component $D^{(\iota)}$ for an intervention $\iota = do(\text{CG} = c)$ is then defined as:

$$D_i^{(\iota)} = \begin{cases} c & \text{if } i = \text{CG}, \\ c_i + \sum_{j \in \text{Pa}(i)} \beta_{ji} \cdot c & \text{if } i \in \text{Ch}(\text{CG}), \\ c_i & \text{otherwise.} \end{cases} \tag{66}$$

---

[4] http://www.causality.inf.ethz.ch/data/LUCAS.html

*Table 5.* Intervention definitions and the $\omega$ map for the LUCAS datasets. The map $\omega$ aggregates low-level interventions $\iota$ into high-level interventions $\eta$.

| Label | High-Level ($\eta$) Definition | Label | Low-Level ($\iota$) mapped to $\eta$ Definition |
|-------|-------------------------------|-------|------------------------------------------------|
| $\eta_0$ | $\varnothing$ (Observational) | $\iota_0$ | $\varnothing$ (Observational) |
| $\eta_1$ | $do(\text{EN}' = 0)$ | $\iota_1$ | $do(\text{SM} = 0)$ |
| $\eta_2$ | $do(\text{EN}' = 1)$ | $\iota_2$ | $do(\text{SM} = 1)$ |
| $\eta_3$ | $do(\text{GE}' = 0)$ | $\iota_3$ $\iota_{12}$ | $do(\text{LC} = 0)$ $do(\text{GE} = 1, \text{SM} = 1)$ |
| $\eta_4$ | $do(\text{GE}' = 1)$ | $\iota_4$ $\iota_{13}$ | $do(\text{LC} = 1)$ $do(\text{GE} = 0, \text{SM} = 1)$ |
| $\eta_5$ | $do(\text{EN}' = 0, \text{GE}' = 0)$ | $\iota_5$ $\iota_{10}$ $\iota_{16}$ | $do(\text{SM} = 0, \text{LC} = 0)$ $do(\text{GE} = 1)$ $do(\text{AL} = 1)$ |
| $\eta_6$ | $do(\text{EN}' = 1, \text{GE}' = 1)$ | $\iota_6$ $\iota_{11}$ $\iota_{17}$ | $do(\text{SM} = 1, \text{LC} = 1)$ $do(\text{GE} = 0, \text{SM} = 0)$ $do(\text{CO} = 0)$ |
| $\eta_7$ | $do(\text{EN}' = 0, \text{GE}' = 1)$ | $\iota_{15}$ | $do(\text{AL} = 0)$ |
| $\eta_8$ | $do(\text{EN}' = 1, \text{GE}' = 0)$ | $\iota_{14}$ $\iota_{20}$ | $do(\text{GE} = 1, \text{SM} = 0)$ $do(\text{CO} = 0, \text{FA} = 1)$ |
| $\eta_9$ | $do(\text{LC}' = 0)$ | $\iota_7$ $\iota_{19}$ | $do(\text{SM} = 0, \text{LC} = 1)$ $do(\text{CO} = 1, \text{FA} = 0)$ |
| $\eta_{10}$ | $do(\text{LC}' = 1)$ | $\iota_8$ $\iota_{18}$ | $do(\text{SM} = 1, \text{LC} = 0)$ $do(\text{CO} = 1, \text{FA} = 1)$ |

Finally, we use the set of real interventions performed during the data collection. The continuous Comma Gap values are aligned via the map $\omega$, as detailed in Table 6.

### A.7.4. COLORED MNIST (cMNIST) (XIA & BAREINBOIM, 2024)

This experiment tests our framework on a high-dimensional, non-linear task based on the Colored MNIST (cMNIST) dataset. The objective is to learn a robust *linear* abstraction T that aligns a complex pixel-level SCM with a disentangled latent-space SCM. While both levels operate on the same graph structure visualized in Fig. 9 with parents Digit $D$ and Color $C$, which are correlated ($p = 0.85$), they are distinguished by their collider targets. The low-level target $X^\ell$ is instantiated as the high-dimensional image $I_P \in \mathbb{R}^{3072}$, while the high-level target $X^h$ is the compact latent code $z \in \mathbb{R}^{64}$ obtained via the pre-trained encoder $E_\phi$ (Xia & Bareinboim, 2024).

We utilize a set of 10 distinct interventions $\mathcal{I}$ alongside the observational data. The set includes atomic interventions (e.g., $\text{do}(D = 6)$) and joint interventions (e.g., $\text{do}(D = 4, C = 4)$). These interventions are critical because the observational data is heavily confounded (e.g., digit 0 is strongly correlated with red). Interventions break these spurious correlations, creating counterfactual combinations (e.g., digit 0 with blue) that are absent in the training distribution. Since both SCMs share identical parent definitions, the alignment map is the identity: $\omega(\iota) = \iota$.

**SCM Construction and Abduction** The original generative process is non-additive ($z = G(D, C, \epsilon)$), preventing direct abduction in our additive noise framework. We explicitly construct and fit two valid ANMs to model the data:

- *Low-Level Model ($f_L$): U-Net with FiLM.* We model the pixel generation as $I_P = f_L(\text{Shape}, D, C) + U^\ell$. We employ a *U-Net* architecture that takes a grayscale "Shape" image as input to define structural content. To inject causal control, we use *FiLM (Feature-wise Linear Modulation)* layers. These layers map the parents $(D, C)$ to affine parameters

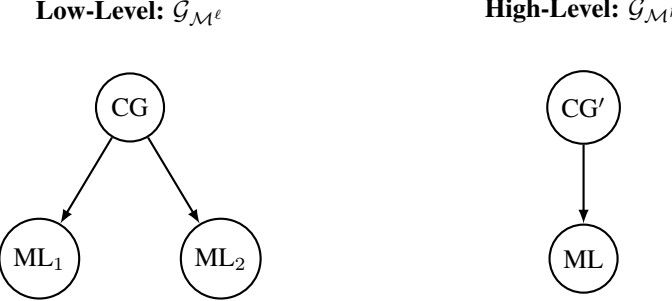

**Figure 8.** Low-level ($\mathcal{G}_{\mathcal{M}^\ell}$) and high-level ($\mathcal{G}_{\mathcal{M}^h}$) causal graphs for the `EBM` dataset. The low-level model captures spatial granularity (ML$_1$, ML$_2$), while the high-level model represents an aggregated view (ML).

*Table 6.* Intervention definitions and the $\omega$ map for the `EBM` dataset. Interventions correspond to setting the Comma Gap (CG) to specific values (in $\mu m$).

| High-Level ($\eta$) | | Low-Level ($\iota$) mapped to $\eta$ | |
| --- | --- | --- | --- |
| **Label** | **Definition** | **Label** | **Definition** |
| $\eta_0$ | $\varnothing$ (Observational) | $\iota_0$ | $\varnothing$ (Observational) |
| $\eta_1$ | $do(\text{CG} = 75)$ | $\iota_1$ | $do(\text{CG}' = 75)$ |
| $\eta_2$ | $do(\text{CG} = 100)$ | $\iota_2$ | $do(\text{CG}' = 110)$ |
| $\eta_3$ | $do(\text{CG} = 200)$ | $\iota_3$ | $do(\text{CG}' = 180)$ |
| | | $\iota_4$ | $do(\text{CG}' = 200)$ |

($\boldsymbol{\gamma}, \boldsymbol{\beta}$), which modulate the internal feature maps of the U-Net via $\mathbf{h}' = \boldsymbol{\gamma} \cdot \mathbf{h} + \boldsymbol{\beta}$. This architecture analytically separates structural content (processed by convolutions) from style (injected by FiLM), allowing for noise abduction.

- *High-Level Model ($f_H$): Vector-Valued Cell-Means.* We model the latent mechanism as $z = f_H(D, C) + U^h$. Given the discrete $10 \times 10$ parent space, we fit a *vector-valued cell-means model*. To ensure robustness against data scarcity in specific $(d, c)$ buckets (caused by confounding), we apply shrinkage regularization. The deterministic vector $m_{d,c}$ is a weighted average of the empirical cell mean and a stable additive baseline ($\hat{\mu}_{\text{base}} = \hat{\mu}_d + \hat{\mu}_c - \hat{\mu}_{\text{global}}$).

**Abduction Procedure.** Following our framework, we (1) fit $f_L$ and $f_H$ on observational data; (2) abduce anchor noise matrices $\mathbf{U}^\ell$ and $\mathbf{U}^h$ as residuals; and (3) synthesize the deterministic components $\mathbf{D}_\ell^{(\iota)}$ and $\mathbf{D}_h^{(\iota)}$ by evaluating the fitted functions under the interventional settings defined in $\mathcal{I}$.

A visual inspection of the dataset (Fig. 10) reveals the core causal challenge: strong spurious correlations in the observational distribution. For instance, specific digits are consistently paired with specific colors (e.g., digit 3 is predominantly green). Interventions are critical for breaking these correlations, generating counterfactual combinations (e.g., digit 3 as red) that expose the true causal mechanism.

**Latent Space Structure.** We analyze the geometry of the high-level latent space $z \in \mathbb{R}^{64}$ by projecting it to two dimensions using PCA (Figure 11). The visualizations show that the latent space $z$ (learned by Xia et al.'s encoder $E_\phi$) is highly structured. The plots show that both the 10 digits and the 10 colors form distinct clusters. This suggests that the latent space is *disentangled*, with different sets of dimensions corresponding to "digit" and "color" information.

**Interventional Geometry.** Finally, we analyze the effect of interventions on the latent distribution $P(z)$, as shown in Figure 12. The distribution of the $l_2$ norm of $z$, $\|z\|_2$, (left plot) shifts significantly under different interventions. The observational distribution (blue) is distinct from the interventional ones (orange, green, purple). This confirms that interventions induce a distributional shift in the high-level space. The right plot quantifies this shift, showing the $l_2$ distance $\|z_{obs} - z_{int}\|_2$ between the latent vector of an observational sample and its interventional counterpart. The distribution is centered far from zero (around 15-20), confirming that interventional samples lie in a different region of the latent space

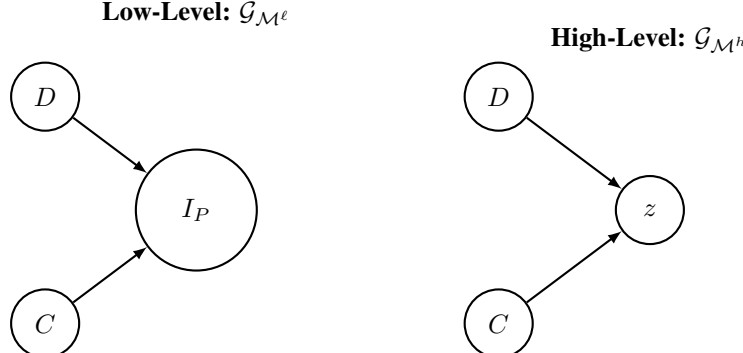

**Figure 9.** Causal graphs for cMNIST. Both levels share parents $D$ (Digit) and $C$ (Color). The low-level generates pixels $I_P$, while the high-level generates latent codes $z$.

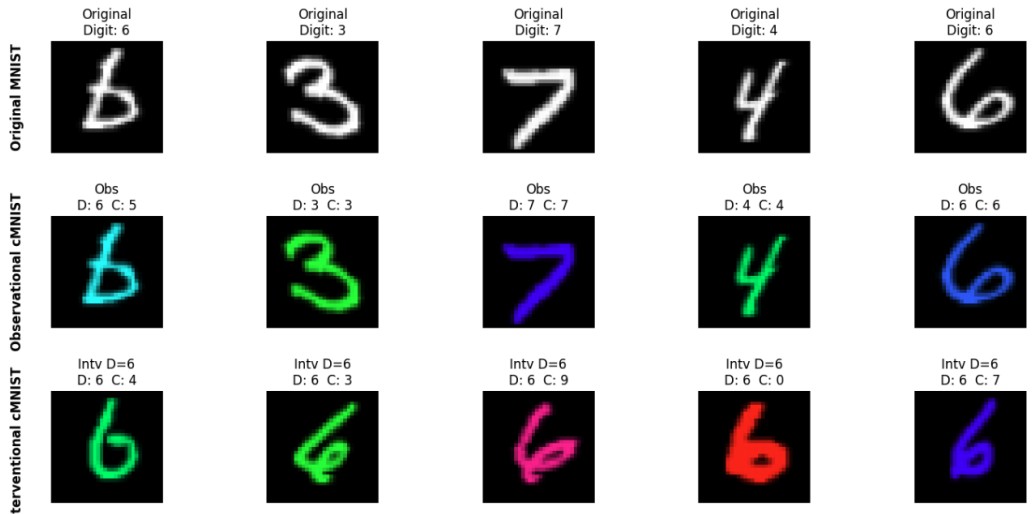

**Figure 10.** cMNIST data samples. **Top:** Original grayscale MNIST. **Middle:** Observational data, showing the spurious correlation (e.g., $D = 3, C = 3$ and $D = 7, C = 7$). **Bottom:** Interventional data for $do(D = 6)$, where the correlation is broken, and the digit 6 is paired with novel colors.

than their observational counterparts.

### A.8. Additional Experimental Results and Settings

We evaluate our framework across a diverse set of experimental settings of different representations of the environment (Gaussian vs. empirical) and datasets. Fig. 13 provides an overview of the datasets, methods, and evaluation metrics used in each setting.

#### A.8.1. MISSPECIFICATIONS AND SETTINGS

**Robustness to Distributional Shifts.** We evaluate robustness using a unified and flexible framework based on a Huber-style contamination model, applied to the held-out test data from each of the $k$ cross-validation folds. The methodology consists of data contamination, error computation, and a multi-level aggregation protocol.

1. Data Contamination. For a given clean interventional test set, $X_{\text{test},\iota} \in \mathbb{R}^{n_{\text{test}} \times d}$, we first generate a noisy outlier copy, $\widetilde{X}_\iota$, by applying an additive stochastic shift. This is achieved by generating a noise matrix $\mathbf{N} \in \mathbb{R}^{n_{\text{test}} \times d}$, where each row is an independent sample from a specified distribution, and setting

$$\widetilde{X}_\iota = X_{\text{test},\iota} + \mathbf{N}.$$

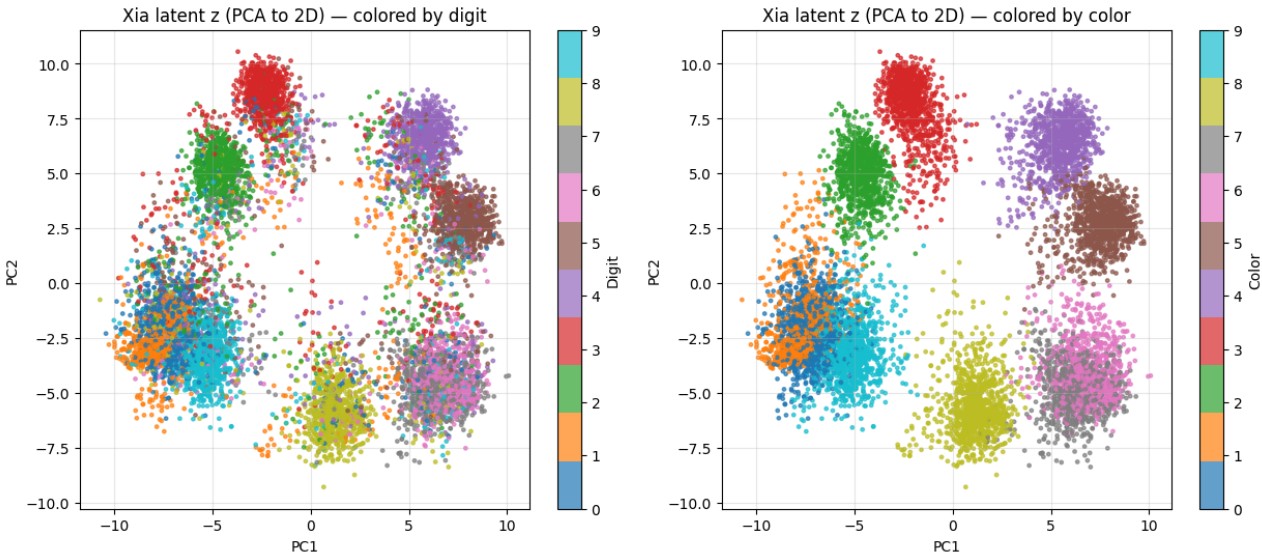

*Figure 11.* PCA projection of the latent space $z \in \mathbb{R}^{64}$ from the observational dataset. **Left:** Colored by digit $D$. **Right:** Colored by color $C$.

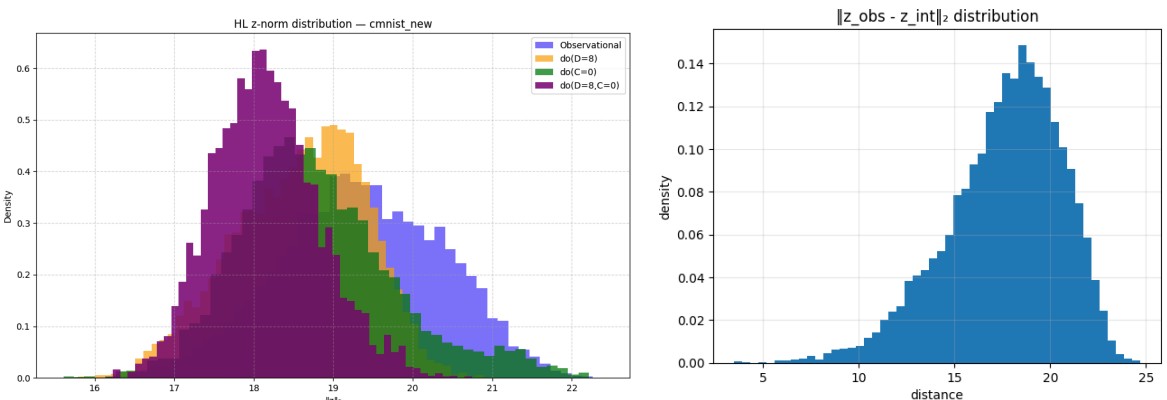

*Figure 12.* **Left:** Distribution of the $l_2$ norm of $z$ under observational and various interventional distributions. **Right:** Distribution of the $l_2$ distance between observational samples $z_{obs}$ and their interventional counterfactuals $z_{int}$.

While the main text focuses on Gaussian noise, $\mathbf{n}_i \sim \mathcal{N}(0, \sigma_{\text{noise}}^2 \mathbf{I})$, our framework also supports other distributions, including heavy-tailed Student-$t$ and skewed Exponential noise. The final contaminated test set, $\bar{X}_\iota$, is then formed by replacing only an $\alpha$-fraction of rows of the clean test set with their noisy counterparts. Specifically, we draw a subset $\mathcal{S} \subseteq [n_{\text{test}}]$ uniformly without replacement, with $|\mathcal{S}| = \lfloor \alpha n_{\text{test}} \rfloor$, and define

$$\bar{X}_{\iota,i} = \left(1 - \mathbb{1}_{i \in \mathcal{S}}\right) X_{\text{test},\iota,i} + \mathbb{1}_{i \in \mathcal{S}} \widetilde{X}_{\iota,i}, \qquad i \in [n_{\text{test}}].$$

Thus, the contamination fraction $\alpha$ controls the prevalence of shifted samples, while the noise scale $\sigma_{\text{noise}}$ controls the magnitude of the stochastic shift.

2. Error Computation. For a single contaminated test set and a given abstraction map $\mathrm{T}$, the abstraction error is computed as the average distance between the transformed low-level and high-level samples, averaged across all interventions $\iota \in \mathcal{I}_\ell$:

$$\mathcal{E}(\tau, \alpha, \sigma_{\text{noise}}) = \mathbb{E}_{\iota \sim q}\left[\mathcal{D}\left(\mathrm{T}\,\bar{X}_\iota^\ell, \bar{X}_{\omega(\iota)}^h\right)\right]. \tag{67}$$

3. Experimental Protocol. To ensure statistically robust results, we perform a multi-level evaluation. The full experiment is a nested procedure that iterates through a grid of contamination parameters ($\alpha$ and $\tilde{\sigma}$). For each point on this grid,

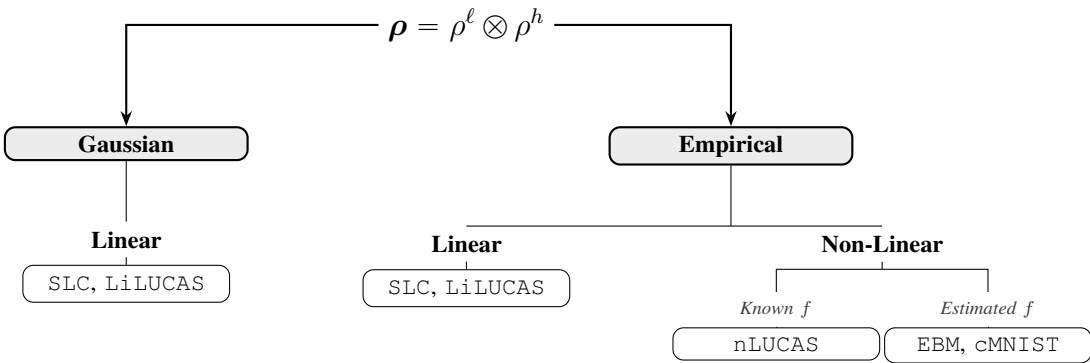

*Figure 13.* **Roadmap of Experimental Datasets.** Evaluation is divided into Gaussian (parametric) and Empirical (data-driven) environments. Under the Empirical setting, we distinguish between **Linear** and **Non-Linear** benchmarks. We further categorize Non-Linear tasks by structural knowledge: **Known** functions (nLUCAS) versus **Estimated** functions (EBM, cMNIST).

our protocol yields a total of $m \times k$ individual error measurements (one for each of the $m$ random samples on each of the $k$ data splits). The final "Robustness Curves" visualize the mean error, averaged over both samples and folds, as a function of the contamination parameters.

**Note.**   For the nLUCAS dataset, we conduct an additional demonstration to validate the impact of the shared residuals optimization strategy discussed in Appendix A.5.1. While our training minimizes discrepancies using a fixed noise realization **U** across interventions, we verify that the learned abstraction is not an artifact of this device. We compare performance under two distinct protocols: **a) Generalization**, which evaluates the abstraction on actual samples with their own (varying) noise samples (Different **U**), and **b) Consistency**, which does the same thing as training, deriving samples from the shared residuals (Shared **U**). In the results below, we explicitly observe the consequence of this distinction. Under the consistency protocol, the non-robust baseline $\text{GRAD}_{(\tau,\omega)}$ outperforms DıRoCA in the clean setting ($\alpha = 0$), as it aggressively fits the specific noise geometry used during optimization. However, under the generalization one, this advantage disappears and DıRoCA outperforms $\text{GRAD}_{(\tau,\omega)}$ even at $\alpha = 0$. This confirms that the baseline overfits the nominal environment's specific noise realization, whereas DıRoCA's adversarial training acts as a regularizer—forcing the model to learn the true, noise-invariant mechanism and preventing the generalization failure described in the main text. Unless otherwise stated, all other results in the main text and appendices report performance under the generalization protocol, as it represents the realistic evaluation setting.

**Robustness to Model Misspecification.**   Beyond robustness to environmental shifts, we evaluate our method's resilience to violations of core modeling assumptions. We challenge two critical assumptions: (a) the linearity of the structural functions ($\mathcal{F}$-misspecification), and (b) the correctness of the intervention mapping ($\omega$-misspecification).

- $\mathcal{F}$-**misspecification.** To test robustness against violations of the linearity assumption, we evaluate our linearly-trained models on test data generated from fundamentally non-linear SCMs. Crucially, these new SCMs share the same causal graph, intervention sets, and underlying environments as those used in training. However, we define a new structural non-linear equations $f_{\text{NL}}$, controlled by a strength parameter $k \in \mathbb{R}$:

$$X_j = k \cdot f_{\text{NL}}(\text{Pa}(X_j)) + U_j. \tag{68}$$

  For each strength level $k$, we simulate a brand new, ground-truth non-linear test set from this SCM and compute the abstraction error. The main text reports the results for $k = 1$ using a sinusoidal function. Here we also provide full robustness curves showing the error as a function of $k \in [0, 100]$ for both sin and tanh non-linearities in Figures 18 and 19 respectively.

- $\omega$-**misspecification.** To test robustness to an incorrect intervention mapping, we contaminate the ground-truth $\omega$ map using a semantic corruption strategy. For a given number of misalignments, we randomly select a subset of low-level interventions and re-assign each one to an incorrect high-level counterpart of similar complexity (i.e., one that intervenes on a similar number of variables), with a fallback to the nearest complexity neighbor if an exact match is not available within a specified delta. An illustration of this semantic $\omega$-misspecification process is provided in Figure 14.

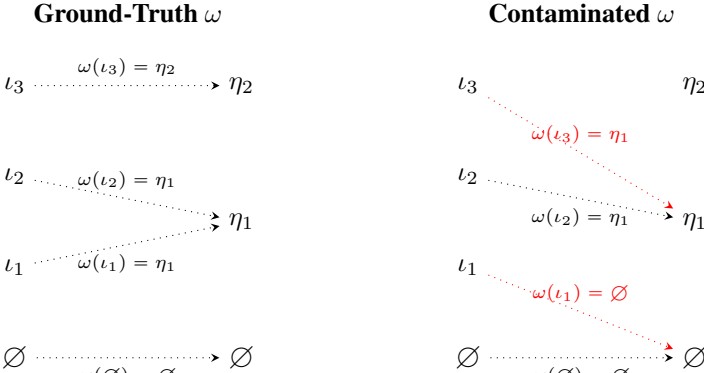

**Figure 14.** Illustration of intervention map ($\omega$) contamination. Left: the ground-truth mapping between low-level interventions ($\iota$) and high-level interventions ($\eta$). Right: a contaminated version where interventions have been re-assigned to incorrect counterparts of similar complexity (e.g., $\omega(\iota_1) = \eta_2$ instead of $\eta_1$) shown in red.

**Robustness to Camera Shifts.** In addition to Huber-style pixel corruptions for the cMNIST dataset, we evaluate the robustness of each learned abstraction under geometric distribution shifts. Unlike pixel noise, which degrades the signal-to-noise ratio, these transformations test the stability of the learned linear operator T under realistic camera shifts. Given a clean image $x \in [-1, 1]^{3 \times 32 \times 32}$ we consider two distinct shifts:

- *Geometric*, which rotates the data manifold in the input space. We generate a rotated version $\tilde{x} = \mathcal{R}_\theta(x)$ at distinct levels of severity $\theta \in \{30°, 90°\}$.

- *Photometric*, which scales pixel intensities to emulate changing environmental lighting. We generate shifted versions $\tilde{x} = \mathcal{B}_\lambda(x)$ by adjusting brightness by a factor $\lambda \in \{0.6, 3.0\}$, representing dim and overexposed conditions, respectively.

A purely linear abstraction trained on canonical orientations may suffer from catastrophic intercept mismatch when the input distribution rotates. To isolate the quality of the learned geometry (the matrix T) from simple mean shifts for a batch of shifted inputs $X_{\text{pix}}$ and corresponding ground-truth latents $Z$, we formulate the prediction as:

$$\hat{Z} = X_{\text{pix}} T^\top + b, \tag{69}$$

where $b = \mathbb{E}[Z - X_{\text{pix}} T^\top]$ is a test-time mean-shift correction vector. This correction effectively re-centers the predicted point cloud to match the first moment of the ground truth, ensuring that the reported *Relative Squared Error* measures strictly the alignment of the learned subspace with the causal variables, independent of the translational offset induced by the rotation.

**Implementation and Optimization Details.** All models were trained on a MacBook Air (13-inch, M1, 2020) equipped with an Apple M1 chip and 16 GB of unified memory, running macOS Monterey version 12.3. Performance is evaluated using a $k$-fold cross-validation scheme (with $k = 5$) on the 10,000 generated samples for each intervention in LiLUCAS and SLC and 5,000 samples for nLUCAS. In DiRoCA each iteration comprises 5 minimization and 2 maximization steps. Finally, we monitored convergence by tracking the absolute change in the objective value across successive epochs, and the optimization is terminated once this change falls below a fixed threshold of $10^{-4}$, ensuring numerical stability while avoiding unnecessary computation. Regarding the operationalization of the theoretical radius (Theorems 1, 2), we treated the concentration constants as hyperparameters fixed for each dataset type. We consistently set the tail exponent $\alpha_d = 2.0$ (assuming sub-Gaussian tails) and confidence level $\eta = 0.05$. The scaling constants were set to $c_{d,1} = 1.0, c_{d,2} = 1.0$ for the linear synthetic datasets (SLC, LiLUCAS), while for the more complex non-linear and high-dimensional tasks (nLUCAS, EBM, cMNIST), we increased the pre-factor $c_{d,1}$ (ranging from 10 to 1000) to account for looser theoretical bounds in higher dimensions, while keeping $c_{d,2} = 1.0$.

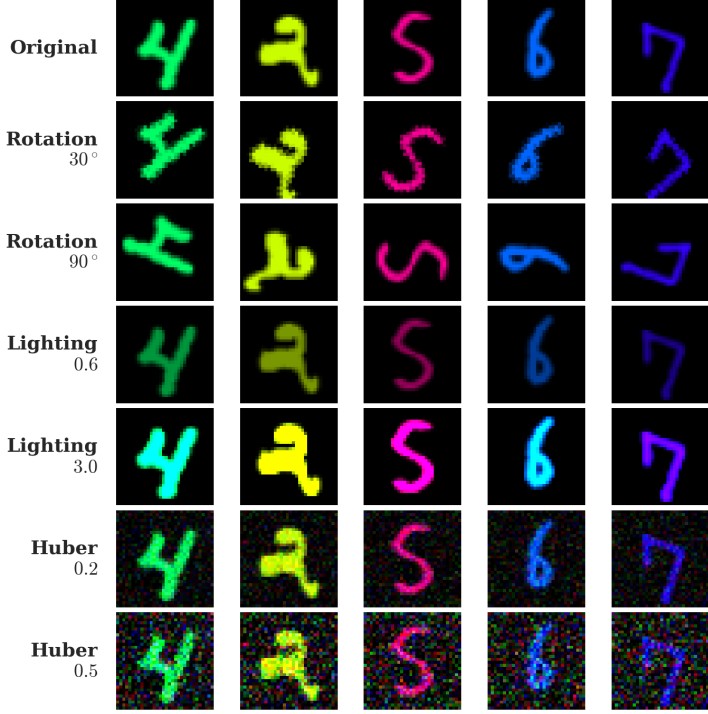

*Figure 15.* Visualizing distributional shifts on `cMNIST`. Top row: Original samples from the test distribution. Subsequent rows display the specific perturbations evaluated in our robustness analysis: Geometric Shifts (rotations of $30°$ and $90°$), Photometric Shifts (lighting adjustments of $\lambda = 0.6$ and $\lambda = 3.0$), and Huber Contamination (additive noise with $\tilde{\sigma} = 0.2$ and $\tilde{\sigma} = 0.5$). These transformations test the stability of learned abstractions against structured geometric changes, environmental lighting variations, and unstructured pixel noise, respectively.

## A.8.2. RESULTS

Below, we present the full experimental results as outlined in the experiments roadmap (Fig. 13). All bolded values in the tables indicate statistically significant differences, determined using paired t-tests with a significance threshold of $p < 0.05$.

*Table 7.* Abstraction error and std under $\alpha = 0$ and $\alpha = 1$ for SLC and LiLUCAS. Top: Gaussian setting with $(\hat{\varepsilon}_\ell, \hat{\varepsilon}_h) = (0.11, 0.11)$. Bottom: Empirical setting with $(\hat{\varepsilon}_\ell, \hat{\varepsilon}_h) = (0.10, 0.03)$

| Method | $\alpha = 0$ | $\alpha = 1$ | $\alpha = 0$ | $\alpha = 1$ |
|---|---|---|---|---|
| | SLC | | LiLUCAS | |
| | Gaussian | | | |
| $\text{BARY}_{(\tau,\omega)}$ | $2.63 \pm 0.01$ | $8.43 \pm 0.01$ | $5.75 \pm 0.01$ | $55.11 \pm 0.02$ |
| $\text{GRAD}_{(\tau,\omega)}$ | $\mathbf{1.25 \pm 0.01}$ | $6.74 \pm 0.00$ | $\mathbf{5.45 \pm 0.02}$ | $64.73 \pm 0.03$ |
| $\text{DIROCA}_{\hat{\epsilon}}$ | $2.67 \pm 0.02$ | $11.28 \pm 0.01$ | $5.86 \pm 0.01$ | $61.86 \pm 0.02$ |
| $\text{DIROCA}_{1,1}$ | $2.01 \pm 0.01$ | $4.14 \pm 0.01$ | $6.97 \pm 0.03$ | $48.05 \pm 0.02$ |
| $\text{DIROCA}_{2,2}$ | $1.74 \pm 0.01$ | $\mathbf{2.37 \pm 0.01}$ | $5.53 \pm 0.03$ | $25.79 \pm 0.01$ |
| $\text{DIROCA}_{4,4}$ | $1.90 \pm 0.01$ | $2.59 \pm 0.01$ | $6.81 \pm 0.03$ | $\mathbf{22.99 \pm 0.01}$ |
| $\text{DIROCA}_{8,8}$ | $1.90 \pm 0.01$ | $2.59 \pm 0.01$ | $6.81 \pm 0.03$ | $\mathbf{23.00 \pm 0.01}$ |
| | Empirical | | | |
| $\text{BARY}_{(\tau,\omega)}$ | $86.46 \pm 0.15$ | $650.33 \pm 0.69$ | $561.19 \pm 3.49$ | $3804.18 \pm 4.39$ |
| $\text{GRAD}_{(\tau,\omega)}$ | $\mathbf{57.04 \pm 0.28}$ | $392.95 \pm 0.30$ | $\mathbf{309.82 \pm 1.71}$ | $1290.65 \pm 1.52$ |
| $\text{AbsLin}_\text{p}$ | $89.29 \pm 0.24$ | $678.89 \pm 0.71$ | $399.21 \pm 1.63$ | $2189.48 \pm 2.15$ |
| $\text{AbsLin}_\text{n}$ | $91.67 \pm 0.29$ | $640.45 \pm 0.68$ | $354.31 \pm 1.46$ | $1878.96 \pm 1.55$ |
| $\text{DIROCA}_{\hat{\epsilon}}$ | $88.39 \pm 0.14$ | $674.40 \pm 0.74$ | $470.13 \pm 3.65$ | $2928.40 \pm 3.41$ |
| $\text{DIROCA}_{1,1}$ | $59.08 \pm 0.33$ | $388.73 \pm 0.31$ | $332.96 \pm 2.98$ | $1385.36 \pm 1.58$ |
| $\text{DIROCA}_{2,2}$ | $58.86 \pm 1.26$ | $\mathbf{357.93 \pm 1.72}$ | $315.21 \pm 7.93$ | $1102.80 \pm 18.15$ |
| $\text{DIROCA}_{4,4}$ | $58.87 \pm 1.25$ | $\mathbf{357.93 \pm 1.72}$ | $311.19 \pm 8.01$ | $\mathbf{981.61 \pm 18.29}$ |
| $\text{DIROCA}_{8,8}$ | $58.89 \pm 1.21$ | $358.20 \pm 1.67$ | $311.43 \pm 8.18$ | $987.53 \pm 17.81$ |

*Table 8.* Average abstraction error (mean $\pm \sigma$) under *model $\mathcal{F}$-misspecification* with sin nonlinearity for the Gaussian setting with $(\hat{\varepsilon}_\ell, \hat{\varepsilon}_h) = (0.11, 0.11)$ and the empirical setting with $(\hat{\varepsilon}_\ell, \hat{\varepsilon}_h) = (0.10, 0.03)$.

| Method | SLC | | LiLUCAS | |
|---|---|---|---|---|
| | Gaussian | Empirical | Gaussian | Empirical |
| $\text{BARY}_{(\tau,\omega)}$ | $1.10 \pm 0.02$ | $185.84 \pm 1.50$ | $4.25 \pm 0.01$ | $578.80 \pm 1.25$ |
| $\text{GRAD}_{(\tau,\omega)}$ | $2.96 \pm 0.01$ | $138.33 \pm 0.96$ | $5.86 \pm 0.03$ | $338.74 \pm 1.01$ |
| $\text{AbsLin}_\text{p}$ | n/a | $198.39 \pm 1.70$ | n/a | $430.34 \pm 1.24$ |
| $\text{AbsLin}_\text{n}$ | n/a | $190.58 \pm 1.67$ | n/a | $381.32 \pm 1.92$ |
| $\text{DIROCA}_{\hat{\epsilon}}$ | $\mathbf{0.40 \pm 0.02}$ | $191.68 \pm 1.54$ | $6.01 \pm 0.03$ | $500.10 \pm 1.44$ |
| $\text{DIROCA}_{1,1}$ | $1.79 \pm 0.02$ | $137.37 \pm 0.92$ | $5.98 \pm 0.03$ | $353.61 \pm 1.83$ |
| $\text{DIROCA}_{2,2}$ | $1.43 \pm 0.01$ | $135.34 \pm 0.51$ | $\mathbf{3.99 \pm 0.03}$ | $333.02 \pm 9.86$ |
| $\text{DIROCA}_{4,4}$ | $2.19 \pm 0.02$ | $135.34 \pm 0.50$ | $4.63 \pm 0.03$ | $\mathbf{327.42 \pm 9.15}$ |
| $\text{DIROCA}_{8,8}$ | $2.19 \pm 0.02$ | $\mathbf{135.32 \pm 0.52}$ | $4.63 \pm 0.03$ | $327.64 \pm 9.00$ |

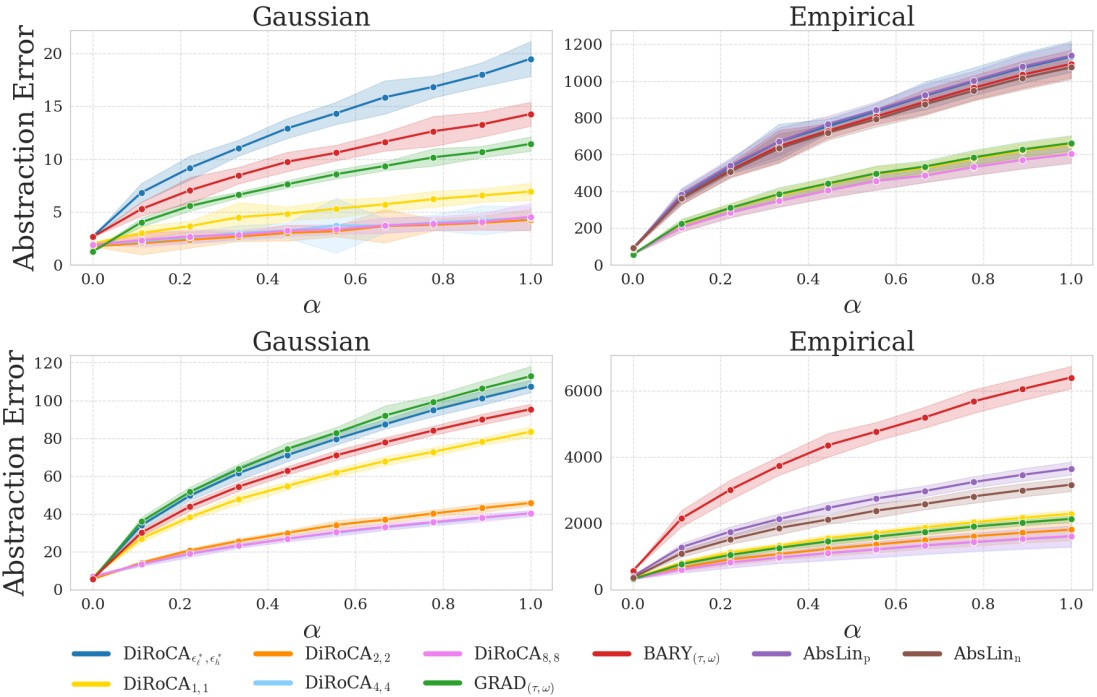

*Figure 16.* Robustness to outlier fraction ($\alpha$) on the `SLC` (top) and `LiLUCAS` (bottom) experiments for the Gaussian (left) and Empirical (right) settings. The evaluation is performed at a fixed *Gaussian* noise intensity ($\tilde{\sigma} = 5.0$ for `SLC` and $\tilde{\sigma} = 10.0$ for `LiLUCAS`)

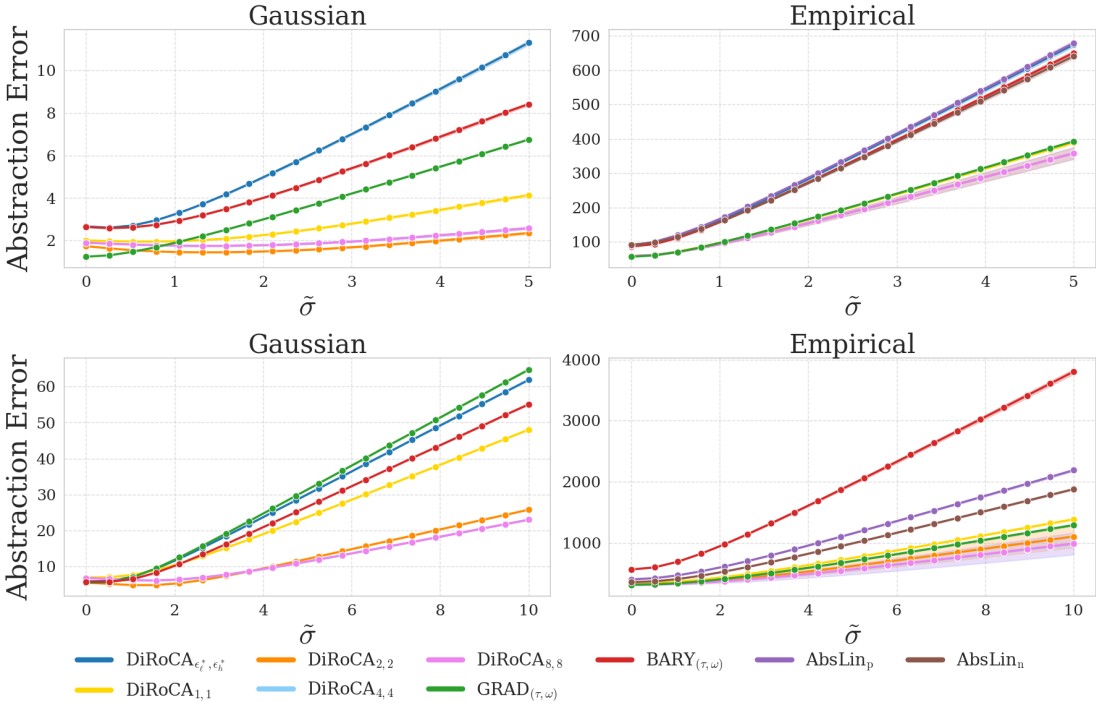

*Figure 17.* Robustness to *Gaussian* noise intensity ($\tilde{\sigma}$) on the `SLC` (top) and `LiLUCAS` (bottom) experiments for the Gaussian (left) and Empirical (right) settings. The evaluation is performed at a fixed outlier fraction of $\alpha = 1.0$ (fully noisy data).

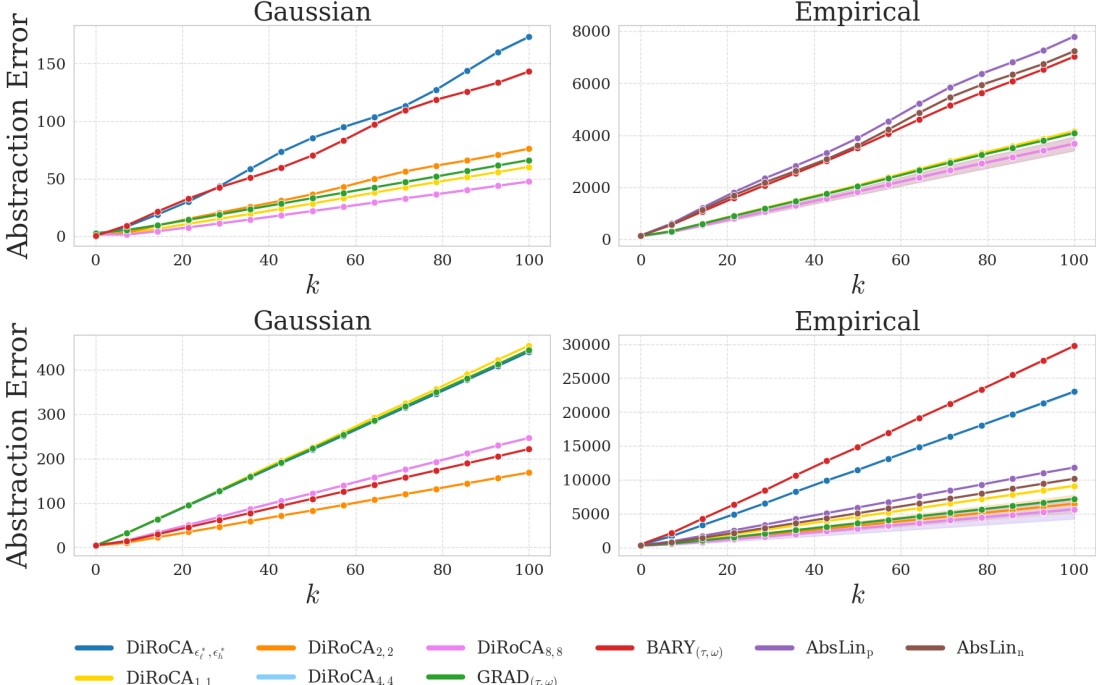

*Figure 18.* Abstraction error comparison as a function of the nonlinearity strength parameter $k$, which controls the strength of the nonlinear component ($\sin$) in the data generation process. The figure shows results for both Gaussian (left) and empirical (right) data distributions in the SLC (top) and LiLUCAS (bottom) settings.

*Table 9.* Average abstraction error (mean $\pm\sigma$) under *model $\mathcal{F}$-misspecification* with $\tanh$ nonlinearity for the Gaussian setting with $(\hat{\varepsilon}_\ell, \hat{\varepsilon}_h) = (0.11, 0.11)$ and the empirical setting with $(\hat{\varepsilon}_\ell, \hat{\varepsilon}_h) = (0.10, 0.03)$.

| Method | SLC | | LiLUCAS | |
|---|---|---|---|---|
| | Gaussian | Empirical | Gaussian | Empirical |
| $\text{BARY}_{(\tau,\omega)}$ | $1.22 \pm 0.01$ | $186.94 \pm 1.39$ | $\mathbf{3.85 \pm 0.01}$ | $564.09 \pm 1.81$ |
| $\text{GRAD}_{(\tau,\omega)}$ | $2.99 \pm 0.01$ | $140.07 \pm 0.87$ | $5.42 \pm 0.04$ | $338.69 \pm 0.93$ |
| $\text{AbsLin}_\text{p}$ | n/a | $195.23 \pm 1.62$ | n/a | $444.13 \pm 1.27$ |
| $\text{AbsLin}_\text{n}$ | n/a | $191.79 \pm 1.57$ | n/a | $386.72 \pm 2.03$ |
| $\text{DIRoCA}_{\hat{\epsilon}}$ | $\mathbf{0.55 \pm 0.01}$ | $192.73 \pm 1.44$ | $5.61 \pm 0.04$ | $486.29 \pm 1.89$ |
| $\text{DIRoCA}_{1,1}$ | $2.05 \pm 0.02$ | $138.89 \pm 0.84$ | $5.86 \pm 0.04$ | $352.50 \pm 1.84$ |
| $\text{DIRoCA}_{2,2}$ | $1.57 \pm 0.01$ | $136.98 \pm 0.42$ | $4.08 \pm 0.03$ | $333.73 \pm 8.63$ |
| $\text{DIRoCA}_{4,4}$ | $2.32 \pm 0.02$ | $136.99 \pm 0.42$ | $4.96 \pm 0.03$ | $\mathbf{328.82 \pm 8.26}$ |
| $\text{DIRoCA}_{8,8}$ | $2.32 \pm 0.02$ | $\mathbf{136.92 \pm 0.45}$ | $4.96 \pm 0.03$ | $329.04 \pm 8.12$ |

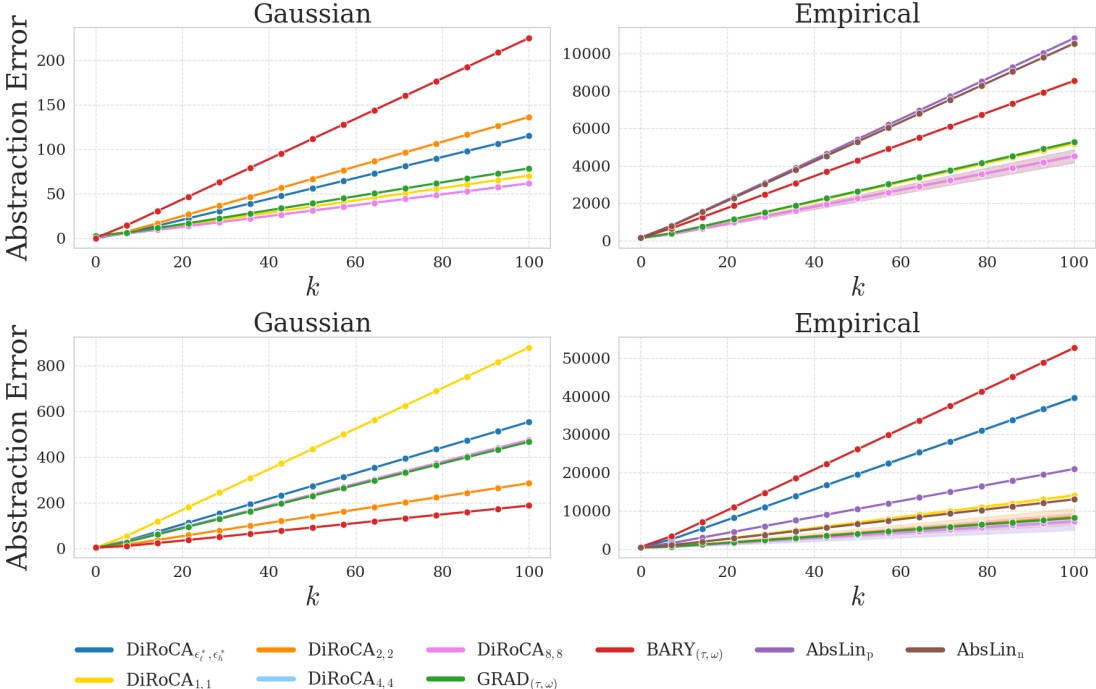

*Figure 19.* Abstraction error comparison as a function of the nonlinearity strength parameter $k$, which controls the strength of the nonlinear component (tanh) in the data generation process. The figure shows results for both Gaussian (left) and empirical (right) data distributions in the SLC (top) and LiLUCAS (bottom) settings.

*Table 10.* Average abstraction error (mean $\pm \sigma$) under *intervention mapping* $\omega$-misspecification for the Gaussian setting with $(\hat{\varepsilon}_\ell, \hat{\varepsilon}_h) = (0.11, 0.11)$ and the empirical setting with $(\hat{\varepsilon}_\ell, \hat{\varepsilon}_h) = (0.10, 0.03)$.

| Method | SLC | | LiLUCAS | |
|---|---|---|---|---|
| | Gaussian | Empirical | Gaussian | Empirical |
| $\text{BARY}_{(\tau,\omega)}$ | $2.77 \pm 0.00$ | $100.98 \pm 0.24$ | $5.86 \pm 0.03$ | $548.55 \pm 2.48$ |
| $\text{GRAD}_{(\tau,\omega)}$ | $\mathbf{0.71 \pm 0.00}$ | $50.39 \pm 0.15$ | $5.63 \pm 0.03$ | $305.46 \pm 1.31$ |
| $\text{AbsLin}_\text{p}$ | n/a | $111.24 \pm 0.17$ | n/a | $414.62 \pm 1.09$ |
| $\text{AbsLin}_\text{n}$ | n/a | $105.50 \pm 0.14$ | n/a | $359.45 \pm 1.44$ |
| $\text{DiRoCA}_{\hat{\epsilon}}$ | $2.57 \pm 0.02$ | $105.03 \pm 0.24$ | $6.08 \pm 0.03$ | $459.06 \pm 2.80$ |
| $\text{DiRoCA}_{1,1}$ | $1.54 \pm 0.00$ | $46.21 \pm 0.14$ | $7.16 \pm 0.03$ | $326.56 \pm 2.59$ |
| $\text{DiRoCA}_{2,2}$ | $1.48 \pm 0.01$ | $\mathbf{39.16 \pm 2.75}$ | $\mathbf{5.55 \pm 0.03}$ | $308.52 \pm 8.54$ |
| $\text{DiRoCA}_{4,4}$ | $1.43 \pm 0.00$ | $\mathbf{39.16 \pm 2.75}$ | $6.86 \pm 0.02$ | $\mathbf{304.11 \pm 7.76}$ |
| $\text{DiRoCA}_{8,8}$ | $1.43 \pm 0.00$ | $39.28 \pm 2.51$ | $6.86 \pm 0.02$ | $304.32 \pm 7.83$ |

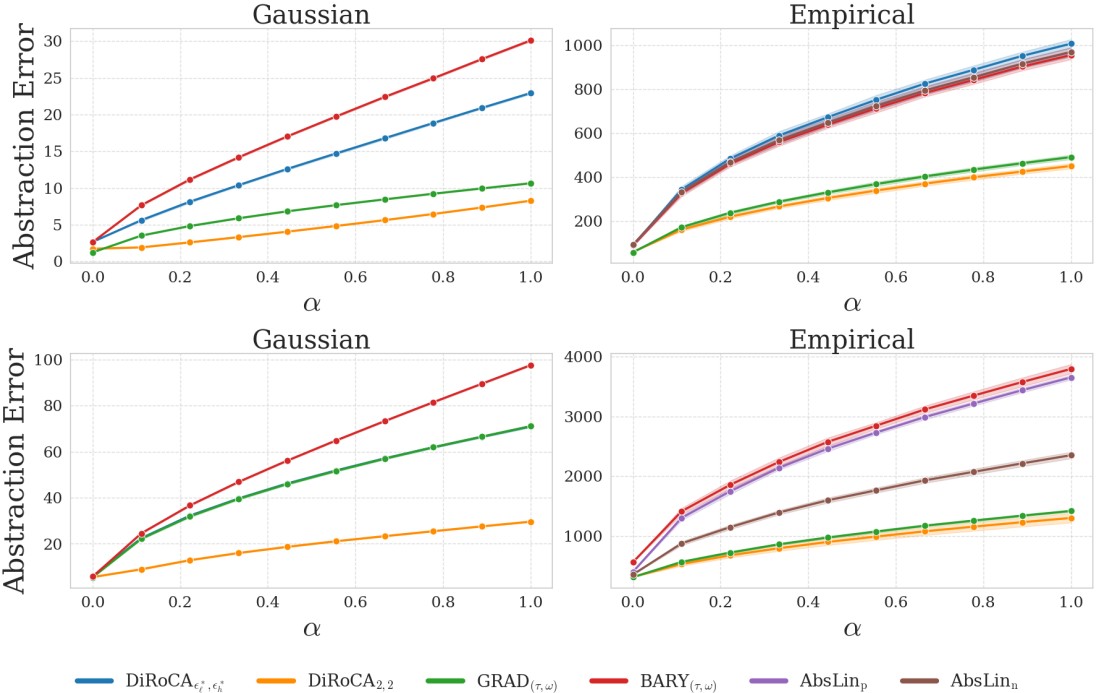

*Figure 20.* Robustness to outlier fraction ($\alpha$) on the **SLC** (top) and `LiLUCAS` (bottom) experiments for the Gaussian (left) and Empirical (right) settings. The evaluation is performed at a fixed *Exponential* noise intensity ($\tilde{\sigma} = 5.0$ for `SLC` and $\tilde{\sigma} = 10.0$ for `LiLUCAS`)

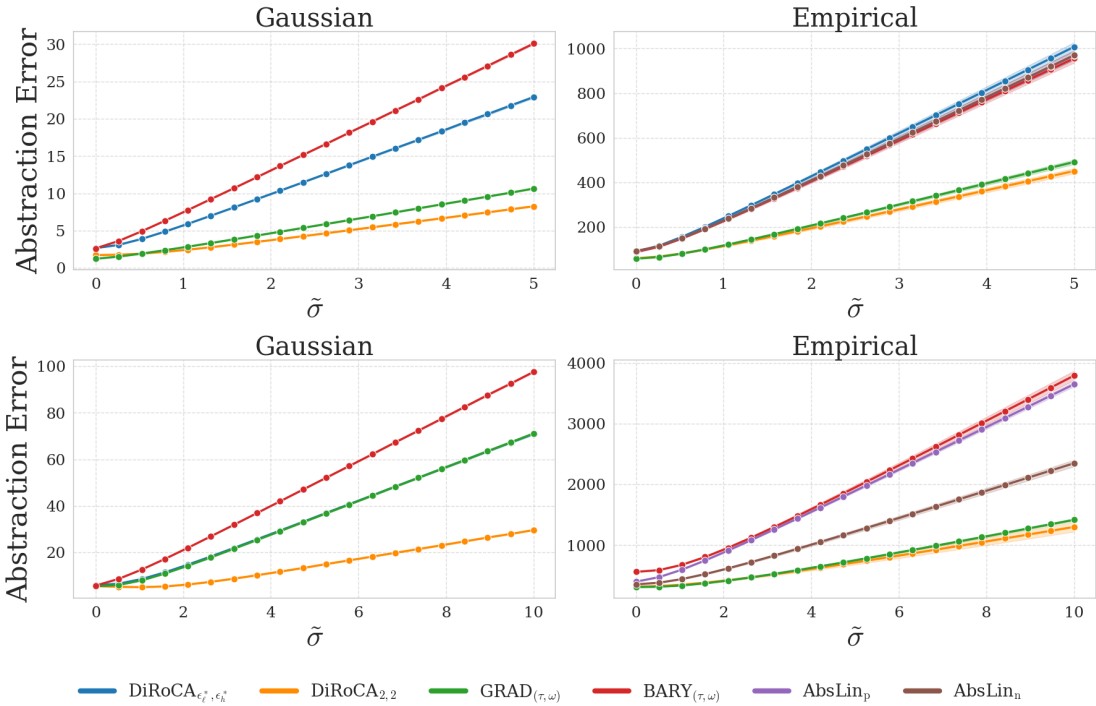

*Figure 21.* Robustness to *Exponential* noise intensity ($\tilde{\sigma}$) on the `SLC` (top) and `LiLUCAS` (bottom) experiments for the Gaussian (left) and Empirical (right) settings. The evaluation is performed at a fixed outlier fraction of $\alpha = 1.0$ (fully noisy data).

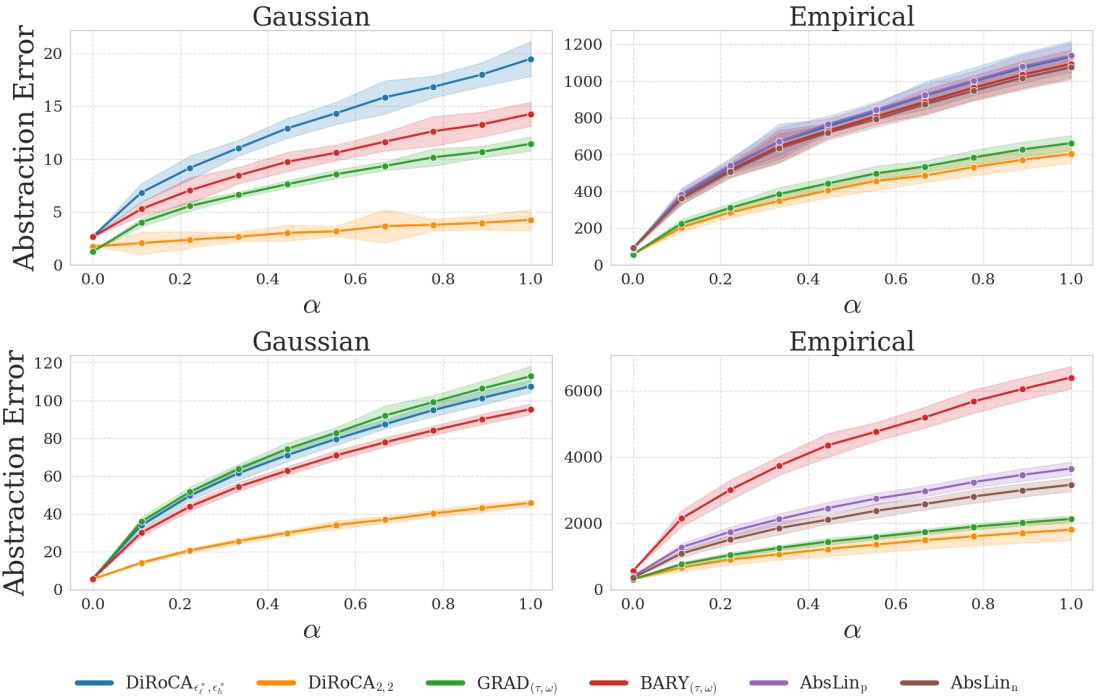

*Figure 22.* Robustness to outlier fraction ($\alpha$) on the `SLC` (top) and `LiLUCAS` (bottom) experiments for the Gaussian (left) and Empirical (right) settings. The evaluation is performed at a fixed *Student-t* noise intensity ($\tilde{\sigma} = 5.0$ for `SLC` and $\tilde{\sigma} = 10.0$ for `LiLUCAS`)

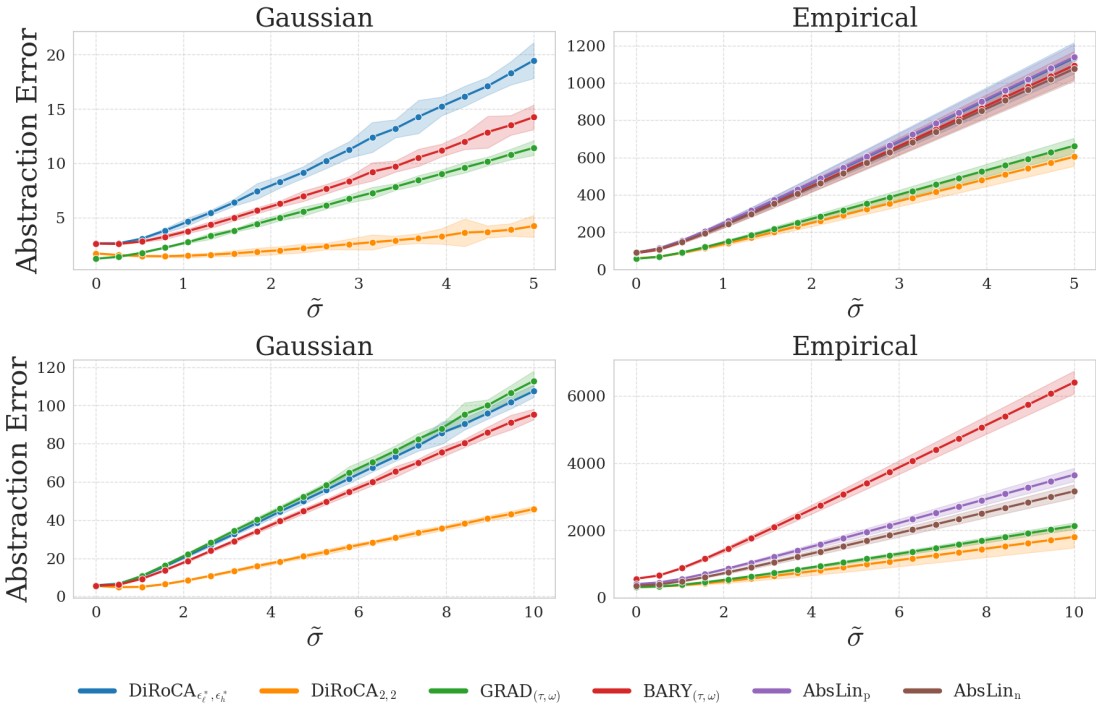

*Figure 23.* Robustness to *Student-t* noise intensity ($\tilde{\sigma}$) on the `SLC` (top) and `LiLUCAS` (bottom) experiments for the Gaussian (left) and Empirical (right) settings. The evaluation is performed at a fixed outlier fraction of $\alpha = 1.0$ (fully noisy data).

*Table 11.* Abstraction error and std under $\alpha = 0$ for `nLUCAS` with $(\hat{\varepsilon}_\ell, \hat{\varepsilon}_h) = (0.47, 0.45)$ comparing two evaluation protocols: **Consistency** (evaluating on the shared noise samples used during training) and **Generalization** (evaluating on new, independent noise samples). See Appendix A.8 for details.

| Method | Consistency | Generalization |
|---|---|---|
| $\text{BARY}_{(\tau,\omega)}$ | $2.98 \pm 0.02$ | $18.74 \pm 0.11$ |
| $\text{GRAD}_{(\tau,\omega)}$ | $\mathbf{2.42 \pm 0.05}$ | $16.39 \pm 0.13$ |
| $\text{AbsLin}_\text{p}$ | $5.94 \pm 0.04$ | $26.09 \pm 0.19$ |
| $\text{AbsLin}_\text{n}$ | $7.93 \pm 0.07$ | $20.36 \pm 0.14$ |
| $\text{DIROCA}_{\hat{\varepsilon}}$ | $2.51 \pm 0.05$ | $15.97 \pm 0.15$ |
| $\text{DIROCA}_{1,1}$ | $6.35 \pm 0.92$ | $10.27 \pm 0.53$ |
| $\text{DIROCA}_{2,2}$ | $8.90 \pm 0.23$ | $9.67 \pm 0.11$ |
| $\text{DIROCA}_{4,4}$ | $13.48 \pm 2.33$ | $\mathbf{8.98 \pm 0.10}$ |
| $\text{DIROCA}_{8,8}$ | $13.49 \pm 2.33$ | $8.98 \pm 0.10$ |

*Table 12.* Abstraction error and std under $\alpha = 1, \tilde{\sigma} = 10.0$ (Gaussian noise) for `nLUCAS` with $(\hat{\varepsilon}_\ell, \hat{\varepsilon}_h) = (0.47, 0.45)$ comparing the Consistency and Generalization evaluation protocols

| Method | Consistency | Generalization |
|---|---|---|
| $\text{BARY}_{(\tau,\omega)}$ | $1008.04 \pm 10.37$ | $1024.79 \pm 10.89$ |
| $\text{GRAD}_{(\tau,\omega)}$ | $863.91 \pm 8.79$ | $877.45 \pm 9.65$ |
| $\text{AbsLin}_\text{p}$ | $1216.24 \pm 12.32$ | $1238.59 \pm 12.69$ |
| $\text{AbsLin}_\text{n}$ | $876.45 \pm 8.77$ | $889.06 \pm 7.01$ |
| $\text{DIROCA}_{\hat{\varepsilon}}$ | $794.99 \pm 8.57$ | $808.21 \pm 8.38$ |
| $\text{DIROCA}_{1,1}$ | $298.52 \pm 44.55$ | $302.46 \pm 45.85$ |
| $\text{DIROCA}_{2,2}$ | $203.82 \pm 2.17$ | $204.80 \pm 2.35$ |
| $\text{DIROCA}_{4,4}$ | $\mathbf{126.14 \pm 23.17}$ | $\mathbf{121.55 \pm 25.34}$ |
| $\text{DIROCA}_{8,8}$ | $\mathbf{126.17 \pm 23.18}$ | $\mathbf{121.54 \pm 25.55}$ |

*Table 13.* Robustness to intervention mapping misspecification on `nLUCAS`: Abstraction error under corrupted intervention map $(\omega)$. Results reported for the Generalization protocol. $\text{DIROCA}_{\hat{\varepsilon}}$ uses radii $(0.47, 0.45)$.

| Method | Error |
|---|---|
| $\text{BARY}_{(\tau,\omega)}$ | $21.37 \pm 1.01$ |
| $\text{GRAD}_{(\tau,\omega)}$ | $18.65 \pm 0.88$ |
| $\text{AbsLin}_\text{p}$ | $29.28 \pm 1.20$ |
| $\text{AbsLin}_\text{n}$ | $22.65 \pm 0.89$ |
| $\text{DIROCA}_{\hat{\varepsilon}}$ | $18.17 \pm 0.86$ |
| $\text{DIROCA}_{1,1}$ | $11.47 \pm 0.78$ |
| $\text{DIROCA}_{2,2}$ | $10.17 \pm 0.24$ |
| $\text{DIROCA}_{4,4}$ | $\mathbf{9.13 \pm 0.29}$ |
| $\text{DIROCA}_{8,8}$ | $\mathbf{9.13 \pm 0.29}$ |

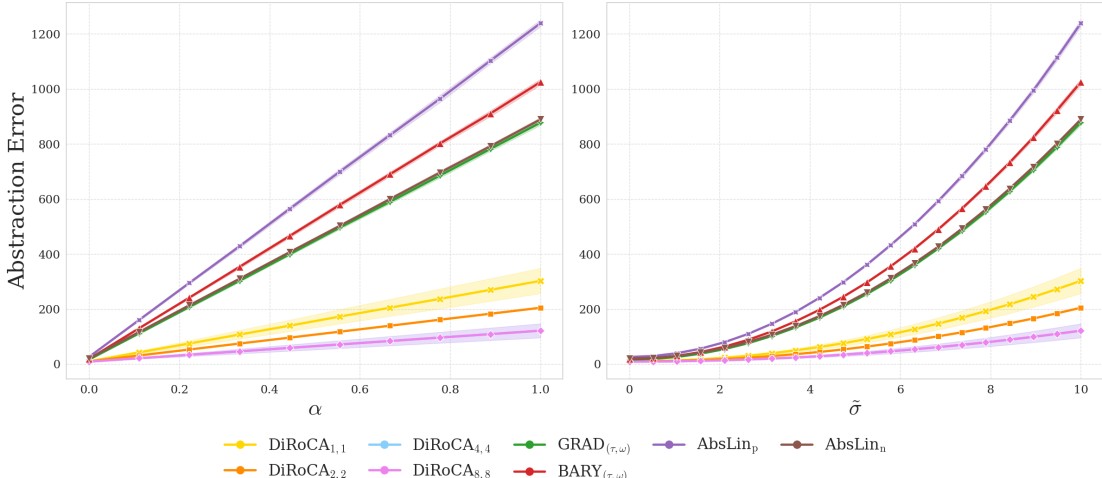

*Figure 24.* For the `nLUCAS` experiment. **Left:** Robustness to outlier fraction ($\alpha$) at a fixed *Gaussian* noise intensity ($\tilde{\sigma} = 10.0$). **Right:** Robustness to *Gaussian* noise intensity ($\tilde{\sigma}$) at a fixed outlier fraction of $\alpha = 1.0$.

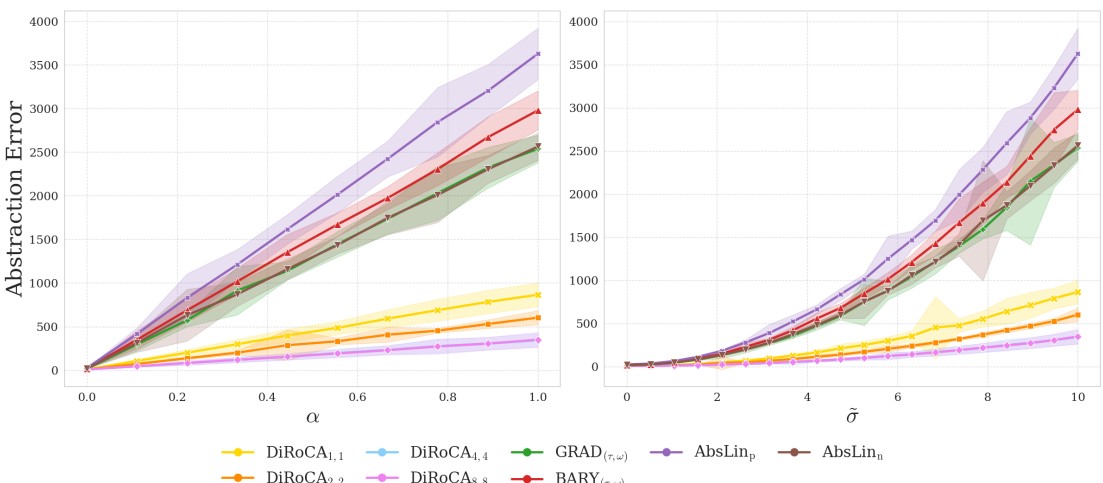

*Figure 25.* For the `nLUCAS` experiment. **Left:** Robustness to outlier fraction ($\alpha$) at a fixed *Student-t* noise intensity ($\tilde{\sigma} = 10.0$). **Right:** Robustness to *Student-t* noise intensity ($\tilde{\sigma}$) at a fixed outlier fraction of $\alpha = 1.0$.

*Table 14.* Abstraction error and std under clean $\alpha = 0$ and fully corrupted $\alpha = 1, \tilde{\sigma} = 10.0$ (Gaussian noise) settings for `EBM` with $(\hat{\varepsilon}_\ell, \hat{\varepsilon}_h) = (0.55, 0.41)$.

| Method | $\alpha = 0$ | $\alpha = 1$ |
|---|---|---|
| AbsLin$_p$ | **$58.85 \pm 11.27$** | $3034.75 \pm 2129.28$ |
| AbsLin$_n$ | $352.82 \pm 47.07$ | $3330.89 \pm 3238.00$ |
| BARY$_{(\tau,\omega)}$ | $84.03 \pm 14.06$ | $357.42 \pm 144.91$ |
| GRAD$_{(\tau,\omega)}$ | $93.53 \pm 26.43$ | $320.15 \pm 148.20$ |
| DIROCA$_{\hat{\epsilon}}$ | $83.96 \pm 28.64$ | **$297.15 \pm 167.75$** |
| DIROCA$_{0.5,0.5}$ | $73.10 \pm 13.12$ | **$268.35 \pm 158.99$** |
| DIROCA$_{1,1}$ | $77.80 \pm 19.93$ | **$268.38 \pm 141.75$** |
| DIROCA$_{2,2}$ | $92.55 \pm 26.11$ | $313.50 \pm 125.72$ |
| DIROCA$_{4,4}$ | $91.76 \pm 27.09$ | $321.39 \pm 166.74$ |
| DIROCA$_{8,8}$ | $91.76 \pm 27.09$ | **$307.86 \pm 172.27$** |

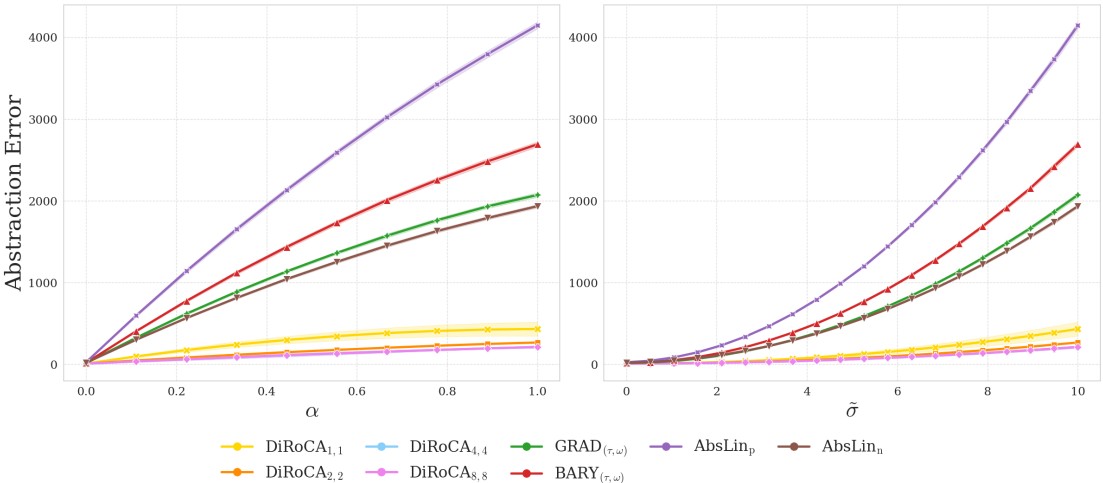

*Figure 26.* For the nLUCAS experiment. **Left:** Robustness to outlier fraction ($\alpha$) at a fixed *Exponential* noise intensity ($\tilde{\sigma} = 10.0$). **Right:** Robustness to *Exponential* noise intensity ($\tilde{\sigma}$) at a fixed outlier fraction of $\alpha = 1.0$.

*Table 15.* Robustness to camera shifts on cMNIST (Relative L2 %) with $(\hat{\varepsilon}_\ell, \hat{\varepsilon}_h) = (61.69, 0)$. We evaluate two distinct shift types at increasing severity levels: **Lighting** (Low: dimming $\lambda = 0.6$, High: overexposure $\lambda = 3.0$) and **Rotation** (Low: $30°$, High: $90°$). DiRoCA remains stable under both shift types, while baselines degrade sharply.

| Method | Lighting | | Rotation | |
|---|---|---|---|---|
| | Low | High | Low | High |
| $\text{BARY}_{(\tau,\omega)}$ | $97.48 \pm 3.10$ | $870.61 \pm 34.88$ | $2287.08 \pm 198.31$ | $4131.29 \pm 363.07$ |
| $\text{GRAD}_{(\tau,\omega)}$ | $32.90 \pm 0.83$ | $330.53 \pm 8.36$ | $1776.67 \pm 235.11$ | $3595.63 \pm 653.21$ |
| $\text{AbsLin}_\text{p}$ | $56.09 \pm 0.87$ | $505.20 \pm 11.27$ | $541.15 \pm 13.20$ | $604.37 \pm 24.12$ |
| $\text{AbsLin}_\text{n}$ | $95.76 \pm 2.68$ | $447.47 \pm 10.69$ | $276.81 \pm 6.65$ | $287.80 \pm 9.26$ |
| $\text{DIRoCA}_{\varepsilon_\ell,0}$ | $35.84 \pm 3.64$ | $137.99 \pm 6.53$ | $101.64 \pm 6.20$ | $106.40 \pm 3.74$ |
| $\text{DIRoCA}_{8,0}$ | $\mathbf{17.82 \pm 0.66}$ | $75.19 \pm 4.03$ | $52.63 \pm 2.04$ | $56.71 \pm 2.59$ |
| $\text{DIRoCA}_{20,0}$ | $18.96 \pm 0.49$ | $\mathbf{26.77 \pm 0.94}$ | $\mathbf{28.02 \pm 0.39}$ | $\mathbf{30.64 \pm 0.72}$ |
| $\text{DIRoCA}_{40,0}$ | $24.13 \pm 2.63$ | $72.93 \pm 5.56$ | $61.48 \pm 4.36$ | $66.60 \pm 2.68$ |

*Table 16.* Huber Noise Robustness on cMNIST (Relative L2 %) for $\alpha = 0$ (Clean) and $\alpha = 1, \tilde{\sigma} = 0.5$ (fully corrupted) with $(\hat{\varepsilon}_\ell, \hat{\varepsilon}_h) = (61.69, 0)$. Values are reported as mean $\pm$ standard deviation across trials. DIRoCA maintains low error even under full noise saturation.

| Method | $\alpha = 0.00$ | $\alpha = 1.00$ |
|---|---|---|
| $\text{BARY}_{(\tau,\omega)}$ | $270.19 \pm 8.69$ | $15338.79 \pm 628.99$ |
| $\text{GRAD}_{(\tau,\omega)}$ | $87.64 \pm 2.17$ | $22207.40 \pm 1501.67$ |
| $\text{AbsLin}_\text{p}$ | $141.64 \pm 2.44$ | $2951.69 \pm 51.59$ |
| $\text{AbsLin}_\text{n}$ | $227.89 \pm 5.78$ | $470.84 \pm 8.45$ |
| $\text{DIRoCA}_{\varepsilon_\ell,0}$ | $69.80 \pm 5.04$ | $214.91 \pm 8.42$ |
| $\text{DIRoCA}_{8.0,0}$ | $35.97 \pm 1.70$ | $60.33 \pm 1.73$ |
| $\text{DIRoCA}_{20,0}$ | $\mathbf{20.05 \pm 0.70}$ | $\mathbf{48.17 \pm 0.64}$ |
| $\text{DIRoCA}_{40,0}$ | $38.68 \pm 3.20$ | $132.86 \pm 7.48$ |

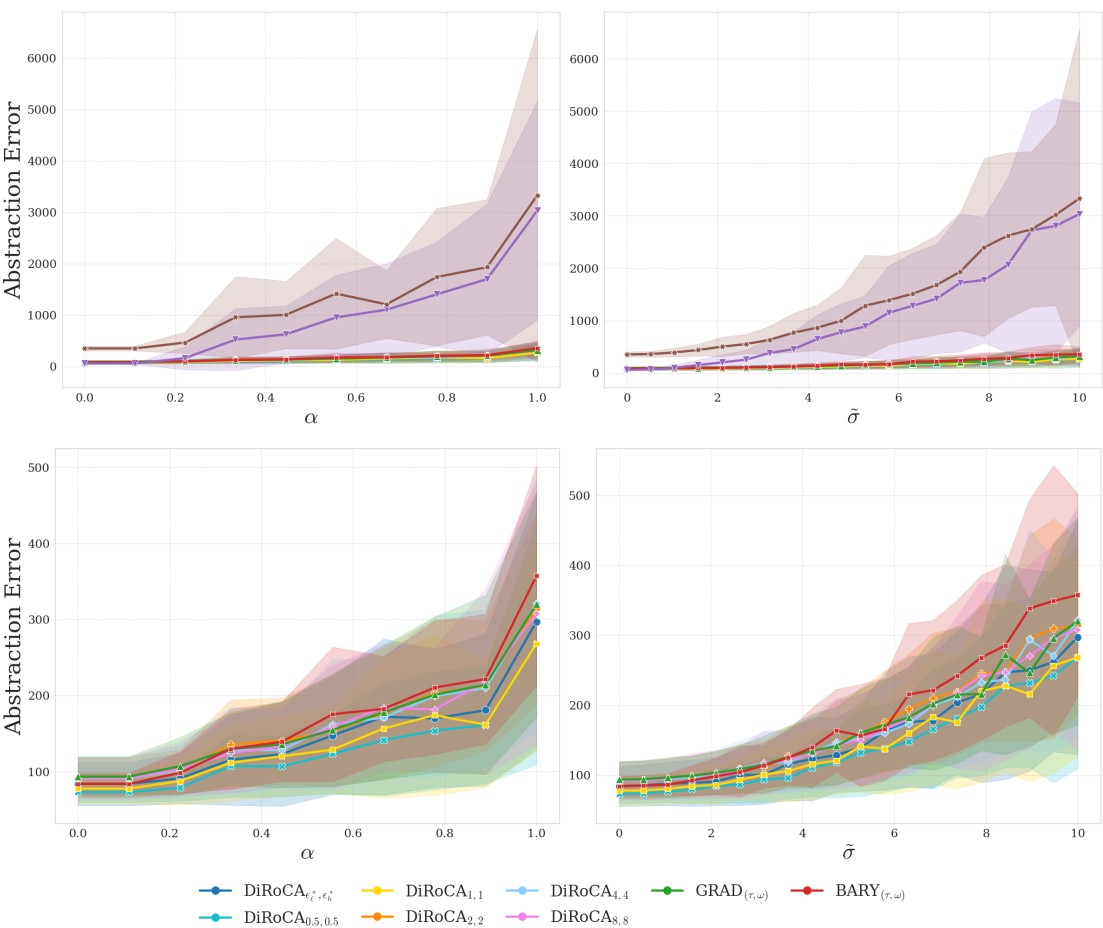

*Figure 27.* For the EBM experiment. **Left:** Robustness to outlier fraction ($\alpha$) at a fixed *Student-t* noise intensity ($\tilde{\sigma} = 10.0$). **Right:** Robustness to *Student-t* noise intensity ($\tilde{\sigma}$) at a fixed outlier fraction of $\alpha = 1.0$. The bottom row presents results with the AbsLin$_\mathbb{P}$ and AbsLin$_\mathbb{n}$ baselines omitted for clearer visualization.

## A.9. Optimization

### A.9.1. DISTRIBUTIONALLY ROBUST CAUSAL ABSTRACTION LEARNING (DIROCA)

In this section, we provide a detailed analytical presentation of the optimization procedures of DIROCA starting from the Linear models both for the Gaussian and empirical settings.

**Gaussian case.** Recall that in the Gaussian case DIROCA solves the following constrained min-max optimization problem:

$$\min_{\mathrm{T}\in\mathbb{R}^{h\times\ell}} \max_{\substack{\mu^\ell,\Sigma^\ell \\ \mu^h,\Sigma^h}} \mathbb{E}_{\iota\sim q}\Big[\mathcal{W}_2^2\big(\mathcal{N}(\mathrm{T}\,\mathbf{L}_\iota\mu^\ell, \mathrm{T}\,\mathbf{L}_\iota\Sigma^\ell\mathbf{L}_\iota^\top\,\mathrm{T}^\top), \mathcal{N}(\mathbf{H}_{\omega(\iota)}\mu^h, \mathbf{H}_{\omega(\iota)}\Sigma^h\mathbf{H}_{\omega(\iota)}^\top))\big)\Big] \tag{70}$$

supported by the environmental uncertainty constraints:

$$\mathcal{W}_2^2(\mathcal{N}(\mu^\ell,\Sigma^\ell), \mathcal{N}(\widehat{\mu^\ell},\widehat{\Sigma^\ell})) \leqslant \varepsilon_\ell^2, \quad \mathcal{W}_2^2(\mathcal{N}(\mu^h,\Sigma^h), \mathcal{N}(\widehat{\mu^h},\widehat{\Sigma^h})) \leqslant \varepsilon_h^2 \tag{71}$$

where $\widehat{\mu^\ell}, \widehat{\Sigma^\ell}, \widehat{\mu^h}, \widehat{\Sigma^h}$ are empirical estimates from observed data. Since we work with Markovian SCMs, the exogenous variables are jointly independent, which implies that their covariance matrices are diagonal. This structural property simplifies the form of the Wasserstein constraints, allowing them to be expressed as:

$$\|\mu^\ell - \widehat{\mu^\ell}\|_2^2 + \|\Sigma^{\ell 1/2} - \widehat{\Sigma^\ell}^{1/2}\|_2^2 \leqslant \varepsilon_\ell^2 \quad \text{and} \quad \|\mu^h - \widehat{\mu^h}\|_2^2 + \|\Sigma^{h 1/2} - \widehat{\Sigma^h}^{1/2}\|_2^2 \leqslant \varepsilon_h^2 \tag{72}$$

Expanding the Gelbrich formula for the Gaussian 2-Wasserstein distance, the objective decomposes as:

$$F(\mathrm{T}, \mu^\ell, \Sigma^\ell, \mu^h, \Sigma^h) = \underbrace{\mathbb{E}_{\iota \sim q}\Big[\| \mathrm{T}\,\mathbf{L}_\iota \mu^\ell - \mathbf{H}_{\omega(\iota)}\mu^h\|_2^2 + \mathrm{Tr}(\mathrm{T}\,\mathbf{L}_\iota \Sigma^\ell \mathbf{L}_\iota^\top \mathrm{T}^\top) + \mathrm{Tr}(\mathbf{H}_{\omega(\iota)}\Sigma^h \mathbf{H}_{\omega(\iota)}^\top)\Big]}_{F_s \text{ (smooth term)}} \tag{73}$$

$$\underbrace{-\mathbb{E}_{\iota \sim q}\Big[2\,\mathrm{Tr}\left((\mathbf{H}_{\omega(\iota)}\Sigma^h \mathbf{H}_{\omega(\iota)}^\top)^{1/2}(\mathrm{T}\,\mathbf{L}_\iota \Sigma^\ell \mathbf{L}_\iota^\top \mathrm{T}^\top)(\mathbf{H}_{\omega(\iota)}\Sigma^h \mathbf{H}_{\omega(\iota)}^\top)^{1/2}\right)^{1/2}\Big]}_{F_n \text{ (non-smooth term)}} \tag{74}$$

We rewrite the objective as the sum of a *smooth* component $F_s$ capturing the quadratic terms and a *non-smooth* component $F_n$ capturing the last trace term. Thus the objective becomes:

$$\min_{\mathrm{T}} \max_{\mu^\ell, \Sigma^\ell, \mu^h, \Sigma^h} F_s(\mathrm{T}, \mu^\ell, \Sigma^\ell, \mu^h, \Sigma^h) + F_n(\mathrm{T}, \Sigma^\ell, \Sigma^h) \tag{75}$$

Let $S_\ell = \mathrm{T}\,\mathbf{L}_\iota \Sigma^\ell \mathbf{L}_\iota^\top \mathrm{T}^\top$ and $S_h = \mathbf{H}_{\omega(\iota)}\Sigma^h \mathbf{H}_{\omega(\iota)}^\top$. Then the non-smooth term simplifies as:

$$\mathrm{Tr}\left( \left(S_h^{1/2} S_\ell S_h^{1/2}\right)^{1/2} \right) = \|S_\ell^{1/2} S_h^{1/2}\|_* \tag{76}$$

To handle the non-smoothness of the nuclear norm in optimization, we use the variational characterization from Srebro et al. (2004). For any matrix $X = UV^\top$,

$$\|X\|_* = \min_{\substack{U,V: \\ X=UV^\top}} \|U\|_F \|V\|_F \tag{77}$$

In our case:

$$\|S_\ell^{1/2} S_h^{1/2}\|_* \leqslant \|S_\ell^{1/2}\|_F \|S_h^{1/2}\|_F \tag{78}$$

Applying this inequality, we upper bound the non-smooth term, allowing the optimization to proceed efficiently using standard gradient and proximal-gradient methods. Specifically, we maximize a surrogate lower bound $\tilde{F} \leqslant F$:

$$\tilde{F} := F_s(\mathrm{T}, \mu^\ell, \Sigma^\ell, \mu^h, \Sigma^h) + F_n^*(\mathrm{T}, \Sigma^\ell, \Sigma^h) \tag{79}$$

where $F_n^*(\mathrm{T}, \Sigma^\ell, \Sigma^h) = -\|(\mathrm{T}\,\mathbf{L}_\iota \Sigma^\ell \mathbf{L}_\iota^\top \mathrm{T}^\top)^{1/2}\|_F \|(\mathbf{H}_{\omega(\iota)}\Sigma^h \mathbf{H}_{\omega(\iota)}^\top)^{1/2}\|_F$.

The optimization algorithm alternates between updating $\mathrm{T}$ via gradient descent on (73), and updating the moments $(\mu^\ell, \Sigma^\ell), (\mu^h, \Sigma^h)$ via proximal-gradient ascent on (79) (see Algorithm 3). Figures 28a and 28b provide a visual illustration, showing the initial empirical distribution ($\mathcal{N}(\hat{\mu}, \hat{\Sigma})$) and the final worst-case distribution ($\mathcal{N}(\mu^\star, \Sigma^\star)$) learned by the adversary for our two experimental datasets.

**Note 1.** The proximal operator of the Frobenius norm of a matrix $A \in R^{m \times n}$ is given as

$$\mathrm{prox}_{\lambda \|\cdot\|_F}(A) = \begin{cases} \left(I - \frac{\lambda}{\|A\|_F}\right) A, & \text{if } \|A\|_F > \lambda \\ 0, & \text{otherwise} \end{cases}. \tag{80}$$

Here $A$ denotes the square root of the transformed covariance matrices (e.g., $(\mathrm{T}\,L_\iota \Sigma^\ell L_\iota^\top \mathrm{T}^\top)^{1/2}$ and $(H_{\omega(\iota)}\Sigma^h H_{\omega(\iota)}^\top)^{1/2}$).

**Note 2.** The projection operation $\mathrm{proj}_{\mathcal{W}_2 \leqslant \varepsilon_h}(\cdot, \cdot, \cdot, \cdot)$ enforces the Wasserstein constraints:

$$\mathcal{W}_2^2\big(\mathcal{N}(\mu, \Sigma), \mathcal{N}(\hat{\mu}, \hat{\Sigma})\big) \leqslant \varepsilon^2 \quad \text{via projection onto a Gelbrich ball.} \tag{81}$$

This is done iteratively using the update:

$$(\mu, \Sigma) \leftarrow \hat{\mu} + \alpha(\mu - \hat{\mu}), \; \hat{\Sigma}^{1/2} + \alpha(\Sigma^{1/2} - \hat{\Sigma}^{1/2}) \tag{82}$$

with a scaling factor $\alpha = \epsilon / \mathcal{W}_2((\mu, \Sigma), (\hat{\mu}, \hat{\Sigma}))$.

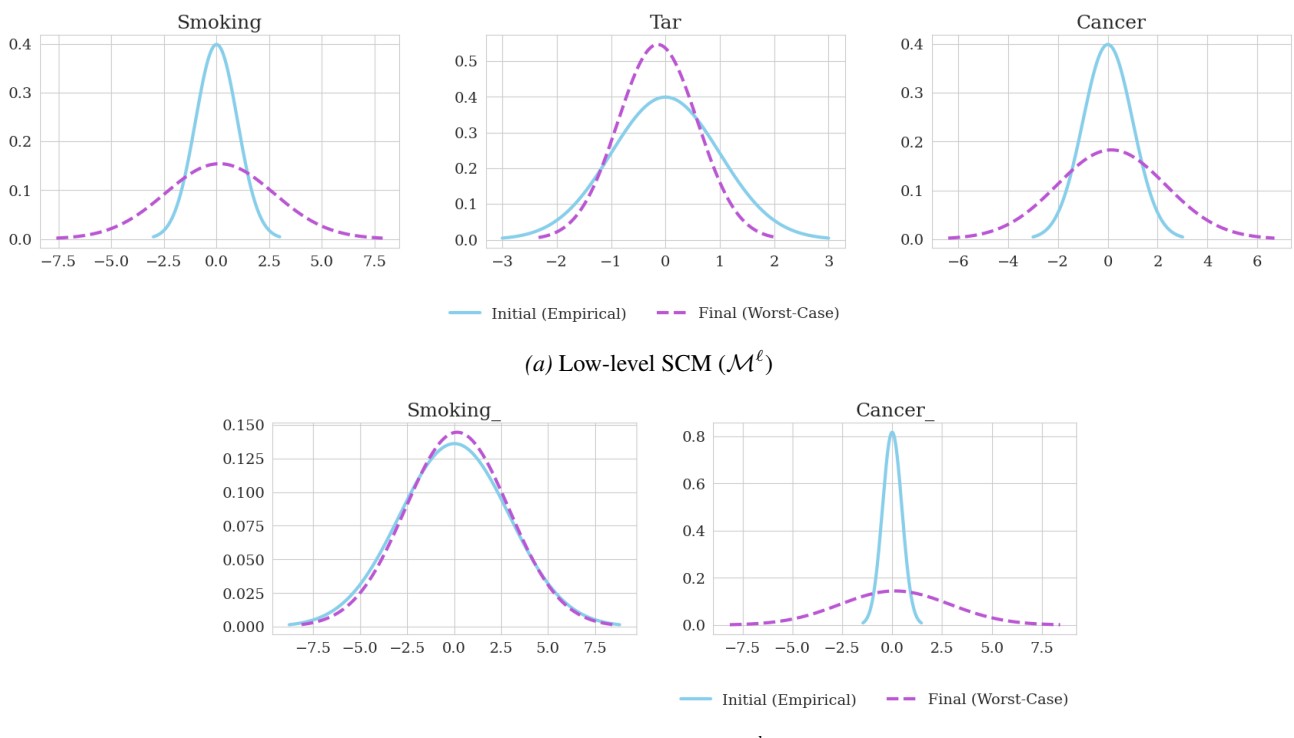

*(a)* Low-level SCM $(\mathcal{M}^\ell)$

*(b)* High-level SCM $(\mathcal{M}^h)$

*Figure 28.* Marginal noise distributions for each variable in the low-level (top) and high-level (bottom) SCMs of the `SLC` dataset. In each subplot, the solid blue curve represents the initial empirical noise distribution, and the dashed purple curve shows the corresponding worst-case distribution obtained via the distributionally robust optimization.

**Empirical case.** When there are no structural assumptions about the distributional form of the environment, the empirical formulation provides a fully data-driven robust alternative, particularly useful when the low- and high-level environments are available only through finite samples. In this case, we introduce perturbation matrices $\Theta_d \in \mathbb{R}^{N \times d}$ over the empirical exogenous samples and define $\rho^d(\Theta_d) = \frac{1}{N}\sum_{i=1}^{N} \delta_{\hat{u}_i + \theta_i}$. The objective then measures the squared Frobenius discrepancy between the perturbed distributions of the two SCMs:

$$F(\mathrm{T}, \Theta_\ell, \Theta_h) := \mathbb{E}_{\iota \sim q}\left[\left\|\mathrm{T}\ \mathbf{L}_\iota(\mathrm{U}^\ell + \Theta_\ell) - \mathbf{H}_{\omega(\iota)}(\mathrm{U}^h + \Theta_h)\right\|_{\mathrm{F}}^2\right], \tag{83}$$

where $\mathrm{U}^\ell, \mathrm{U}^h$ are the empirical exogenous samples for $\mathcal{M}^\ell$ and $\mathcal{M}^h$, respectively. The perturbations capture adversarial shifts around these nominal empirical distributions. A proposed reparameterization reduces the Wasserstein ambiguity sets to constrained Frobenius norm balls (Kuhn et al., 2019):

$$\|\Theta_\ell\|_F \leqslant \varepsilon_\ell \sqrt{N_\ell}, \quad \|\Theta_h\|_F \leqslant \varepsilon_h \sqrt{N_h} \tag{84}$$

We solve this via alternating projected gradient descent. The optimization alternates between updating the abstraction map $\mathrm{T}$ with gradient descent and the adversarial perturbations of the samples through projected gradient ascent. Specifically,

- **Minimization step (updating $\mathrm{T}$):** Fix perturbations $(\Theta_\ell, \Theta_h)$ applied to the samples and perform gradient descent to update the abstraction map $\mathrm{T}$ by minimizing the empirical loss.

- **Maximization step (updating perturbations $\Theta_\ell, \Theta_h$):** Fix $\mathrm{T}$ and perform adversarial updates to the perturbations:
  - Perturbations $\Theta_\ell$ are added to $U^\ell$ and $\Theta_h$ to $U^h$.
  - Gradient ascent steps are performed to maximize the abstraction error.
  - After each ascent step, perturbations are projected onto Frobenius norm balls to ensure they lie within a fixed robustness radius ($\varepsilon_\ell$ for $\Theta$, $\varepsilon_h$ for $\Theta_h$), enforcing constraints derived from the Wasserstein DRO ambiguity sets.

**General Case.** For general SCMs where mechanisms can be nonlinear, we rely on the $(D, U)$ decomposition detailed in Appendix A.5.1. Unlike the linear case where mechanism matrices can be explicitly multiplied, here we operate on the abduced residuals directly.

We define the empirical marginal environments as $\widehat{\rho}^d = \frac{1}{N} \sum_{i=1}^N \delta_{\hat{u}_i^d}$ for $d \in \{\ell, h\}$, and the joint environment as $\widehat{\rho} = \widehat{\rho^\ell} \otimes \widehat{\rho^h}$. By the finite-dimensional reduction of Kuhn et al. (2019), any distribution in a 2-Wasserstein ball around an empirical measure can be represented as a perturbed empirical distribution. We thus perturb the noise samples as $U^d \mapsto U^d + \Theta_d$, where $\Theta_d \in \mathbb{R}^{N \times d}$. The ambiguity radii $\varepsilon_\ell, \varepsilon_h$ correspond to Frobenius budgets $r_d = \varepsilon_d \sqrt{N}$, such that the condition $\|\Theta_d\|_F \leqslant r_d$ is analytically equivalent to restricting the joint environment to $\mathbb{B}_{\epsilon,2}(\widehat{\rho})$.

To formulate the robust objective, let $\mathbf{D}_\ell^{(\iota)}$, $\mathbf{D}_h^{(\omega(\iota))}$ be the pre-computed deterministic component matrices for intervention $\iota$ and the shared observational residuals are $\mathbf{U}^\ell$ and $\mathbf{U}^h$. We define the *nominal misalignment* for an abstraction T as:

$$Z_{\mathrm{T}}^\iota(\mathbf{0}) := \mathrm{T}(\mathbf{D}_\ell^{(\iota)} + \mathbf{U}^\ell)^\top - (\mathbf{D}_h^{(\omega(\iota))} + \mathbf{U}^h). \tag{85}$$

Introducing perturbations $\mathbf{\Theta} := (\Theta_\ell, \Theta_h)$, the *perturbed misalignment* shifts to:

$$Z_{\mathrm{T}}^\iota(\mathbf{\Theta}) := Z_{\mathrm{T}}^\iota(\mathbf{0}) + (\mathrm{T}\,\Theta_\ell^\top - \Theta_h). \tag{86}$$

The DIROCA objective minimizes the expected perturbed misalignment under the worst-case observational shift within the ambiguity set:

$$\min_{\mathrm{T}} \sup_{\|\Theta_\ell\|_F \leqslant r_\ell,\ \|\Theta_h\|_F \leqslant r_h} \mathbb{E}_{\iota \sim q}\left[\|Z_{\mathrm{T}}^\iota(\mathbf{\Theta})\|_F^2\right]. \tag{87}$$

This formulation highlights that the adversary attempts to align the perturbation shift $(\mathrm{T}\,\Theta_\ell^\top - \Theta_h)$ with the nominal direction $Z_{\mathrm{T}}^\iota(\mathbf{0})$ to maximize the error. We optimize this problem via Alternating Projected Gradient Descent-Ascent, where the perturbations $\Theta$ are updated via projected gradient ascent (projecting onto the Frobenius balls at each step) and the abstraction T is updated via gradient descent, as detailed in Algorithm 5.

### A.9.2. BARYCENTRIC ABSTRACTION LEARNING (BARY$_{(\tau,\omega)}$).

We also consider an abstraction learning method based on *barycentric aggregation* of interventional distributions, inspired by Felekis et al. (2024).

**Gaussian case.** In the case of linear models when the environments are Gaussian distributions, the key idea is to aggregate the different interventional distributions at both the low- and high-level SCMs into representative barycentric means and covariances, project the low-level distribution into the high-level space, and then minimize the Wasserstein distance between these two distributions.

Formally, given mixing matrices $\{\mathbf{L}_\iota\}_{\iota \in \mathcal{I}^\ell}$ and $\{\mathbf{H}_\eta\}_{\eta \in \mathcal{I}^h}$ corresponding to the low- and high-level SCMs under interventions, we define the *barycentric mean* at each level according to Eq. 21:

$$\mu_{\mathrm{bary}}^\ell = \frac{1}{|\mathcal{I}^\ell|} \sum_{\iota \in \mathcal{I}^\ell} \mathbf{L}_\iota \mu_U^\ell, \quad \mu_{\mathrm{bary}}^h = \frac{1}{|\mathcal{I}^h|} \sum_{\eta \in \mathcal{I}^h} \mathbf{H}_\eta \mu_U^h. \tag{88}$$

The *barycentric covariance* at the low level, $\Sigma_{\mathrm{bary}}^\ell$, is obtained as the Wasserstein barycenter of the set $\{\mathbf{L}_\iota \Sigma_U^\ell \mathbf{L}_\iota^\top\}_{\iota \in \mathcal{I}^\ell}$, solving the fixed-point equation from Eq. 22:

$$\Sigma_{\mathrm{bary}}^\ell = \frac{1}{|\mathcal{I}^\ell|} \sum_{\iota \in \mathcal{I}^\ell} \lambda_\iota \left(\Sigma_{\mathrm{bary}}^{\ell\,1/2} \left(\mathbf{L}_\iota \Sigma_U^\ell \mathbf{L}_\iota^\top\right) \Sigma_{\mathrm{bary}}^{\ell\,1/2}\right)^{1/2}, \tag{89}$$

The barycentric covariance $\Sigma_{\mathrm{bary}}^h$ at the high level is computed analogously over the set $\{\mathbf{H}_\eta \Sigma_U^h \mathbf{H}_\eta^\top\}_{\eta \in \mathcal{I}^h}$. To map from the low-level to the high-level space, we first project the low-level barycentric distribution onto the high-level dimension. Specifically, letting $V \in \mathbb{R}^{\ell \times h}$ denote the projection matrix, obtained via PCA, we define the projected mean and covariance:

$$\mu_{\mathrm{proj}}^\ell = V^\top \mu_{\mathrm{bary}}^\ell, \quad \Sigma_{\mathrm{proj}}^\ell = V^\top \Sigma_{\mathrm{bary}}^\ell V. \tag{90}$$

Consequently, by defining the abstraction map as the optimal transport Monge map between the distributions $\mathcal{N}(\mu^\ell_{\text{proj}}, \Sigma^\ell_{\text{proj}})$ and $\mathcal{N}(\mu^h_{\text{bary}}, \Sigma^h_{\text{bary}})$, from Eq. 17, since both distributions are Gaussian, this admits the following closed-form expression:

$$T(x) = \mu^h_{\text{bary}} + A(x - \mu^\ell_{\text{proj}}), \quad \text{where} \quad A = {\Sigma^h_{\text{bary}}}^{1/2} \left(\Sigma^\ell_{\text{proj}}\right)^{-1/2}. \tag{91}$$

Finally, since we initially projected the low-level distribution, the full transformation matrix is:

$$\mathrm{T} = AV^\top, \tag{92}$$

In both cases, the resulting abstraction approximately aligns the barycentric summaries of the low- and high-level interventional distributions.

**Empirical case.** In the linear setting where only empirical samples of the environments are available and no structural assumptions are made about their underlying distributions, we simply take the arithmetic mean across all the interventions of the mixing matrices $\mathbf{L}_{bary}$ and $\mathbf{H}_{bary}$ and propagate the exogenous datasets $\mathrm{U}^\ell$ and $\mathrm{U}^h$ through them and solve with gradient descent the following problem:

$$\min_{\mathrm{T} \in \mathbb{R}^{h \times \ell}} \left\| \mathrm{T}\, \mathbf{L}_{bary}(\mathrm{U}^\ell) - \mathbf{H}_{bary}(\mathrm{U}^h) \right\|^2_{\mathrm{F}} \tag{93}$$

While $\mathrm{BARY}_{(\tau,\omega)}$ is computationally efficient, it does not explicitly optimize robustness to distributional shifts, unlike DIROCA.

**General Case.** In the general setting (including nonlinear ANMs), we implement $\mathrm{BARY}_{(\tau,\omega)}$ using the $(D, U)$ decomposition. Instead of averaging linear mixing matrices, we compute the *barycentric deterministic component* by taking the arithmetic mean of the pre-computed deterministic matrices $\mathbf{D}^{(\iota)}$ across all training interventions. This yields an "average mechanism" $\overline{\mathbf{D}}^\ell = \frac{1}{|\mathcal{I}^\ell|} \sum_\iota \mathbf{D}^{(\iota)}_\ell$ and $\overline{\mathbf{D}}^h = \frac{1}{|\mathcal{I}^h|} \sum_\eta \mathbf{D}^{(\eta)}_h$. We then form a representative barycentric dataset by combining these centroids with the empirical residuals: $\mathbf{X}^\ell_{\text{bary}} = \overline{\mathbf{D}}^\ell + \mathbf{U}^\ell$ and $\mathbf{X}^h_{\text{bary}} = \overline{\mathbf{D}}^h + \mathbf{U}^h$. The abstraction $\mathrm{T}$ is learned by minimizing the Frobenius error between these two representative datasets, effectively aligning the "average" causal behavior of the low-level system to that of the high-level system, while ignoring specific interventional variations.

---

**Algorithm 2** General $\mathrm{BARY}_{\tau\omega}$ $\hfill [d \in \{\ell, h\}]$

---
**Require:** $\omega, \mathcal{I}^d, \mathcal{G}_{\mathcal{M}^d}, \{\mathbf{X}^d_\iota\}_{\iota \in \mathcal{I}^d}$
**Require:** Hyperparameters: $\eta_\tau$ (learning rate)
1: *// Step 1: Abduction and deterministic computation*
2: $\mathbf{U}^d \leftarrow \text{Abduct}(\mathbf{X}^d, \mathcal{M}^d)$ $\hfill$ *#Abduct observational noise*
3: *// Step 2: Compute barycentric deterministic components*
4: $\overline{\mathbf{D}}^\ell \leftarrow \frac{1}{|\mathcal{I}^\ell|} \sum_{\iota \in \mathcal{I}^\ell} \mathbf{D}^{(\iota)}_\ell$
5: $\overline{\mathbf{D}}^h \leftarrow \frac{1}{|\mathcal{I}^h|} \sum_{\eta \in \mathcal{I}^h} \mathbf{D}^{(\eta)}_h$ $\hfill$ *#Average mechanism*
6: *// Step 3: Optimization*
7: **Initialize:** $\mathrm{T}^{(0)}$
8: **repeat**
9: $\quad \mathcal{L}(\mathrm{T}) \coloneqq \left\| \mathrm{T}(\overline{\mathbf{D}}^\ell + \mathbf{U}^\ell)^\top - (\overline{\mathbf{D}}^h + \mathbf{U}^h) \right\|^2_F$
10: $\quad \mathrm{T}^{(t+1)} \leftarrow \mathrm{T}^{(t)} - \eta_\tau \nabla_\mathrm{T} \mathcal{L}(\mathrm{T}^{(t)})$
11: **until** convergence
$\quad$**Return:** $\mathrm{T}^\star$

---

---

**Algorithm 3** Gaussian DIROCA for linear models

---

**Require:** $\mathcal{S}^\ell, \mathcal{S}^h, \mathcal{I}^\ell, \mathcal{I}^h, \omega : \mathcal{I}^\ell \to \mathcal{I}^h$, samples from $\mathbb{P}_{\mathcal{M}_\iota^\ell}(\mathcal{X}^\ell), \mathbb{P}_{\mathcal{M}_{\omega(\iota)}^h}(\mathcal{X}^h)$

1:  $\widehat{\rho^\ell} \sim \mathcal{N}(\widehat{\mu_\ell}, \widehat{\Sigma_\ell}) \leftarrow (\mathbf{g}_\#^\ell(\mathbb{P}_{\mathcal{M}^\ell}))^{-1}$  #*Abduction for $\mathcal{M}^\ell$*

2:  $\widehat{\rho^h} \sim \mathcal{N}(\widehat{\mu_h}, \widehat{\Sigma_h}) \leftarrow (\mathbf{g}_\#^h(\mathbb{P}_{\mathcal{M}^h}))^{-1}$  #*Abduction for $\mathcal{M}^h$*

3:  $\epsilon \leftarrow$ from Theorem 1  #*Compute robustness radius*

4:  $\mathbb{B}_{\epsilon,2}(\widehat{\boldsymbol{\rho}}) \leftarrow \mathbb{B}_{\varepsilon_\ell,2}(\widehat{\rho^\ell}) \times \mathbb{B}_{\varepsilon_h,2}(\widehat{\rho^h})$

5:  **Initialize:** $\mathrm{T}^{(0)} \in \mathbb{R}^{h \times \ell}, \boldsymbol{\rho}^{(0)} = \mathcal{N}(\mu^{\ell(0)}, \Sigma^{\ell(0)}) \otimes \mathcal{N}(\mu^{h(0)}, \Sigma^{h(0)}), \eta_\mathrm{T}, \eta_{\boldsymbol{\rho}}, k_{\min}, k_{\max}$

6:  **repeat**

7:  $\quad$ **for** $k_{\min}$ steps **do**

8:  $\quad\quad$ $\mathrm{T}^{(t+1)} \leftarrow \mathrm{T}^{(t)} - \eta_\mathrm{T} \nabla_\mathrm{T} F(\mathrm{T}^{(t)}, \rho^{(t)})$  #*Update abstraction T*

9:  $\quad$ **end for**

10:  $\quad$ **for** $k_{\max}$ steps **do**

11:  $\quad\quad$ $\mu^{\ell(t+1)} \leftarrow \mu^{\ell(t)} + \eta_{\boldsymbol{\rho}} \nabla_{\mu^\ell} F(\mu^\ell, \mathrm{T}^{(t+1)})$

12:  $\quad\quad$ $\mu^{h(t+1)} \leftarrow \mu^{h(t)} + \eta_{\boldsymbol{\rho}} \nabla_{\mu^h} F(\mu^h, \mathrm{T}^{(t+1)})$

13:  $\quad\quad$ $\Sigma^{\ell(t+\frac{1}{2})} \leftarrow \Sigma^{\ell(t)} + \eta_\rho \nabla_{\Sigma^\ell} \tilde{F}(\mathrm{T}^{(t+1)}, \rho^{(t)})$

14:  $\quad\quad$ $\Sigma^{\ell(t+1)} \leftarrow \mathrm{prox}_{\lambda_\ell}\left(\Sigma^{\ell(t+\frac{1}{2})}\right)$  #*Proximal step*

15:  $\quad\quad$ $\Sigma^{h(t+\frac{1}{2})} \leftarrow \Sigma^{h(t)} + \eta_\rho \nabla_{\Sigma^h} \tilde{F}(\mathrm{T}^{(t+1)}, \rho^{(t)})$

16:  $\quad\quad$ $\Sigma^{h(t+1)} \leftarrow \mathrm{prox}_{\lambda_h}\left(\Sigma^{h(t+\frac{1}{2})}\right)$  #*Proximal step*

17:  $\quad\quad$ $(\mu^{\ell(t+1)}, \Sigma^{\ell(t+1)}) \leftarrow \mathrm{proj}_{\mathcal{W}_2 \leqslant \varepsilon_\ell}(\mu^{\ell(t+1)}, \Sigma^{\ell(t+1)}, \widehat{\mu^\ell}, \widehat{\Sigma^\ell})$  #*Project to Gelbrich ball*

18:  $\quad\quad$ $(\mu^{h(t+1)}, \Sigma^{h(t+1)}) \leftarrow \mathrm{proj}_{\mathcal{W}_2 \leqslant \varepsilon_h}(\mu^{h(t+1)}, \Sigma^{h(t+1)}, \widehat{\mu^h}, \widehat{\Sigma^h})$  #*Project to Gelbrich ball*

19:  $\quad\quad$ $\boldsymbol{\rho}^{(t+1)} \leftarrow \mathcal{N}(\mu^{\ell(t+1)}, \Sigma^{\ell(t+1)}) \otimes \mathcal{N}(\mu^{h(t+1)}, \Sigma^{h(t+1)})$  #*Update joint environment $\boldsymbol{\rho}$*

20:  $\quad$ **end for**

21:  **until** convergence

$\quad$ **Return:** $\mathrm{T}^\star \in \mathbb{R}^{h \times \ell}, \boldsymbol{\rho}^\star = \rho^{\star\ell} \otimes \rho^{\star h}$

---

---

**Algorithm 4** Empirical DIROCA for linear models

---

**Require:** $\mathcal{S}^\ell, \mathcal{S}^h, \mathcal{I}^\ell, \mathcal{I}^h, \omega : \mathcal{I}^\ell \to \mathcal{I}^h$, samples from $\mathbb{P}_{\mathcal{M}_\iota^\ell}(\mathcal{X}^\ell), \mathbb{P}_{\mathcal{M}_{\omega(\iota)}^h}(\mathcal{X}^h)$

1: $\widehat{\rho^\ell} = \frac{1}{N_\ell} \sum_{i=1}^{N_\ell} \delta_{\widehat{u^\ell}_i} \leftarrow \mathbf{L}^{-1} \mathcal{X}^\ell$            *#Abduction for $\mathcal{M}^\ell$*

2: $\widehat{\rho^h} = \frac{1}{N_h} \sum_{i=1}^{N_h} \delta_{\widehat{u^h}_i} \leftarrow \mathbf{H}^{-1} \mathcal{X}^h$            *#Abduction for $\mathcal{M}^h$*

3: $\widehat{\boldsymbol{\rho}} \leftarrow \widehat{\rho^\ell} \otimes \widehat{\rho^h}$            *#Joint environment*

4: $\epsilon \leftarrow$ from Theorem 4.1            *#Robustness radius*

5: **Initialize:** $\mathrm{T}^{(0)}, \Theta_\ell^{(0)}, \Theta_h^{(0)}$

6: $\rho^{\ell(0)} = \frac{1}{N_\ell} \sum_{i=1}^{N_\ell} \delta_{\widehat{u}_i^\ell + \theta_i^{\ell(0)}}$

7: $\rho^{h(0)} = \frac{1}{N_h} \sum_{i=1}^{N_h} \delta_{\widehat{u}_i^h + \theta_i^{h(0)}}$

8: $\boldsymbol{\rho}^{(0)} \leftarrow \rho^{\ell(0)} \otimes \rho^{h(0)}$

9: **repeat**

10:      **for** $k_{\min}$ steps **do**

11:          $\mathrm{T}^{(t+1)} \leftarrow \mathrm{T}^{(t)} - \eta \, \nabla_{\mathrm{T}} F(\mathrm{T}^{(t)}, \Theta_\ell^{(t)}, \Theta_h^{(t)})$            *#Update abstraction* $\mathrm{T}$

12:      **end for**

13:      **for** $k_{\max}$ steps **do**

14:          $\Theta_\ell^{\text{temp}} \leftarrow \Theta_\ell^{(t)} + \eta \, \nabla_{\Theta_\ell} F(\mathrm{T}^{(t+1)}, \Theta_\ell^{(t)}, \Theta_h^{(t)})$

15:          $\Theta_\ell^{(t+1)} \leftarrow \text{proj}_{\|\cdot\|_F \leqslant \varepsilon_\ell \sqrt{N_\ell}}(\Theta_\ell^{\text{temp}})$            *#Project onto Frobenius norm ball*

16:          $\Theta_h^{\text{temp}} \leftarrow \Theta_h^{(t)} + \eta \, \nabla_{\Theta_h} F(\mathrm{T}^{(t+1)}, \Theta_\ell^{(t)}, \Theta_h^{(t)})$

17:          $\Theta_h^{(t+1)} \leftarrow \text{proj}_{\|\cdot\|_F \leqslant \varepsilon_h \sqrt{N_h}}(\Theta_h^{\text{temp}})$            *#Project onto Frobenius norm ball*

18:          $\rho^{\ell(t+1)} \leftarrow \frac{1}{N_\ell} \sum_{i=1}^{N_\ell} \delta_{\widehat{u}_i^\ell + \theta_i^{\ell(t+1)}}$

19:          $\rho^{h(t+1)} \leftarrow \frac{1}{N_h} \sum_{i=1}^{N_h} \delta_{\widehat{u}_i^h + \theta_i^{h(t+1)}}$

20:          $\boldsymbol{\rho}^{(t+1)} \leftarrow \rho^{\ell(t+1)} \otimes \rho^{h(t+1)}$            *#Update joint environment* $\boldsymbol{\rho}$

21:      **end for**

22: **until** convergence

     **Return:** $\mathrm{T}^\star \in \mathbb{R}^{h \times \ell}, \boldsymbol{\rho}^\star$

---

---

**Algorithm 5** General DIRoCA $[d \in \{\ell, h\}]$

---

**Require:** $\omega, \mathcal{I}^d, \mathcal{G}_{\mathcal{M}^d}, \{\mathbf{X}_\iota^d \in \mathbb{R}^{N \times d}\}_{\iota \in \mathcal{I}^d}, \varepsilon_d$ from Thm. 4.1.

**Require:** Hyperparameters: $\eta_\tau, \eta_\theta$ (learning rates), $N$ (batch size), $k_{\min}, k_{\max}$ (inner steps)

1: *// Step 1: Abduction and initialization*
2: $\mathbf{U}^d \leftarrow \text{Abduct}(\mathbf{X}^d, \mathcal{M}^d)$                  *#Abduct observational noise*
3: $\widehat{\rho^d} \leftarrow \frac{1}{N} \sum_{i=1}^N \delta_{u_i^d}$                  *#Define empirical marginals*
4: $\widehat{\boldsymbol{\rho}} \leftarrow \widehat{\rho^\ell} \otimes \widehat{\rho^h}$                  *#Joint environment*
5: *// Step 2: Define constraints*
6: $r_d \leftarrow \varepsilon_d \sqrt{N}$                  *#Frobenius budgets*
7: *// Step 3: Adversarial optimization*
8: **Initialize:** $\mathrm{T}^{(0)}, \Theta_\ell^{(0)} \leftarrow \mathbf{0}, \Theta_h^{(0)} \leftarrow \mathbf{0}$
9: $\rho^{d(0)} \leftarrow \frac{1}{N} \sum_{i=1}^N \delta_{u_i^d + \theta_i^{d(0)}}$
10: $\boldsymbol{\rho}^{(0)} \leftarrow \rho^{\ell(0)} \otimes \rho^{h(0)}$
11: **repeat**
12:    *// Minimization step (update abstraction)*
13:    **for** $k_{\min}$ steps **do**
14:       $\mathcal{J}(\mathrm{T}, \boldsymbol{\Theta}) \coloneqq \mathbb{E}_\iota \left[ \left\| \mathrm{T}(\mathbf{D}_\ell^{(\iota)} + \mathbf{U}^\ell + \Theta_\ell)^\top - (\mathbf{D}_h^{(\omega(\iota))} + \mathbf{U}^h + \Theta_h) \right\|_F^2 \right]$
15:       $\mathrm{T}^{(t+1)} \leftarrow \mathrm{T}^{(t)} - \eta_\tau \nabla_{\mathrm{T}} \mathcal{J}(\mathrm{T}^{(t)}, \boldsymbol{\Theta}^{(t)})$
16:    **end for**
17:    *// Maximization step (update joint environment)*
18:    **for** $k_{\max}$ steps **do**
19:       $\Theta_\ell^{\text{temp}} \leftarrow \Theta_\ell^{(t)} + \eta_\theta \nabla_{\Theta_\ell} \mathcal{J}(\mathrm{T}^{(t+1)}, \boldsymbol{\Theta}^{(t)})$
20:       $\Theta_h^{\text{temp}} \leftarrow \Theta_h^{(t)} + \eta_\theta \nabla_{\Theta_h} \mathcal{J}(\mathrm{T}^{(t+1)}, \boldsymbol{\Theta}^{(t)})$
21:       $\Theta_\ell^{(t+1)} \leftarrow \text{Proj}_{\|\cdot\|_F \leqslant r_\ell}(\Theta_\ell^{\text{temp}})$
22:       $\Theta_h^{(t+1)} \leftarrow \text{Proj}_{\|\cdot\|_F \leqslant r_h}(\Theta_h^{\text{temp}})$        *#Project onto ambiguity set*
23:       $\rho^{\ell(t+1)} \leftarrow \frac{1}{N} \sum_{i=1}^N \delta_{u_i^\ell + \theta_i^{\ell(t+1)}}$
24:       $\rho^{h(t+1)} \leftarrow \frac{1}{N} \sum_{i=1}^N \delta_{u_i^h + \theta_i^{h(t+1)}}$        *#Update marginals*
25:       $\boldsymbol{\rho}^{(t+1)} \leftarrow \rho^{\ell(t+1)} \otimes \rho^{h(t+1)}$        *#Update joint environment*
26:    **end for**
27: **until** convergence
   **Return:** $\mathrm{T}^\star \in \mathbb{R}^{h \times \ell}, \boldsymbol{\rho}^\star$

---

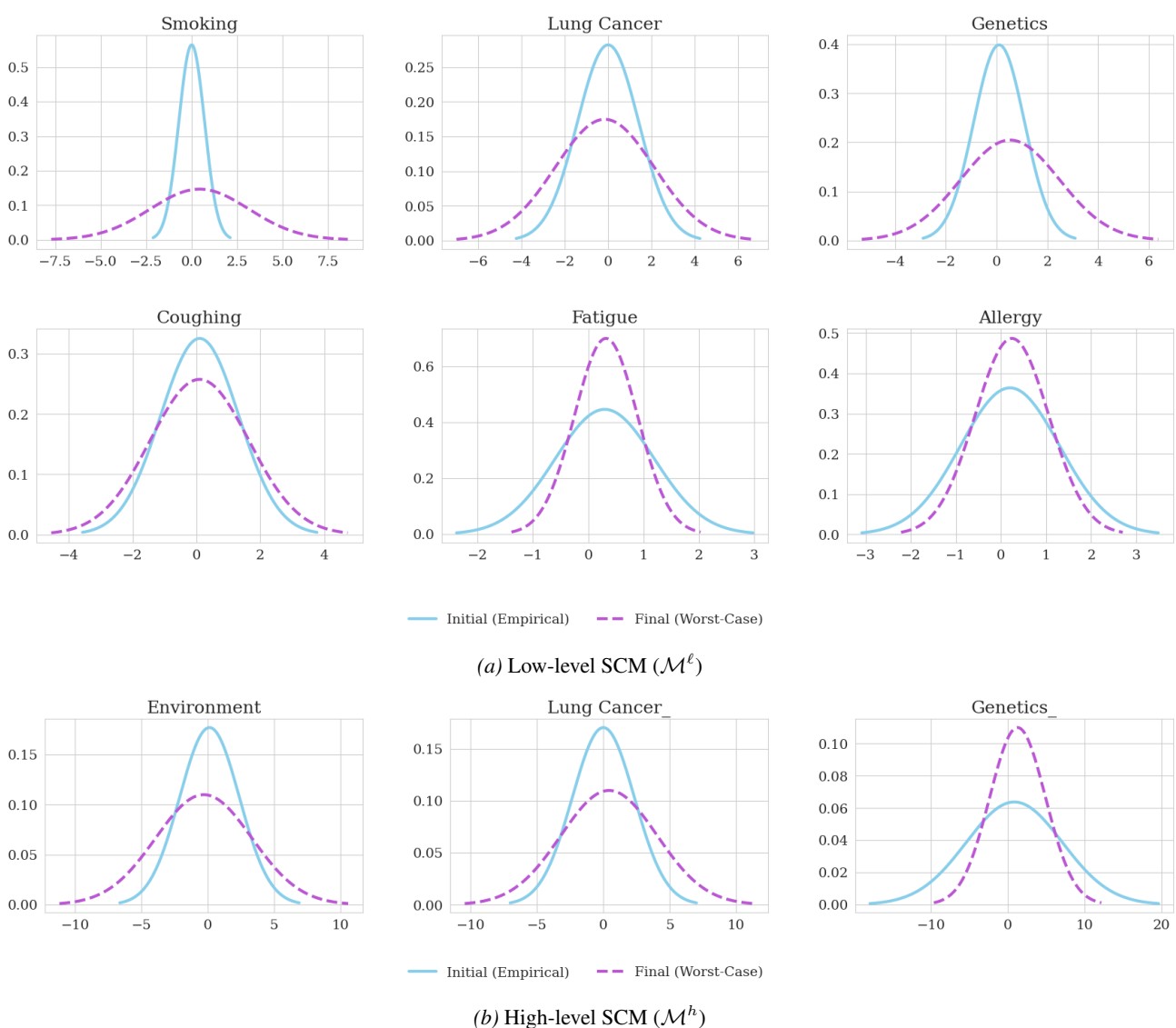

*(a)* Low-level SCM ($\mathcal{M}^{\ell}$)

*(b)* High-level SCM ($\mathcal{M}^{h}$)

*Figure 29.* Marginal noise distributions for each variable in the low-level (top) and high-level (bottom) SCMs of the `LiLUCAS` dataset. In each subplot, the solid blue curve represents the initial empirical noise distribution, and the dashed purple curve shows the corresponding worst-case distribution obtained via the distributionally robust optimization.

*Table 17.* Notation Table.

| Symbol | Definition | Description |
|---|---|---|
| **Structural causal models** | | |
| $\mathcal{M}^d$ | $(\mathcal{S}^d, \rho^d)$ | A $d$-dimensional SCM with deterministic causal basis $\mathcal{S}^d$ and environment $\rho^d$. |
| $\mathcal{S}^d$ | $\langle \mathcal{X}, \mathcal{U}, \mathcal{F} \rangle$ | Deterministic causal basis of an SCM: endogenous variables, exogenous variables, and structural functions. |
| $\mathcal{X}$ | $\{X_i\}_{i=1}^d$ | Endogenous variables of an SCM. |
| $\mathcal{U}$ | $\{U_i\}_{i=1}^d$ | Exogenous noise variables of an SCM. |
| $\mathcal{F}$ | $\{f_i\}_{i=1}^d$ | Structural functions defining $X_i = f_i(\mathrm{PA}(X_i), U_i)$. |
| $\rho^d$ | $\prod_{i=1}^d \mathbb{P}(U_i)$ | Environment, i.e., joint distribution over exogenous variables. Under Markovianity the exogenous variables are independent. |
| $\mathcal{G}_{\mathcal{M}^d}$ | – | DAG induced by the structural functions of $\mathcal{M}^d$. |
| $\mathbf{g}$ | $\mathrm{dom}[\mathcal{U}] \to \mathrm{dom}[\mathcal{X}]$ | Mixing function or reduced-form map obtained by recursively composing the structural functions. |
| $\mathbb{P}_{\mathcal{M}^d}(\mathcal{X})$ | $\mathbf{g}_\#(\rho^d)$ | Observational endogenous distribution induced by pushing the environment through the reduced form. |
| $\mathcal{D}$ | $(f_i(\mathrm{PA}(X_i)))_{i=1}^d$ | Deterministic component in an additive noise model. |
| $\mathcal{U}$ | $(U_i)_{i=1}^d$ | Stochastic residual/noise component in an additive noise model. |
| $\mathbf{B}$ | – | Weighted adjacency matrix of a linear additive noise model. |
| $\mathbf{M}$ | $(I - \mathbf{B}^\top)^{-1}$ | Linear mixing matrix for a linear additive noise model. |
| **Low- and high-level models** | | |
| $\mathcal{M}^\ell$ | $(\mathcal{S}^\ell, \rho^\ell)$ | Low-level SCM, with dimension $\ell$. |
| $\mathcal{M}^h$ | $(\mathcal{S}^h, \rho^h)$ | High-level SCM, with dimension $h \leqslant \ell$. |
| $\boldsymbol{\rho}$ | $\rho^\ell \otimes \rho^h$ | Joint product environment over low- and high-level exogenous variables. |
| $\boldsymbol{\mathcal{U}}$ | $\mathrm{dom}[\mathcal{U}^\ell] \times \mathrm{dom}[\mathcal{U}^h]$ | Product exogenous domain. |
| **Interventions and abstraction context** | | |
| $\iota$ | $\mathrm{do}(\mathcal{A} = \mathbf{a})$ | Exact intervention fixing variables $\mathcal{A}$ to values $\mathbf{a}$. |
| $\mathcal{M}_\iota^d$ | – | Post-interventional SCM obtained after applying intervention $\iota$. |
| $\mathcal{I}^\ell, \mathcal{I}^h$ | – | Low- and high-level intervention sets. |
| $\omega$ | $\mathcal{I}^\ell \to \mathcal{I}^h$ | Surjective, order-preserving intervention map. |
| $\mathcal{A}^\ell, \mathcal{A}^h$ | $\subseteq \mathcal{P}(\mathcal{U}^\ell), \mathcal{P}(\mathcal{U}^h)$ | Relevant low- and high-level environment sets. |
| $\mathcal{A}$ | $\mathcal{A}^\ell \otimes \mathcal{A}^h$ | Relevant joint environment space. |
| $(\mathcal{A}, \mathcal{I})$ | – | Abstraction context specifying the environments and interventions over which consistency is evaluated. |
| **Causal abstraction quantities** | | |
| $\tau$ | $\mathrm{dom}[\mathcal{X}^\ell] \to \mathrm{dom}[\mathcal{X}^h]$ | Abstraction map from low-level to high-level endogenous variables. |
| $\mathrm{T}$ | $\in \mathbb{R}^{h \times \ell}$ | Matrix representation of a linear abstraction map, so that $\tau(x) = \mathrm{T}\,x$. |
| $e_\tau(\mathcal{M}^\ell, \mathcal{M}^h)$ | Eq. (5) | Total abstraction error aggregated over environments and interventions. |
| **Distributional robustness** | | |
| $\mathbb{B}_{\varepsilon_d, 2}(\widehat{\rho^d})$ | $\{\rho \in \mathcal{P}(\mathcal{U}^d) : W_2(\rho, \widehat{\rho^d}) \leqslant \varepsilon_d\}$ | Marginal 2-Wasserstein ambiguity set for level $d \in \{\ell, h\}$. |
| $\mathbb{B}_{\varepsilon, 2}(\widehat{\boldsymbol{\rho}})$ | $\mathbb{B}_{\varepsilon_\ell, 2}(\widehat{\rho^\ell}) \times \mathbb{B}_{\varepsilon_h, 2}(\widehat{\rho^h})$ | Product ambiguity set over low- and high-level environments. |
| $\boldsymbol{\rho}^\star$ | $\rho^{\ell\star} \otimes \rho^{h\star}$ | Worst-case joint environment selected by the adversary inside the ambiguity set. |
| $\widehat{\epsilon}$ | – | Default radius instantiated from the concentration result after manually fixing the unknown distribution-dependent constants. |
| $\epsilon^\star$ | – | Best-performing radius selected post hoc from a pre-specified grid for reporting; not a theoretically optimal radius. |

