# OpenReview forum: "Distributionally Robust Causal Abstractions"
_ICML.cc/2026/Conference — ICML 2026 regular_

### Official Review · Reviewer_UqrA · 2026-03-13

**Soundness:** 2
**Presentation:** 3
**Significance:** 2
**Originality:** 2
**Overall Recommendation:** 4
**Confidence:** 3

**Summary:**

The paper studies causal abstraction learning under distribution shift where the abstraction maps remain consistent across environments. A new robustness causal abstraction concept is proposed which stand in the middle ground between exact abstractions that hold in one fixed environment and uniform abstractions that hold across environments. To solve the problem, the paper proposes a method called DIROCA which uses Distributionally Robust Optimization (DRO) from (Kuhn et al., 2019) to optimize across environment distribution shifts. Specifically it uses a min-max optimization of an abstraction error defined as standard abstraction error aggregated across environtments and the distribution shift is sampled from a Wasserstein ambiguity sets given some robustness radius. For the experiments, the method is benchmarked on synthetic, semi-synthetic, and real settings, and the paper reports improved robustness to environmental shifts as well as to some structural and intervention misspecifications. Overall, the paper’s idea is that the causal abstraction can be learned robustly by modeling uncertainty over environments rather than fixing a single one.

**Compliance With Llm Reviewing Policy:**

Affirmed.

**Final Justification:**

My concerns are resolved, I'll raise my score from 3 to 4.

**Key Questions For Authors:**

- How do you position the paper relative to prior work that already studies abstraction learning across multiple interventional distributions, multiple environments, soft interventions, shift interventions, inaccessible SCMs, and lossy abstractions?
- The experiments need to include some important recent causal abstraction baselines like in the references above.
- The method depends on abduction to recover exogenous residuals. Since there may be errors in recovering exact SCM functions, how do we guarantee the robustness?

**Limitations:**

yes

**Strengths And Weaknesses:**

Strengths:

The paper is technically clear and coherent. The use of the ambiguity set over the product of low and high level exogenous environments is beneficial. The formulation to turn the abstraction learning into a min-max problem over intervention mismatch under Wasserstein perturbations is reasonable. I think the paper is promissing in how it connects causal abstraction with DRO learning in a way that is easy to interpret. The robustness bounds and the discussion of ambiguity radius selection give some theoretical guarantee. Overall, it has a reasonable formulation and some promising empirical results that environment robustification can help under distribution shift and misspecification.

Weaknesses:

However, the paper is not the first distributionally robust CA framework but is more a reframing of a distribution shift problem and application of Wasserstein-DRO. Therefore, I do not think that broader story is really justified. Earlier work had already moved away from the hard interventions setting. For example, the study in [1] already learns abstractions across multiple interventional distributions. While, the study in [2] learns abstraction maps from observational and interventional data, explicitly motivated by multiple experimental environments, using multi-marginal OT with causal constraints. In [3], it already generalizes abstraction theory from hard to soft interventions. In [4], it studies the much harder setting where SCMs are inaccessible, interventions are unavailable, and samples are misaligned, or even lossy abstractions [5]. So, although earlier work did not already do this exogenous-Wasserstein robustness, they are solving similar problems in robustness or misspecification. So, I think the novelty is one among robustness work under distribution shift.

[1] Zennaro, Fabio Massimo, et al. "Jointly learning consistent causal abstractions over multiple interventional distributions." Conference on Causal Learning and Reasoning. PMLR, 2023.
[2] Felekis, Yorgos, et al. "Causal optimal transport of abstractions." Causal Learning and Reasoning. PMLR, 2024.
[3] Massidda, Riccardo, et al. "Causal abstraction with soft interventions." Conference on Causal Learning and Reasoning. PMLR, 2023.
[4] D'Acunto, Gabriele, et al. "Causal abstraction learning based on the semantic embedding principle." arXiv preprint arXiv:2502.00407 (2025).
[5] Xia, Kevin, and Elias Bareinboim. "Causal abstraction inference under lossy representations." arXiv preprint arXiv:2509.21607 (2025).

---

> ### Author Rebuttal · Authors · 2026-03-30
>
> We thank the reviewer for the constructive feedback and for recognizing the technical coherence of the formulation, the usefulness of the product ambiguity set, and the importance of connecting CA with DRO.
>
> W1/Q1/Q2: Our core contribution is twofold: *(a) we introduce the first class of robust CAs, namely $(\rho, \iota)$-abstractions, which make abstraction consistency explicitly depend on a constrained set of exogenous distributions and (b) we propose the first learning framework, DiRoCA, for learning CAs robust to exogenous distribution shifts and misspecification*. **None of the cited works makes environmental robustness part of the abstraction definition itself, nor learns abstractions against uncertainty in the joint exogenous environment of the SCMs.**
>
> **We do not ignore the works**. Figure 1 already positions [1,2,4] as prior CAL methods learning under a fixed nominal environment, whereas our contribution introduces environmental uncertainty through a Wasserstein ambiguity set around that environment. Multiple interventional distributions are not the same as uncertainty in exogenous distributions. An abstraction can be interventionally consistent under the training environment and still fail completely when the exogenous distribution shifts, even if the intervention structure is unchanged. This is precisely the failure mode DiRoCA addresses, and one that training on many interventions under a fixed environment cannot prevent. Also, these works do not match our supervision regime, abstraction class, and evaluation target. The closest applicable baselines in our regime are existing CAL methods assuming accessible SCMs, aligned interventions, and linear abstractions, which we already compare against [6]. Concretely:
>
> [1]learns across multiple interventional distributions via differentiable programming, but abstraction consistency remains intervention-indexed under a fixed underlying exogenous environment. It does not formulate or solve the problem of robust abstractions to exogenous shifts.
>
> [2]learns a stochastic abstraction from observational/interventional data via multi-marginal OT under different supervision assumptions and a different learning objective. It does not optimize worst-case abstraction error over an ambiguity set of exogenous environments. We only draw inspiration from its barycentric component as a non-robust comparison point.
>
> [3]broadens CA theory from hard to soft interventions, i.e. intervention semantics, but does not study robustness over exogenous environments. It is relevant related work, but not an experimental learning baseline for our setting.
>
> [4]studies CAL in a substantially different weaker-information regime (inaccessible SCMs, unavailable interventions, misaligned samples) under a different CA framework. Their method searches over a different abstraction class (SEP / Stiefel manifold), and optimizes a different objective from robust interventional abstraction under environment shift.
>
> [5]studies lossy abstractions and the identifiability of high-level causal queries, not robustness of abstractions across exogenous distributions. It addresses a different problem and is therefore not a comparison baseline for our setting.
>
> We therefore respectfully disagree with the claim that the paper is: *“one among robustness work under distribution shift.”* The cited papers address important adjacent directions, but **they do not formulate or solve the problem of learning robust CAs to exogenous distributions shifts**. This is the precise novelty of our work.
>
>
>
> -*“earlier work had already moved away from the hard interventions setting”*
>
> we think this conflates two orthogonal axes. Hard vs. soft interventions [3] concerns the semantics of interventions; robust vs. non-robust abstraction concerns the set of exogenous environments over which consistency must hold. Our contribution is entirely along the latter axis. Our setting remains in the standard hard-intervention case yet studies robustness across shifts in the exogenous distribution.
>
> We will make all of these distinctions explicit in the revised related-work and experimental-positioning discussion.
>
> Q3: We want to thank the reviewer for raising this point. Our guarantees are not against arbitrary SCMs or abduction error. When structural functions are known, abduction is exact. However, in practice, when estimating, the recovered environment is only approximate and depends on the estimation quality of the chosen model (see Remark 2 and App. A.6.). Thus, the introduced ambiguity set provides robustness guarantees assuming the SCM is not misspecified. Addressing this type of misspecification remains an important direction for future work. We will make this conditional nature of the guarantee explicit in the revised main text and limitations.
>
> We hope these clarifications address the concerns raised and would appreciate a reconsideration of the score.
>
> [6] Massidda, R., et al. Learning Causal Abstractions of Linear SCMs

---

> > ### Author Rebuttal · Reviewer_UqrA · 2026-04-02
> >
> > My concerns are resolved, I'll raise the score from 3 to 4.

---

### Official Review · Reviewer_MPjF · 2026-03-13

**Soundness:** 2
**Presentation:** 3
**Significance:** 2
**Originality:** 3
**Overall Recommendation:** 4
**Confidence:** 2

**Summary:**

This paper introduces distributionally robust causal abstractions for Causal Abstraction Learning. It can find the robust  abstraction under environmental shifts.

**Compliance With Llm Reviewing Policy:**

Affirmed.

**Final Justification:**

After considering the authors’ rebuttal, I have decided to raise my score from 3 to 4.

**Key Questions For Authors:**

See weaknesses.

**Limitations:**

yes

**Strengths And Weaknesses:**

Strengths:

1. Considering environmental shift is a good idea for causal abstraction learning.

2. It gives thoretical guarantees and worist-case abstraction error.

Weaknesses:

1. Line 337-338 you define alpha as prevalence of shifted samples, but it seems is the scale of the noise. Combine line 348 X^hat=X+N, and line 353 X^hat=(1-alpha)X+alpha X^hat, we can get X^hat =X+alpha N.


2. Line 314 you mention you report the best-performing for DIROCA in each setting. Could you add more details here like how you do the hyperparameter tunning and choose epsilon? It is chosen based on the validation set or test robustness? Seems GRAD's result is not from hyperparameter tunning so the comparison may be not very fair.

---

> ### Author Rebuttal · Authors · 2026-03-30
>
> We thank the reviewer for recognizing the value of considering environmental shifts in causal abstraction learning and the importance of theoretical guarantees on worst-case abstraction error. We address both concerns raised below.
>
> W1: We thank the reviewer for pointing this out. The convex-combination in lines 348 and 353 was a typo for $\bar{X_i} = (1 - \mathbf{1_{i\in S}}) X_i + \mathbf{1_{i\in S}} \tilde{X}_i$, where $|S| = \lfloor \alpha n \rfloor$ and $S$ is drawn uniformly without replacement from the test indices; i.e., an $\alpha$-fraction of rows is selected uniformly without replacement and replaced by their noisy counterparts $\tilde{X}_i = X_i + n_i$, while the remaining rows are left unchanged. Under this protocol, $\alpha \in [0,1]$ indeed represents the contamination prevalence, going from a fully clean test set at $\alpha = 0$ to a fully shifted one at $\alpha = 1$, while $\tilde{\sigma}$ independently controls the magnitude of the perturbation applied to each contaminated sample. All reported results were produced using this replacement-based implementation. We will correct the typo in the revision to accurately reflect the actual protocol.
>
> W2: In the main text, $DiRoCA_{\epsilon^\star}$ denotes the best-performing radius configuration for a given benchmark setting, i.e. the one with the lowest mean abstraction error. Concretely, we pre-specify a grid of radius values, train DiRoCA separately for each radius under the same 5-fold cross-validation protocol, and evaluate each learned abstraction on the corresponding held-out fold; the contamination shift is applied only at evaluation time on that held-out set, and $\epsilon^\star$ is then reported as the radius with the lowest mean abstraction error across those held-out evaluations and not selected on a separate validation split. More details alongside the full results across all tested radius values are already reported in Appendix A.8.; we will revise the paper to make this fully explicit. Regarding the fairness of the comparison with GRAD$_{(\tau,\omega)}$: this baseline corresponds exactly to the zero-radius special case of our framework, i.e.$\epsilon=0$, as noted in Remark 1 and in the baselines description. Thus, it does not have an analogous robustness radius to tune; it is, by construction, the nominal-environment, non-robust baseline. Its degradation under increasing contamination is precisely the empirical phenomenon that motivates our paper: non-robust abstractions can fit the nominal environment well but deteriorate substantially under distributional shift. We commit to making this more explicit in the revision.
>
> We hope the responses above resolve both concerns. The typo has been acknowledged and will be fixed, and the experimental protocol will be further explained. We would appreciate it if the reviewer could consider revising the score in light of these clarifications.

---

> > ### Author Rebuttal · Reviewer_MPjF · 2026-04-01
> >
> > Thanks for the rebuttal. It addresses my concern. I will raise the score from 3 to 4.

---

### Official Review · Reviewer_nmGr · 2026-03-16

**Soundness:** 3
**Presentation:** 2
**Significance:** 3
**Originality:** 3
**Overall Recommendation:** 4
**Confidence:** 3

**Summary:**

The article introduces a robust variant of causal abstraction, called (ρ,ι)-abstractions and a learning framework called DIROCA that replaces the fixed-environment assumption (or more general consistency in all worlds)  with focusing on relevant environments in prior causal abstraction learning with Wasserstein ambiguity sets over exogenous variables. Main claims consist of a new robustness notion which is between exact and uniform abstractions, optimization procedure (inspored from Robsut optimisation) for empirical and Gaussian settings, concentration bounds for choosing ambiguity radius, and experiments on various dataset showing improved robustness under shifts, corruptions/misspecifications.

**Compliance With Llm Reviewing Policy:**

Affirmed.

**Final Justification:**

Overall, the work definitely has merits, and authors provided clarifying answers to my questions about the technicalities of the work.  The work needed improvement in its accessibility to be published without a doubt, and the authors promises to take these suggestions into account and reorganise the paper, which I cannot entirely verify. Therefore, reflecting these, I maintain my "slightly positive but not very high" score.

**Key Questions For Authors:**

Q1: In Figure 3,it is not possible to see all colors on the left side plots e.g.,  AbsLinp and n, and DiRoCa 4,4 are missing.

Q2: In your optimisation process, if you were not reusing the observational residuals, but fresh ones (disregarding the computational cost), would you still expect similar accuracy to your current results?  How?

Q3: Any comment W2 and W3? (For the latter,  do we have access to good constants and confidence levels?? If not, what would be a principled or effective approach to have them?)

**Limitations:**

Limitations are only partially articulated. It comes mostly in conclusion very briefly which can easily escape readers eye: that is the current framework is limited to linear abstractions, and assumes it has access to interventions adn true causal DAGs (or a good estimation of it). Another is that: the trick of re-using the base residuls across interventions  for efficient Frobenius based evaluation is actually, to me,  a limitation on the training setup, but told as a technical choice.

**Strengths And Weaknesses:**

Strengths:

S1: The paper takcles an important problem:namely causal abstractions that are learned in one environment is brittle under shifts or misspecifications.  It is topical: We definitely need more work on that line.

S2: The approach is novel and elegant which is approaching causal abstraction learning as a distributionally robust optimisation problem over reduced environments.

S3: Results are both theoretical and empirical. To me, the main strength is in theoretical depth, but it also carries a good mixture of experiments for justification of the theory.

Weaknesses:

W1: The paper's exposition is not in its best state, and make it difficult to follow, as it is very compressed, it becomes a guess game easily. (I had to check appendix many times, and also internet back and forth just to understand the conceptual derivations and notations. This also Includes some plots (see the Q1. ))

W2: Overall the paper, there is a big difference between the tuned/best configuration DiroCa and the, say, theorem-based default one. Take simply table 1, cMnist. This is a huge difference, as it seems the best radius comes from the manual choice rather than the theorem which triggers that the merit is rather on the best configuration/tuneing rather than the theoretical innovation.

W3: Theorem 4.1 gives an existential concentration bound. But I am not truly sure, how much of a real practical recipe it does provide. (Similar to above, in Table 1 see the tuned radiusses are quite quite different than the theorem based ones.)

---

> ### Author Rebuttal · Authors · 2026-03-30
>
> We thank the reviewer for the detailed feedback and for recognizing the importance of the problem addressed, the novelty of the DRO framing, as well as both the depth and strength of the theoretical contributions and experimental analysis. We address each concern below.
>
> W1: We commit to utilizing the extra page to improve the flow and readability of the main text by: a) making the algorithmic pipeline more explicit earlier in Section 4; b) including a clearer step-by-step description of the building blocks of the method and c) including a notation table in the Appendix.
>
> W2/W3/Q3:We agree that the current wording should better distinguish between the theoretical importance of Theorem 4.1 and the practical use of the radius $\hat{\epsilon}$. Thm. 4.1 is a concentration result specialized to the CA setting: it extends single-distribution Wasserstein concentration results to the joint product environment of the low- and high-level SCMs and shows that the ambiguity set contains the true joint environment with high probability while identifying the correct dependence of the radius on sample size, dimension, confidence level, and distribution-dependent constants. This is the main theoretical value of the result. At the same time, as in the underlying Wasserstein concentration results of [2.1, Thm. 18,21], the confidence level $1-\delta$ is user-specified and allocated across environments, and the constants $c_{d,1},c_{d,2}$ are existential and distribution-dependent (e.g., tail- and dimension-related) and unavailable in closed form in practice. Hence, Thm. 4.1 does not provide a parameter-free numerical recipe for selecting a practically optimal radius. In our experiments, $\hat\epsilon$ is instantiated by fixing simple default values for the constants to empirically assess how such a radius behaves in practice. Further, we apologize for any notational confusion: $\epsilon^\star$ denotes the best-performing radius on a given benchmark, a presentation choice, not a theoretically optimal quantity. In the DRO literature, theory typically fixes the interpretation and scale of the ambiguity radius, while its practical numerical value is calibrated empirically, e.g. via holdout or cross-validation [2.2, 2.3]. We will connect our radius selection practice to the broader DRO literature and revise the notation and surrounding discussion accordingly, describe $\hat{\epsilon}$ more accurately, and make explicit that our setting does not claim any theoretically backed optimality of the radii presented. Practical guidance for radius selection will be provided in the Appendix.
>
> Q1: The AbsLin baselines are not applicable in the Gaussian setting (first two subfigures) because the original method is not formulated under a parametric Gaussian environment model. Hence, the empirical setting is the appropriate comparison regime for them. Regarding the absence of $DiRoCA_{4,4}$, this is numerically identical to $DiRoCA_{8,8}$ in this setting, and its curve is hidden beneath it. We will make this explicit in the revised figure caption.
>
> Q2: DiRoCA is trained using a single shared residual base $U$ across interventions, solving for a common worst-case perturbation matrix $\Theta$. At test time, we use independent per-intervention residuals, which represent the realistic evaluation setting and is strictly harder than evaluating on the same shared residuals seen during training. A formulation that trains on intervention-specific perturbations is possible in principle through optimizing for a vector of worst-case perturbations, but it would solve a different optimization objective and would be easier, as training and evaluation distributions would be aligned. To examine this, Appendix A.6 includes a dedicated study on nLUCAS comparing: (i) a Consistency protocol, evaluating on the shared residual base, and (ii) a Generalization protocol with fresh independent residuals. Although the non-robust baseline benefits from the Consistency protocol in the clean setting, this advantage disappears under Generalization, whereas DiRoCA remains strong, suggesting that the shared-residual formulation does not limit the quality of the learned map. We will move an example of this comparison into the main text using the extra response page.
>
> We hope the responses clarify your concerns and would appreciate an upward revision of the score.
>
> [2.1]Kuhn, D., et al. Wasserstein Distributionally Robust Optimization: Theory and Applications in Machine Learning; [2.2]Esfahani, P.M.,Data-driven DRO using the Wasserstein Metric;[2.3]Aolaritei, L., Wasserstein Distributionally Robust Estimation in High Dimensions

---

> > ### Author Rebuttal · Reviewer_nmGr · 2026-04-04
> >
> > Thank you for your diligent and helpful responses; I am satisfied with your answers.  The only reservation I have is about the accessibility of the paper, and the lack of any example (the otherwise would strongly improve the paper). To reflect this,  I will maintain my positive but not very strong score for the time being. To decide between adjusting it higher or not, I need to discuss with other reviewers, and have more reflection.  Thank you once again!

---

> > > ### Author Response · Authors · 2026-04-05
> > >
> > > Thank you for your reply. We are glad that we were able to address your concerns. We highlight once more the concrete improvements we have already made in the revised version of our paper, in case this is helpful for your discussions:
> > >
> > > We have sharpened the main-text presentation by (a) making the pipeline clearer in Section 4, (b) bringing key material from the Appendix into the main text, using the extra page, (c) adding a notation table with descriptions and definitions for all key mathematical quantities, and (d) using one of the empirical causal abstraction examples in the paper to motivate the framework and provide intuition. The paper already provides broad empirical coverage through 5 causal abstraction examples, including 3 synthetic and 2 real-world datasets, together with multiple misspecification studies and extensive robustness results across contamination levels, noise magnitudes, and distributions. We anchor the exposition around one of these examples, introducing it early and revisiting it throughout the paper as a running example.
> > >
> > > Thank you again for your constructive engagement and feedback.

---

### Official Review · Reviewer_kLXJ · 2026-03-18

**Soundness:** 3
**Presentation:** 1
**Significance:** 3
**Originality:** 3
**Overall Recommendation:** 4
**Confidence:** 2

**Summary:**

**Below I provide the summary of this paper:** \
The proposed method relaxes the limitation of existing causal abstraction learning methods, which consider a fixed exogenous distribution, which is a major assumption. In practice, the correct exogenous distribution of causal models might not be recovered properly due to inaccuracy in the model or distribution shifts. The authors address this issue by introducing distributionally robust optimization to minimize the worse case error in causal abstraction learning. Finally, they provide experimental results on variaous datasets.

**Compliance With Llm Reviewing Policy:**

Affirmed.

**Final Justification:**

My concerns are addressed after the rebuttal. The theories and the algorithmic approach discussed in this paper appear correct to me. Thus I keep positive score.

**Key Questions For Authors:**

**Below I provide the questions of this paper:**
1. How does the intervention set play a role in CAL? How does the learning depend on the number of interventions? If the datasets are all observational, what challenges does the algorithm face? Most real-world datasets are observational, how applicable is the proposed algorithm in such cases?
2. What would be the practical applications or real-world scenarios where distributionally robust causal abstractions are impactful?

**Limitations:**

Yes.

**Strengths And Weaknesses:**

**Below I provide the strength of this paper:** \
The theories and the algorithmic approach discussed in this paper appear correct and solid to me. The idea to introduce DRO in causal abstraction is quite novel. Also, the authors show experimental results on diverse datasets and setups.

**Below I provide the weaknesses of this paper:** \
1. The paper is a little hard to read. The authors should provide some intuitive examples to present their approaches. For example, what would be a real-world example of an exact abstraction and a uniform abstraction.
2. Main algorithmic approach is obscured due to dense theoretical formalization. The authors should clearly present their full algorithm in the main paper. For ex: section A.6 can be moved to the main paper to provide the overall picture.
3. The abduction and the estimation step are not defined or discussed properly in the main paper. These building blocks should be discussed intuitively for a general audience.
4. Assumption 1 and Theorem 4.1 should be discussed in more detail. How assumption 1 is playing a role in theorem 4.1 is not clear.
5. Although the authors provide experimental results on one real-world dataset Electric Battery Manufacturing (EBM) in Table 1, the experiment setup is not clear. What are the two causal model learning? From which datasets? What does a high-level and low-level SCM mean for such a dataset? These questions should be answered more clearly.

I wish my comments become useful to the authors.

---

> ### Author Rebuttal · Authors · 2026-03-30
>
> We thank the reviewer for the positive assessment and for recognizing the solidity of the theory and algorithm, as well as the novelty of introducing DRO into CAL. We address the concerns raised below.
>
> W1/W2/W3:In the revision we will improve the presentation by: a) using the extra page to move key material from the Appendix to the main text and include a notation table; b) restructuring Sec. 4 to present the algorithmic pipeline more clearly; and c) expanding the main-text explanation of its building blocks. Full details are already in App. A.9 (algorithm) and A.5,A.6 (abduction/estimation). We discuss the relationship between CA frameworks in App. A.1 and visualize it in Fig. 1, but we will add a concrete example: In climate modeling, a low-level SCM models fine-grained atmospheric variables (wind speed, humidity), while a high-level SCM models regional-level quantities (temperature). The scientific goal is to learn a consistent abstraction between these two levels. The difficulty is that the exogenous noise driving the system (e.g. natural variability patterns) may shift across decades or regions, causing an abstraction learned in one regime to fail at deployment. An exact abstraction, learned, for example, from one decade of data may therefore collapse under such a shift. A uniform abstraction, requiring consistency accross every conceivable atmospheric regime, is physically unrealizable since one cannot collect data from infinitely many climate conditions. A $(\rho, \iota)$-abstraction instead requires consistency over a realistic range of conditions a practitioner might actually encounter around the empirical environment; precisely the DiRoCA setting.
>
> W4:Theorem 4.1 is a finite-sample concentration result that guarantees the true joint environment lies within a Wasserstein ball around the empirical one with high probability. Assumption 1 is precisely what enables it to work. It requires each marginal environment to be light-tailed, a standard condition in Wasserstein DRO [1.1], holding trivially under compact support. The light-tail condition is therefore not an arbitrary technical requirement but a necessary condition to apply the Wasserstein concentration inequalities of [1.1] (Thm. 18 and 21) to each individual measure to derive a joint coverage result for the CA setting. We will make this connection explicit in the revision.
>
> W5:EBM [1.2] is a published dataset used previously in the CA literature [1.2, 1.3], and it is described in detail in App. A.7.3. It consists of two labs (WMG and LRCS) observing the same battery manufacturing process at different spatial granularities: the low-level SCM models a process control parameter, Comma Gap (CG), causing material deposit measurements (Mass Loading) at two locations (ML1, ML2), while the high-level SCM aggregates these into a single scalar (ML). The abstraction map must therefore remain consistent under real interventions on CG. Beyond EBM, we also use cMNIST, built from real digits and used in prior CA work [1.4, 1.5]. We will add concise main-text descriptions and link them to prior usage.
>
> Q1:The intervention set restricts attention to interventions that can be meaningfully abstracted, either because of scientific interest or feasibility. Its structure and implications for learned abstractions were analyzed in [1.3] via optimal transport cost. The number of interventions directly affects the constraints of the abstraction map: the larger the set, the more constrained the abstraction map becomes, as it must align more intervention-induced distributions simultaneously. If only observational data is available, the algorithm remains well-posed but guarantees only observational alignment. In such cases, partial interventional supervision may still come from additional assumptions or identification strategies (e.g., natural experiments, do-calculus) if the goal is indeed an interventionally consistent map. We will include a relevant discussion in the revised version to make this clear.
>
> Q2:DiRoCA addresses a critical challenge in CAL: environments shift across regions, time periods, and experimental setups. As in the climate example (W1), abstractions learned under one environment may fail under another. DiRoCA treats the empirical environment as an uncertain estimate and optimizes the abstraction to remain valid across a principled neighborhood, which can be particularly impactful in any multi-scale real-world CA setting where the environment is uncertain or misspecified. We will make this practical motivation more explicit.
>
> We hope these clarifications address your concerns and would appreciate an upward revision of your score and confidence.
>
> [1.1]Kuhn, D., et al. Wasserstein DRO [1.2]Zennaro, F.M., et al. Jointly Learning Consistent Causal Abstractions [1.3]Felekis, Y., et al. Causal Optimal Transport of Abstractions [1.4]Xia, K., et al. Neural Causal Abstractions [1.5]Xia, K., et al. Causal abstraction inference under lossy representations

---

> > ### Author Rebuttal · Reviewer_kLXJ · 2026-04-02
> >
> > My concerns are addressed. Thus I keep positive score.

---

> > > ### Author Response · Authors · 2026-04-02
> > >
> > > We are glad that your concerns have been fully addressed. Given the overall review context, we would be very grateful if you would consider raising the score in your final assessment.

---

### Decision · Program_Chairs · 2026-04-30

**Decision:**

Accept (regular)

**Comment:**

The reviewers agreed that the topic of reliable causal abstractions under distribution shifts is an important and timely topic. They raised several concerns, primary among them are:
1. readability of the paper: several of the reviewers noted that the paper is hard to parse, not easily readable and is not stand-alone in the sense that it requires consulting other papers extensively. The authors' response was promissory in nature and did not satisfy all the reviewers' concerns
2. Disjoint theory/practice: some of the reviewers found the presented theory rather disconnected from the practical implementation. The authors explained that the theory does not necessarily inform some practical algorithmic choices and clarified notational errors + practical norms in the DRO literature.
3. Novelty in light of related work: the authors clarified the position of their work against a backdrop of literature that largely considers fixed nominal environments

Note: I downweighted reviewer MPjF's score due to brevity and lack of engagement in the discussion. The other 3 reviewers provided ample feedback ensuring fairness in the review process.